# FORMULACODE: Evaluating Agentic Optimization on Large Codebases

**Atharva Sehgal** [1][*]   **James Hou** [2][*]   **Akanksha Sarkar** [3]   **Ishaan Mantripragada** [2]
**Swarat Chaudhuri** [1]   **Jennifer J. Sun** [3]   **Yisong Yue** [2]

## Abstract

Large language model (LLM) coding agents increasingly operate at the repository level, motivating benchmarks that evaluate their ability to optimize entire codebases under realistic constraints. Existing code benchmarks largely rely on synthetic tasks, binary correctness signals, or single-objective evaluation, limiting their ability to assess holistic optimization behavior. We introduce FORMULACODE, a benchmark for evaluating agentic optimization on large, real-world codebases with fine-grained, multi-objective performance metrics. FORMULACODE comprises 957 performance bottlenecks mined from scientific Python repositories on GitHub, each paired with expert-authored patches and, on average, 264.6 community-maintained performance workloads per task, enabling the holistic ability of LLM agents to optimize codebases under realistic correctness and performance constraints. Our evaluations reveal that repository-scale, multi-objective optimization remains a major challenge for frontier LLM agents. Project website at: https://formula-code.github.io.

## 1. Introduction

Large Language Models (LLMs) for code are rapidly evolving from isolated function-level synthesis to file-level editing, and now, to repository-level optimization (Merrill et al., 2026; Jimenez et al., 2024; Zhang et al., 2025; Zhao et al., 2024; Shetty et al., 2025; Ma et al., 2025). These models are now transitioning from assistants into autonomous coding agents, increasingly tasked with navigating complex, interconnected software ecosystems to diagnose bottlenecks and improve performance. However, we currently lack frameworks to study these emerging capabilities for the full optimization lifecycle; for example, how agents balance multiple workloads, maintain function integrity, and structure improvements at different levels of the codebase hierarchy.

While there exist coding benchmarks based on real GitHub repositories (Jimenez et al., 2024; Zhang et al., 2025; Zhao et al., 2024), they generally do not fully capture the multi-workload real-world tasks that engineers and researchers face in practice. These benchmarks often rely on binary pass/fail feedback, which is insufficient for measuring optimization, or synthetic (e.g., LLM generated) tasks, which lack the complexity and characteristics of real-world coding. For example, real-world optimization is rarely isolated, diagnosing and improving performance often requires reasoning about architectural decisions, component interactions, and design trade-offs on the system-level rather than tuning an isolated function (Balsamo et al., 2004; Woodside et al., 2007; Jin et al., 2012). Consequently, this requires a new evaluation standard capable of measuring the emerging ability of agents across this entire optimization workflow under realistic software engineering constraints.

We identify several directions for improving agentic coding benchmarking: (1) Fine-grained metrics: evaluation must move beyond binary correctness to capture continuous performance changes and trade-offs; (2) Real-world measurements: metrics should be derived from established execution environments (e.g., standard profiling suites) rather than synthetic proxies; (3) Reliable baselines: agent performance must be assessed against human optimization to provide a meaningful standard; and (4) Repository scale: agents must operate within large, evolving codebases.

We introduce FORMULACODE[1], a novel benchmark designed for advancing agentic optimization on large, evolving software ecosystems. FORMULACODE is constructed from 957 real-world performance bottlenecks mined from

*Equal contribution  [1]The University of Texas at Austin  [2]California Institute of Technology  [3]Cornell University. Correspondence to: Atharva Sehgal <atharvas@utexas.edu>.

*Proceedings of the 43rd International Conference on Machine Learning*, Seoul, South Korea. PMLR 306, 2026. Copyright 2026 by the author(s).

[1]FORMULACODE draws inspiration from Formula 1, where constructors must optimize entire systems—not just individual components—to achieve peak performance on the track. Similarly, FORMULACODE challenges code agents to perform holistic, codebase-level optimizations, reflecting the complexity and interdependence found in real-world software.

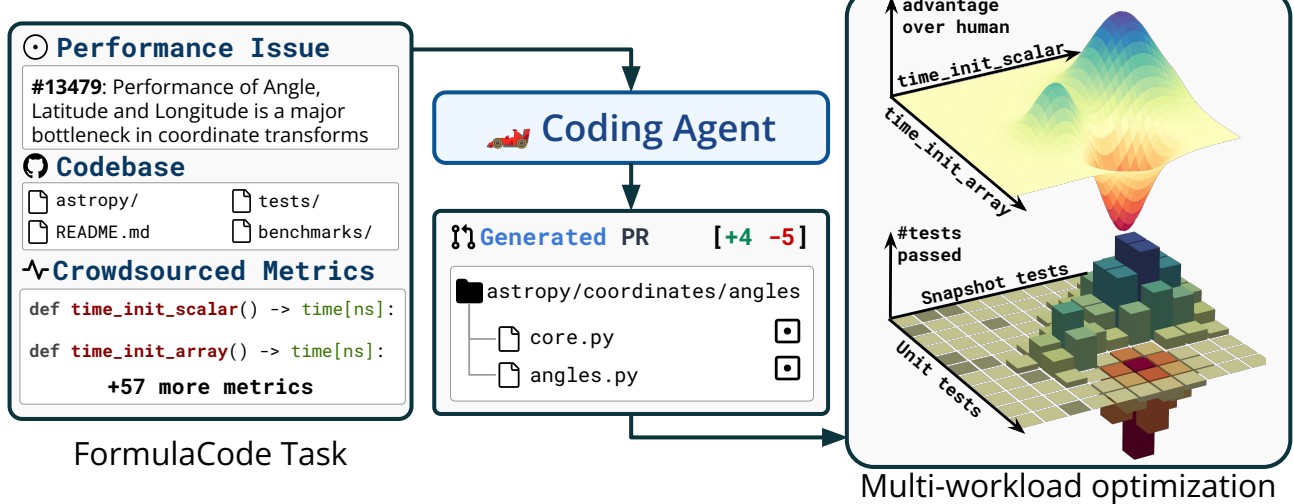

*Figure 1.* FORMULACODE is a continuously updating benchmark for evaluating the holistic ability of agents to optimize large codebases. Each task in FORMULACODE comprises a problem description of a performance regression from GitHub, an environment containing a baseline repository snapshot, and multiple expert-written crowdsourced performance workloads, along with the tools to execute them. An agent's performance improving edits are assessed based on their ability to outperform expert-written edits in optimizing *multiple workloads* while meeting multiple forms of correctness guarantees.

70 scientific, open-source Python repositories, like Pandas, Scikit-learn, and SciPy. Unlike previous datasets, each task in FORMULACODE is paired with an average of $264.6$ community-maintained performance workloads alongside expert-authored patches. This unique construction enables the use of the airspeed-velocity (asv) framework to assess the full lifecycle of optimization (triage, diagnosis, and resolution) in a way that isolated coding tasks cannot.

We conduct a large-scale evaluation of frontier and open weights models (GPT-5, Claude 4.0 Sonnet, Gemini 2.5 Pro, Qwen 3 Coder) within multiple agentic frameworks (Terminus 2, OpenHands). **Our main findings are:**

- Agents generally can improve run-time performance, but perform worse than experts (§4.1).

- Agents are better at local or function-level optimization, rather than repository-level optimization (§4.2).

- Agents excel at using specific optimization strategies (e.g., parallelizing or batching) and struggle with others (e.g., vectorized operations) (§4.3).

- Agent performance relative to experts can vary dramatically by popularity of the repository, performing worst on the 4th quintile and best on the 2nd quintile (§4.4).

- Despite being more expensive per call, agents using frontier LLMs are overall more cost effective than those using open weights models (e.g., due to open weights models having much longer reasoning chains) (§4.5.1).

- Compared to experts, agents negotiate multi-workload performance trade-offs less effectively (§4.5.2).

- We observe minimal effects from data leakage (i.e., using LLMs potentially trained on expert solutions) (§4.5.3).

We open-source FORMULACODE as a community resource[2], to not only measure what code agents can generate, but to understand how they can reliably optimize and maintain complex real-world systems.

## 2. FORMULACODE Benchmark Design

Each FORMULACODE task evaluates the ability of an agent to optimize a real-world codebase under strict correctness constraints. A task begins with a baseline repository, denoted $\text{Code}_0$, which represents the unmodified implementation. The agent operates on $\text{Code}_0$ and produces a modified version of the repository, denoted $\text{Code}_\text{agent}$, by making arbitrary repository-level edits.

Each task is paired with two forms of evaluation signals:

- **Correctness.** Correctness is measured via a suite of tests on the functional behavior. A proposed code modification is considered valid only if $\text{Code}_\text{agent}$ passes all tests that $\text{Code}_0$ passes.

- **Performance Workloads.** Each task includes a large collection of expert-written performance workloads that

---

[2]Project website at `https://formula-code.github.io/`.

exercise known performance-critical execution paths in the codebase. Each workload measures a single performance dimension, such as runtime or memory usage, and may exhibit natural variability due to execution noise.

Figure 1 depicts our benchmark setup. The top half shows a task from the Astropy repository, highlighting a performance issue with three functions: Angle, Latitude, and Longitude. There are 59 workloads defined by community-sourced expert-written metrics. The goal of the coding agent is to modify the repository to optimize these workloads while still maintaining correctness.

Performance evaluation proceeds by executing the full set of workloads on both $Code_0$ and $Code_{agent}$ and comparing their measured outcomes. Improving performance on one workload may degrade performance on others (Balsamo et al., 2004; Woodside et al., 2007; Jin et al., 2012). As a result, optimization in FORMULACODE is inherently multi-objective: agents must reason about trade-offs across subsystems and deliver improvements that are broad and consistent rather than localized to a single execution path.

## 2.1. Metrics

**Speedup.** For each $workload_i$, we compare the performance ratio of $Code_{agent}$ versus $Code_0$:

$$\text{speedup}_{agent,i} = \frac{\text{workload}_i(\text{Code}_0)}{\text{workload}_i(\text{Code}_{agent})}.$$

Having speedup $> 1$ indicates an improvement. These ratios are dimensionless and allow performance changes to be compared across heterogeneous workloads. If $Code_{agent}$ does not pass correctness tests for $workload_i$, then $speedup_i = 1$ (i.e., the modifications were reverted).

For $n$ workloads, the overall speedup is the geometric mean:

$$\text{speedup}_{agent} = \left( \prod_{\text{workload}_i} \text{speedup}_{agent,i} \right)^{\frac{1}{n}}. \quad (1)$$

**Advantage.** For each task, we also have expert-written code modifications, $Code_{expert}$. For example, the performance issue in Figure 1 was eventually resolved by an expert. We use the performance of $Code_{expert}$ as a reference point to characterize the difficulty of each task. We can then define the advantage of each $workload_i$ as:

$$\text{Adv}_i = \text{speedup}_{agent,i} - \text{speedup}_{expert,i}.$$

and subsequently, $Adv_{agent}$ is the geometric mean over $n$ workloads. Notice that if an agent had simply memorized the expert solution (e.g., due to training data contamination), then the advantage is zero. Indeed, the goal of super-human

optimization is to achieve a large positive advantage. Appendix Figure 26 provides a geometric intuition for this metric.

**Stratified Advantage.** We now turn to measuring advantage aggregated at different levels of granularity. We use $\ell \in \{0, 1, \ldots\}$ to denote the code hierarchy level.

- At the coarsest level ($\ell = 0$), we group workloads by entire modules such as `algorithms.*`.

- At finer levels, we group workloads under individual classes or functions (e.g., `algorithms.Sorting.*`, `algorithms.Sorting.time_sort_int.*`).

Each level $\ell$ thus partitions the workloads into groups: $\mathcal{G}^{(\ell)} = \{g_1^{(\ell)}, \ldots, g_{K_\ell}^{(\ell)}\}$, where each $workload$ belongs in some $g_k^{(\ell)}$. We can then define per-group advantage as:

$$\text{Adv}_{agent,g} = \text{speedup}_{agent}(g) - \text{speedup}_{expert}(g),$$

where $speedup_*(g)$ is defined using Equation 1 computed only over workloads in $g$. The *stratified advantage at level* $\ell$ is then the average across all groups at that level:

$$\text{Adv}_{agent}^{(\ell)} = \frac{1}{|\mathcal{G}^{(\ell)}|} \sum_{g \in \mathcal{G}^{(\ell)}} \text{Adv}_{agent,g}.$$

The family $\{\text{Adv}_{agent}^{(\ell)} | \ell \in \mathbb{Z}_{\geq 0}\}$ thus forms a multi-scale profile of an agent's performance. Because aggregation is performed over multiplicative speedup ratios within each group, $\text{Adv}_{agent}^{(\ell)}$ remains in the same metric family as the global advantage, but is sensitive to how performance gains are organized across the codebase hierarchy (Figure 15).

**Normalized Advantage.** Finally, we introduce a normalized version of advantage that explicitly accounts for noise and heterogeneity across workloads. Given the variance of the per-workload speedup ratios for an $agent$, $\sigma^2(\text{agent})$, we define the *normalized advantage* of an $agent$ as:

$$\widetilde{\text{Adv}}_{agent} = \frac{\text{Adv}_{agent}}{\sqrt{\sigma^2(\text{agent}) + \sigma^2(\text{expert})}}.$$

Conceptually, $\widetilde{\text{Adv}}_{agent}$ captures a signal-to-noise ratio of the agent advantage, and rewards consistency across workloads.

**Cost Weighted Metrics.** In practice, we also care about the inference budget of the optimization agent. We estimate the total inference cost as $\text{cost}_{agent} = c_{in} N_{agent}^{in} + c_{out} N_{agent}^{out}$ where $N_{agent}^{in}$ and $N_{agent}^{out}$ denote the total number of input and output tokens, and $c_{in}$ and $c_{out}$ are the per-token prices. This allows us to define the cost-weighted advantage:

$$\text{cost}(\text{Adv}_{agent}) = \frac{\text{Adv}_{agent}}{\text{cost}_{agent}},$$

*Figure 2.* Overview of FORMULACODE construction pipeline. FORMULACODE follows a four stage pipeline to identify real-world performance optimization tasks. (1) Scrape compliant repositories (§A.1). (2) Apply rule-based and LLM-based filters to identify candidate performance improvement pull requests (§A.2). (3) Construct reproducible Docker environments for each candidate (§A.5). (4) Validate each candidate for correctness and statistically significant performance improvement (§A.6). The pipeline is fully automated and updates FORMULACODE with new tasks every month.

which captures the human-relative improvement obtained per unit of inference budget. We will use these metrics in §4 to evaluate code optimization agents' performance on real-world codebases.

## 3. FORMULACODE: Dataset Construction

FORMULACODE consists of 957 multi-workload, real-world code optimization problems drawn from 70 repositories. In this section, we briefly describe an automated four-stage pipeline that extracts these problems from 105074 pull requests across 766 GitHub repositories. The key idea in each stage is to maximize recall as ambiguity is resolved in the last stage when we apply the expert-written patch and validate the performance improvement with statistical testing. Full details are presented in Appendix §A, Figure 2 provides an overview of the pipeline, and Table 1 summarizes key statistics about the final dataset.

### 3.1. Construction Pipeline

**Stage 1: Repository Scraping.** Our benchmarking apparatus relies on mature tools developed within the Python performance benchmarking community, for which a package's core developers write customized performance workloads in a pre-specified format. We can therefore identify both crowdsourced workloads and repositories with an established benchmarking procedure by searching for the presence of these tools, which yields 766 repositories (Appendix §A.1).

**Stage 2: Attribute Filtering.** From these repositories we scrape 105074 merged pull requests; rule-based filters then retain 26717 pull requests from 127 repositories that each reference at least one issue. We then construct a knowledge graph of the issues and comments referenced by each pull request (restricted to content created on or before the pull request) and an LLM agent analyzes it to gauge whether the pull request's primary intent is performance-oriented. This yields 3181 candidate performance-improving tasks from 101 repositories (Appendix §A.2, §A.3).

**Stage 3: Environment Synthesis.** Before validating a performance improvement, we must build and install a development copy of the package. This proves to be a non-trivial task, since scientific packages often require bespoke build processes, their build processes evolve as the project matures, and their documentation is fragmented across many files. We automate this with a reflexive LLM agent (Shinn et al., 2023) that iteratively refines a shell script to build an editable environment, and we aggressively cache and reuse previously synthesized scripts to lower the amortized cost. This yields 1232 tasks with reproducible Docker containers from 75 repositories (Appendix §A.5).

**Stage 4: Statistical & Correctness Validation.** For each task we apply the expert-produced patch and retain only tasks whose workloads exhibit a statistically significant speedup. We additionally provide two kinds of correctness tests: unit tests, which we manually validate for the 108 problems in FORMULACODE-V, and automated snapshot tests that compare each workload's return values against a reference, greatly increasing the correctness-verification surface of each task. This yields the final 957 tasks (Appendix §A.6).

### 3.2. Dataset Properties

**Multi-workload optimization.** Code optimizations rarely have isolated effects: an optimization in one part of the code can slow down another. FORMULACODE therefore

|  |  | Mean | Max |
|---|---|---|---|
| Issue Text | Length (Tokens) | 2718.03 | 15781 |
| Gold Patch | # Lines edited | 38.088 | 526 |
|  | # Files edited | 3.93 | 34 |
|  | # Func. edited | 6.06 | 54 |
| Workloads | # Eval. Fns | 264.58 | 1364 |
|  | % Coverage | 41.24% | 97.86% |

*Table 1.* (Micro-averaged) statistics characterizing different attributes of a FORMULACODE task instance. The average FORMULACODE gold patch requires 5.2 more lines of code spread over $1.29\times$ more files and $1.01\times$ more functions than the average SWE-Bench (Jimenez et al., 2024) patch.

frames performance optimization as a *multi-workload* problem: each task exposes on average 264.58 performance workloads (Table 1) presented to the agent alongside the problem description, and the agent is scored on the aggregate improvement across all of them, so it must reason about competing objectives rather than a single hot path.

**Live, contamination-resistant benchmark.** Data contamination is known to skew the performance of frontier models on code tasks mined from GitHub (Zhang et al., 2025). To mitigate this, FORMULACODE functions as a live dataset, updated with new problems each month. Tasks in FORMULACODE span 2017-10-21 to 2025-11-21, and their creation timestamps enable the temporal out-of-distribution analysis in §4.5.3.

**Hierarchical workloads.** Following the benchmark directory structure assigned by core developers, workloads are organized at three levels of granularity—module, class, and function (Figure 15)—which lets us aggregate workloads by semantic grouping for the scale-resolved, stratified advantage analysis in §4.2.

**Task diversity.** Restricting collection to a hand-curated set of repositories creates a cumulative "Matthew effect" (Koch et al., 2021) that disconnects a benchmark from the broader task distribution. FORMULACODE instead samples performance benchmarks from a large set of repositories, based on whether they satisfy the four pipeline stages above. We report the composition of the dataset in Appendix §A.8.

## 4. Experiments

We organize our experimental findings into three categories.

- First, we present overall performance metrics to investigate whether agents can achieve meaningful runtime speedups and whether they can outperform experts.
- Second, we provide detailed breakdown of agent capabilities, examining performance across optimization strategies, optimization scope, and repository popularity.
- Third, we present findings on cost-effectiveness, multi-workload optimization, and data leakage.

We compare four frontier LLMs – GPT-5 (Singh et al., 2025), Claude 4.0 Sonnet (Anthropic, 2025), Gemini 2.5 Pro (Comanici et al., 2025), and Qwen 3 Coder (Yang et al., 2025) – under two LLM Frameworks – Terminus 2 (Merrill et al., 2026) and OpenHands (Wang et al., 2025). Terminus 2 is evaluated with all four models, while OpenHands is evaluated with GPT-5, Claude 4.0 Sonnet, and Qwen 3 Coder. Additional discussion of model and framework choices appears in Appendix §B.2. Evaluations are conducted on FORMULACODE-V due to compute constraints, using the metrics defined in §2. Full experimental details and additional analyses are provided in Appendix §B.

### 4.1. Global Leaderboard

For each agent–model configuration, we compute the human-relative advantage Adv and normalized advantage $\widetilde{\text{Adv}}$ defined in §2. We then aggregate configurations into a global leaderboard using the Ranked Pairs (RP) method (Tideman, 1987), yielding a transitive ordering. Table 2 summarizes the resulting rankings.

*Observation: Agents achieve non-trivial speedups over the baseline.* All evaluated configurations attain speedup $> 1$ on FORMULACODE-V relative to the baseline codebase (associated with the issue), indicating that agents can successfully identify and implement runtime-relevant changes.

*Observation: Agents underperform experts on performance optimization tasks.* For all agents, the overall advantage, Adv, is negative, indicating a fundamental performance gap. We also notice a disagreement between the Adv and speedup metrics for many configurations, where large performance gains on certain 'easier' tasks have a disproportionate influence on the global speedup score. The influence of such tasks is diminished in the Adv score, which compares each agent improvement to the corresponding expert improvement; since tasks that are "easier" typically also admit larger expert speedups, this relative metric yields a more consistent difficulty reference.

### 4.2. Large-Scale vs. Small-Scale Refactors

To disentangle performance by optimization scale, we use the hierarchical structure of FORMULACODE-V workloads (Figure 15) and stratified advantage $\text{Adv}_{\text{agent}}^{(\ell)}$ from §2. We construct per-configuration profiles across three strata: Module level aggregation ($\ell = 1$), Class level aggregation ($\ell = 2$), and Function level aggregation ($\ell = 3$). For each configuration and level $\ell$, we compute group-level speedups and advantages, shown in Figure 3.

*Observation: Agents demonstrate characteristic performance profiles.* In Figure 3 models exhibit diverse performance profiles. OpenHands + Claude 4.0 Sonnet performs best at the module-level optimization but underperforms

| Agent | Model | RP Rank (Adv) ↓ | Adv ↑ | $\widetilde{\text{Adv}}$ ↑ | speedup ↑ |
|---|---|---|---|---|---|
| Terminus 2 | GPT-5 | 7 | -0.0504 | -0.1387 | 1.0585 |
| | Claude 4.0 Sonnet | 4 | -0.0410 | -0.1065 | 1.0987 |
| | Gemini 2.5 Pro | 6 | -0.0433 | -0.1138 | 1.0963 |
| | Qwen 3 Coder | 5 | -0.0454 | -0.1257 | 1.0677 |
| OpenHands | GPT-5 | 3 | -0.0209 | -0.0702 | 1.0825 |
| | Claude 4.0 Sonnet | 1 | -0.0112 | -0.0483 | 1.0539 |
| | Qwen 3 Coder | 2 | -0.0301 | -0.1529 | 1.0346 |
| Human Expert | - | - | **0.0000** | **0.0000** | **1.1040** |

*Table 2.* Global leaderboard of agent-model configurations on FORMULACODE-V. We report the Ranked Pairs (RP) position induced by human-relative advantage (Adv), the normalized advantage ($\widetilde{\text{Adv}}$), and the speedup (speedup) as defined in §2.

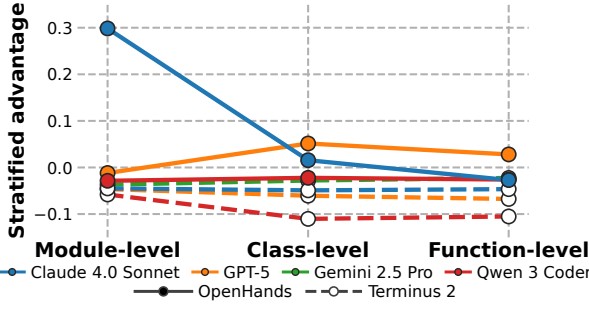

*Figure 3.* Showing stratified advantage across hierarchy levels for each agent–model configuration. Each line traces the stratified advantage ($\text{Adv}_{\text{agent}}^{(\ell)}$) over $\ell \in \{1, 2, 3\}$, revealing whether a configuration prefers coarse module-level changes or fine-grained function-level edits.

at the function-level, indicating that this configuration can overlook small-scale optimizations in favor of large-scale ones. Conversely, OpenHands + GPT-5 performs best at the function-level but loses effectiveness at the module-level.

*Observation: Agents are comparatively stronger on local optimizations.* With few exceptions (notably Claude 4.0 Sonnet + OpenHands), configurations achieve higher stratified advantage at function-level aggregation.

## 4.3. Type of Optimization Problem

We investigate whether models can outperform human experts on particular classes of optimizations. For each problem in FORMULACODE-V, we label the optimization attempted by the expert patch using an LLM (see §A.8 for details). Next, we aggregate the advantage of each agent-model pair within each optimization class. Table 3 summarizes the results.

*Observation: Some optimization classes remain systematically difficult for agents.* We observe certain optimization categories where agents outperform experts. Specifically, all agents were able to find faster solutions in tasks where the expert attempted a parallelization or batching based solution. Conversely, all agents struggle when the human solutions require delegating to lower-level system implementations

(C extensions, vectorized operations).

## 4.4. Long-Tail Generalization Across Repository Popularity

We next study how performance varies by repository popularity (measured using GitHub stars). We compute advantage statistics for each popularity quintile.

*Observation: Agents underperform experts on tail repositories.* Agent performance is lower in the first popularity quintile (Q1; bottom 20%), which comprises repositories with 133-202 GitHub stars. Expert patches, however, yield comparatively large gains in this regime: $\text{speedup}_{\text{expert}}(Q1) = 1.1104$, the second-largest speedup across quintiles. One hypothesis is that smaller repositories contain more heterogeneous, high-impact micro-optimizations that may have already been discovered in larger, more mature repositories, leading to more variable (but sometimes high-impact) optimization opportunities. A second plausible hypothesis is distribution shift: smaller repositories may be less represented in training corpora, reducing agent effectiveness.

*Observation: Agents are most competitive on mid-popularity repositories.* In the 20th to 60th percentile range, mean advantages are closest to expert performance, and some configurations perform comparably with experts. We hypothesize that this is due to two reasons. First, moderately popular repositories more closely match the agent's training distribution than tail repositories. Second, these repositories have more unexploited optimization avenues relative to highly popular projects.

*Observation: Performance dips in high-popularity repositories.* Agent performance is lowest in the fourth quintile (Q4; 6,371-10,343 stars). In this regime, expert patches also yield the smallest gains: $\text{speedup}_{\text{expert}}(Q4) = 1.0822$, the lowest expert speedup across all quintiles. This pattern indicates reduced remaining optimization headroom in these repositories, where many simpler improvements may have already been realized. Additionally, slight distribution shift may persist and limit agent effectiveness.

*Table 3.* Per-tag advantage for each agent–model configuration. Columns correspond to optimization tags (see 7), and cells report the human-relative advantage restricted to workloads whose patches are annotated with the respective tag. OpenHands + GPT-5 shows strong advantage on algorithmic rewrites and data-structure changes, while other models perform comparatively better on micro-optimizations or caching.

| Agent | Model | Algo | Data | Lower | Approx | Parallel | Reduce | Cache | Batch | Scale | DB | Micro | I/O | Higher | Uncat |
|---|---|---|---|---|---|---|---|---|---|---|---|---|---|---|---|
| Terminus 2 | GPT-5 | -0.064 | -0.112 | -0.233 | – | 0.010 | -0.006 | -0.054 | 0.028 | – | – | 0.001 | – | -0.002 | – |
| | Claude 4.0 Sonnet | -0.019 | 0.011 | -0.720 | – | 0.013 | -0.028 | -0.048 | 0.041 | – | – | -0.038 | – | -0.009 | – |
| | Gemini 2.5 Pro | -0.029 | 0.011 | -0.676 | – | 0.013 | -0.028 | -0.048 | 0.041 | – | – | -0.038 | – | -0.007 | – |
| | Qwen 3 Coder | -0.023 | 0.007 | -0.455 | – | 0.007 | -0.079 | -0.027 | 0.042 | – | – | -0.066 | – | 0.005 | – |
| OpenHands | GPT-5 | 0.015 | -0.052 | -0.211 | – | 0.015 | -0.051 | -0.018 | 0.040 | – | – | -0.018 | – | -0.008 | – |
| | Claude 4.0 Sonnet | -0.028 | 0.023 | -0.180 | – | 0.007 | -0.049 | -0.017 | 0.047 | – | – | 0.086 | – | -0.005 | – |
| | Qwen 3 Coder | -0.020 | -0.004 | -0.203 | – | 0.012 | -0.016 | -0.019 | 0.051 | – | – | -0.063 | – | 0.013 | – |

*Table 4.* Performance across repository popularity quintiles (by GitHub stars). We report $\text{Adv}_{\text{agent}}$ for workloads drawn from repositories in each quintile, from least popular (Q1) to most popular (Q5). Red signifies worse performance.

| Agent | Model | Q1 | Q2 | Q3 | Q4 | Q5 |
|---|---|---|---|---|---|---|
| Terminus 2 | GPT-5 | -0.0194 | 0.0423 | -0.0045 | -0.2754 | -0.0123 |
| | Claude 4.0 Sonnet | -0.0450 | -0.0062 | 0.0025 | -0.3529 | -0.0220 |
| | Gemini 2.5 Pro | 0.0077 | -0.0062 | 0.0024 | -0.3311 | -0.0445 |
| | Qwen 3 Coder | -0.0691 | 0.0052 | -0.0179 | -0.1669 | -0.0332 |
| OpenHands | GPT-5 | -0.0387 | 0.0315 | 0.0072 | -0.0769 | -0.0068 |
| | Claude 4.0 Sonnet | -0.1041 | 0.0291 | -0.0200 | -0.0378 | 0.0263 |
| | Qwen 3 Coder | -0.0159 | 0.0137 | 0.0227 | -0.0878 | -0.0270 |

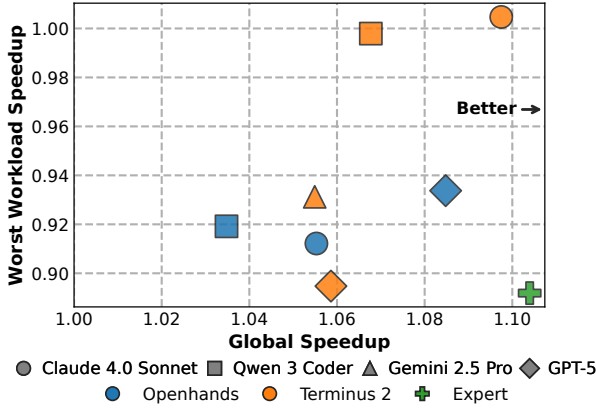

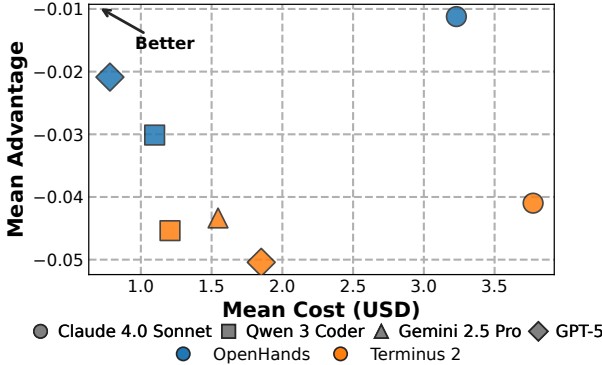

*Figure 5.* Multi-workload tradeoff performance of agent-model configurations. We quantify a model's speedup performance as a function of its worst regression. The expert patch achieves the highest speedup while negotiating considerably high workload regressions.

*Figure 4.* Cost-Performance tradeoff of agent-model configurations. As most agents struggle on code optimizations tasks, the pareto set is primarily dominated by the most expensive model (Claude 4.0 Sonnet).

### 4.5. Practical Considerations

#### 4.5.1. COST EFFICIENCY.

Frontier models differ substantially in end-to-end inference cost due to provider pricing and the number of tokens consumed by a given agent configuration. In this experiment, we consider the cost–performance tradeoff within our agent configurations using the cost-weighted objectives defined in §2. Table 14 reports a leaderboard based on cost-weighted normalized advantage, and Figure 4 summarizes the resulting trade-off.

*Observation: Higher-priced models rank best under the*

*cost-weighted objective.* When weighted by cost, top-ranked configurations tend to use the higher-priced (and more capable) models. A contributing factor is that lower-capability models often consume more tokens within the agent loop, which can offset lower per-token prices. This might also indicate that smaller models lack the capabilities to reason effectively about performance optimizations.

#### 4.5.2. MULTI-WORKLOAD TRADEOFF PERFORMANCE.

Performance optimization necessitates a holistic understanding of competing workloads. In this experiment, we compare the global speedup achieved by a model with the largest regression it causes. For each agent–model configuration, we compute (i) global speedup aggregated across tasks and workloads, and (ii) the average *worst-workload* speedup, defined as follows: for each task, we take the minimum speedup across the task workloads, and then average this minimum across tasks. Figure 5 plots these two quantities.

*Observation: Multi-workload optimization remains challenging for agents.* Despite causing large regressions, human code edits achieve the best global speedup, indicating

*Table 5.* Temporal analysis of model performance across knowledge cutoff boundaries. Each column represents a temporal bin defined by distance (in months) from the model's training data cutoff; values indicate mean global speedup ($\text{speedup}_{\text{agent}}$) within each bin. We find no consistent drop in performance.

| Model | Before Cutoff | | | After Cutoff | | |
|---|---|---|---|---|---|---|
| | 6+ mo | 3-6 mo | 0-3 mo | 0-3 mo | 3-6 mo | 6+ mo |
| Claude 4.0 Sonnet | 1.0892 | 1.0564 | 0.9966 | 1.0915 | 1.0951 | 1.0519 |
| GPT-5 | 1.1708 | 1.0454 | 0.9871 | 1.0378 | 1.0679 | 1.0500 |
| Gemini 2.5 Pro | 1.1071 | 0.9989 | 1.0219 | 1.0523 | 1.1063 | 1.0251 |

a superior ability to negotiate multi-workload performance tradeoffs than our configurations.

### 4.5.3. TEMPORAL GENERALIZATION.

**Motivation.** FORMULACODE is a live benchmark: tasks are continuously added and include creation timestamps. This enables us to probe the temporal out-of-distribution behavior of agents on performance optimization tasks. Related work on code correctness finds large gains when tasks are present in training corpora (Jain et al., 2024a).

We bucket tasks by their month of creation and compute mean global speedup in windows defined by the temporal distance to each model's knowledge cutoff (§B.2). We use 3-month bins and consider bins up to 6 months before/after the cutoff. Table 5 summarizes results.

*Observation: Limited evidence of a cutoff-aligned leakage effect.* Performance shows no consistent shift when moving from pre-cutoff to post-cutoff task creation dates, suggesting the gap is capability-based rather than data-based.

## 5. Related Work

**Algorithms for Code optimization.** There is a long history of research on iterative code optimization using execution feedback. Classical approaches to this problem were based on stochastic search and constraint solving (Schkufza et al., 2013; Sasnauskas et al., 2018). Among deep-learning based approaches, AlphaTensor and AlphaDev produce super-optimized matrix multiplication and sorting routines, respectively (Fawzi et al., 2022; Mankowitz et al., 2023). These systems combine large, publicly sourced pretraining datasets with carefully chosen inductive biases to make optimization faster. The more general *agentic Optimization* workflows operate by iteratively running LLM-generated code, evaluating the output, and feeding the output back to the model. Terminus 2 and OpenHands represent two such configurations out of many that benefit from iterative feedback (Yao, 2024; Yang et al., 2024; Merrill et al., 2026; Wang et al., 2025; Merrill & Shaw, 2025). FORMULA-CODE is the first benchmark purpose built to assess the multi-workload optimization ability of such agentic AI algo-

rithms in real-world codebases and provides the fine-grained evaluation functions needed for iterative optimization.

*Evolutionary Optimization* algorithms equipped with LLMs (Romera-Paredes et al., 2024; Grayeli et al., 2024) iteratively improve a candidate pool of programs using execution feedback. Systems like AlphaEvolve (Novikov et al., 2025) and OpenEvolve (Sharma, 2025) demonstrate that such agents can efficiently discover and refine novel, high-performance code-based heuristics across diverse scientific domains. These methods are scalable but require high quality evaluation functions to penalize degenerate solutions. While FORMULACODE provides the necessary evaluation functions, we could not benchmark evolutionary methods due to their substantial compute needs.

**Code Generation Benchmarks.** Coding benchmarks can be differentiated by their synthesis scope. For a list of differences, consult Table 6.

*Function and file level.* HumanEval (Chen et al., 2021) and MBPP (Austin et al., 2021) present hand-written programming problems in Python with corresponding unit tests. Many contributions extend these benchmarks to have more testing (Liu et al., 2023), broader scope (Yin et al., 2022; Yang et al., 2023), and more task diversity (Muennighoff et al., 2023; Lai et al., 2022; Zan et al., 2022). CruxEval (Gu et al., 2024) benchmarks the code execution and reasoning ability of LLMs more deeply. LiveCodeBench (Jain et al., 2024a) attempts to mitigate data-leakage by annotating problems with release dates. All these benchmarking efforts utilize unit testing suites to gauge program correctness. FOR-MULACODE *supplements* the evaluation signal provided by the above datasets by using community-maintained evaluation functions that continually update with each commit.

*Repository level.* Function and file level benchmarks evaluate coding ability on self-contained coding tasks. However, real software issues typically span multiple modules and files. Repository level benchmarks (Jimenez et al., 2024; Tang et al., 2024; Jain et al., 2024b; Shetty et al., 2025) aim to preserve the inherent challenges in real-world software engineering beyond text completion, such as finding relevant files, capturing relationships between modules, tracing information flow, etc. SWE-Bench (Jimenez et al., 2024) collects GitHub issues from popular repositories and evaluates coding agents' ability to resolve the issues. Follow-up efforts benchmark agents on repository-conditioned code synthesis (Tang et al., 2024), scale-up benchmarking by admitting smaller codebases with LLM-generated unit tests (Jain et al., 2024b), and introduce continually updating pipelines for the task (Zhang et al., 2025). Such extensions provide valuable insights into LLM agent behavior yet ground their evaluations in correctness tests, that present a discrete optimization surface for the agents. FORMULACODE *complements* these benchmarks by assessing agents on community-maintained

*Table 6.* Comparing FORMULACODE with related codebase benchmarks. FORMULACODE is the only benchmark that satisfies the desired properties for evaluating LLM agents on real-world code optimization tasks. ++ denotes continually updating benchmarks. Data is sampled from real distributions like GitHub (⬤), Leetcode (⇌), AtCoder (👥), and Codeforces (▮▮); and LLM-generated or synthetic distributions (🎰). An extended analysis is presented in §5.

| Benchmark | Evaluation framework | # Tasks | # Workloads / Task | Live updates | Data source | Search space | Synthesis scope | Leakage resistant? |
|---|---|---|---|---|---|---|---|---|
| GSO-Bench | Performance | 102 | Single | ✗ | ⬤ ; 🎰 | Large | Repo | ✗ |
| SWE-Bench | Unit Tests | 2292 | - | ✗ | ⬤ | Small | Repo | ✗ |
| LiveCodeBench | Unit Tests | 300 [++] | - | ✓ | ⇌ ; 👥 ; ▮▮ | Small | File | ✓ |
| SWEfficiency | Performance & Unit Tests | 400 | Single | ✗ | ⬤ | Large | Repo | ✗ |
| SWE-Perf | Performance & Unit Tests | 140 | Single | ✗ | ⬤ | Large | Repo | ✗ |
| CruxEval | Unit Tests | 800 [++] | - | ✗ | 🎰 | Small | File | ✓ |
| FormulaCode | Performance & Unit Tests | 957 [++] | 264.58 | ✓ | ⬤ | Large | Repo | ✓ |

evaluation functions that present a smoother optimization landscape and higher fidelity than unit tests.

**Optimization Benchmarks.** There are prior benchmarks for efficient code synthesis on function and file-level tasks. COFFE (Peng et al., 2025) samples tasks from HumanEval, MBPP, CodeContests, and APPS (Chen et al., 2021; Austin et al., 2021; Hendrycks et al., 2021) and auto-generates stress tests while ECCO (Waghjale et al., 2024) curates a function and file-level efficient synthesis dataset from IBM CodeNet (Puri et al., 2021) with data-mined test cases.

Recent repository-level benchmarks like GSO-Bench (Shetty et al., 2025) and SWEfficiency (Ma et al., 2025) also study LLM agents' ability to optimize code. However, these benchmarks only optimize for a single target function at a time. SWE-Perf (He et al., 2025) is a closely related concurrent benchmark comprising 140 instances derived from performance-improving PRs across 9 Python repositories, measuring performance improvement via unit test runtime. FORMULACODE differs in scale (957 tasks from 70 repositories), evaluation signal (~264.58 dedicated performance workloads per task vs. unit test timing as a proxy), and contamination resistance (live monthly updates). In contrast to all prior optimization benchmarks, FORMULACODE focuses on: (1) using community-maintained benchmarks specifically designed to profile performance inefficiencies instead of using hand-curated stress tests or unit test timing, (2) benchmarking on repository-level codebases, which better capture the natural challenges with real-world code optimization, and (3) presenting multiple workloads that can compete with one another to assess the holistic optimization ability of agents.

## 6. Conclusion

We present FORMULACODE, a comprehensive coding benchmark for repository-level agentic optimization. In this benchmark, coding agents must not only write code that

passes standard correctness tests, but also improve runtime, and our benchmark design enables us to study the impact of repository popularity, temporal cutoffs, and multi-scale optimization to guide the design of future agents capable of surpassing human experts. As code-writing agents become more capable at the repository-level, FORMULACODE provides a rigorous foundation for development. To ensure longevity and prevent saturation, we operate as a live benchmark, continually ingesting new tasks to test agents against an evolving human baseline. Our evaluations show that FORMULACODE is a challenging benchmark for frontier LLMs and agentic frameworks, leaving open significant room for future agent development.

**Limitations.** Our conclusions are bounded in four ways, which we detail in Appendix §A.9. First, FORMULACODE currently covers scientific Python repositories that ship Airspeed Velocity (ASV) benchmarks so many applications like web frameworks, distributed systems, embedded software, and other languages remain out of scope. Extending FORMULACODE to these domains is an interesting problem for future work. Second, the fidelity of our performance signal is bounded by the quality of the community-maintained workloads: a patch's measured gain reflects only the code paths that the workloads exercise, so speedups on uncovered paths go unmeasured. The large number of workloads per task (264.6 on average) keeps this coverage broad, though not exhaustive. Third, due to compute constraints our full agent evaluation runs on the FORMULACODE-V subset (108 tasks) rather than the complete 957-task benchmark; FORMULACODE-V preserves the repository and difficulty distribution of FORMULACODE, but results may not transfer perfectly to the full benchmark. Fourth, all FORMULACODE tasks are run on a specific AWS configuration, so absolute measurements are hardware-specific—advantage, computed as a difference of speedup ratios on identical hardware, is more robust to this than raw timing, but cross-platform behavior remains untested.

## Acknowledgements

This work was supported in part by a Laude Institute Slingshot Award, NSF awards III-#2505097, PPoSS-#2316161, NSF #2505096, NSF #2505098, and gifts from Point72 and OpenAI.

We also thank Alex Shaw, Braden Hancock, Miles Cranmer, Neehar Kondapaneni, Rogério Guimarães, Anant Asthana, Arjun Sharma, Alex Farhang, and Markus Marks for helpful discussions.

## Impact Statement

We have presented FORMULACODE: a benchmark for measuring the capabilities of LLM-guided agents to optimize performance on large codebases. FORMULACODE is designed to serve two audiences: researchers (those developing new LLMs / Agents) and practitioners (those using Agents for daily workflows). For researchers, we hope that FORMULACODE accelerates the development of coding agents by providing contamination-free training and evaluation signals. For practitioners, we hope FORMULACODE offers *comparative* metrics that gauge the utility of LLMs and agents in specialized repositories under diverse cost-performance constraints. In this section, we discuss the broader societal impacts and ethical considerations of our work.

*Potential for Misuse.* Benchmark results are only as reliable as the interpretations drawn from them. To ground evaluations in realistic developer workflows, we use community-maintained workloads that already exist in each repository and attempt to preserve the same information and performance instrumentation available to a human contributor. This design also supports practical impact: strong model-generated changes can, in principle, be merged upstream to reduce maintenance burden, particularly for smaller repositories after thorough manual analysis. At the same time, reliance on repository workloads introduces an attack surface: an adversary could submit pull requests that alter or add workloads to make tasks artificially easier. While such additional workloads can increase regression coverage (thereby providing some downstream utility), practitioners should treat workload provenance and review practices as part of the evaluation's trust boundary.

*Privacy Concerns.* FORMULACODE is an 'open-book' benchmark and necessarily includes interactions from open-source software developers. We include such context to provide models access to the same information a human would use when solving these tasks. Although we anonymize usernames and remove personally identifiable information to the best of our ability, some contributors may remain indirectly identifiable via secondary cues (e.g., writing style, repeated project-specific references).

*Bias and Fairness.* Benchmarks can incentivize and influence which capabilities are prioritized by the community. We strive to make FORMULACODE's metrics explicit and stable, and we apply statistical analyses to reduce unintended measurement artifacts. Yet, FORMULACODE inherits limitations from the underlying repository benchmarks. In particular, FORMULACODE is susceptible to a form of the Quantitative Fallacy: aspects of agent competence that are difficult to measure may be underweighted or omitted, inflating the true utility of such algorithms. This is a limitation of all execution-based benchmarks. We therefore recommend using FORMULACODE as a *complementary* signal rather than as a substitute for careful manual assessment of Agent / LLM behavior.

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

# A. Dataset Construction Details

In this section, we provide details on the dataset construction process. Our core aim is to provide an automated pipeline for constructing a dataset of pull requests that are relevant for performance benchmarking. The dataset was constructed on a single machine with Ubuntu 22.04 LTS running on a machine with 503 GiB RAM, a dual-socket Intel Xeon Platinum 8352Y CPU @ 2.20 GHz (128 hardware threads), equipped with 4xNVIDIA A40 GPUs (46 GiB VRAM each). Making the dataset from scratch takes $\sim 32$ hours, consuming $\sim 100$ GB of disk space for the metadata and $\sim 2$ TB of disk space for the docker image cache.

We use two LLMs during the dataset construction process. For less complex tasks such as textual classification and extraction, we use `openai/gpt-oss-120b` model served locally (Kwon et al., 2023; OpenAI et al., 2025). For complex tasks such as environment build script synthesis, we first attempt to use the local LLM and fallback to the `anthropic/claude-3-5-sonnet-20241022` (Anthropic, 2024) model (with a one-time total cost of $\$446$ for the entire dataset). The additional cost may change if a different locally available LLM is utilized.

## A.1. Repository Scraping

We identify compliant repositories by searching for the presence of mature tools developed within the Python performance benchmarking community. To search for these repositories at scale, we develop a CommonSQL script to search for the presence of performance-oriented tools and workloads in the GitHub Public Dataset on Google BigQuery (GitHub & Google Cloud Platform, 2025), which snapshots about $2.8 \times 10^6$ open-source repositories and $2 \times 10^9$ code files. We add additional filters to ensure only mature software packages are considered. Specifically, we ensure that each valid repository has (1) Markers identifying the presence of at least one performance workload (e.g., `asv.conf.json`); (2) does not fork an existing repository. (3) Presence of PR merges and active maintenance in the last three years. (4) Support for Python 3.8+. This leaves us with 766 repositories.

The CommonSQL script executes in about $48$ seconds and cost $\$9.4$. As an alternative, we can also use the GitHub Search API to query for the repositories. This yields the same number of repositories, but can be much slower due to API rate limits.

## A.2. Rule-based Filtering

Once we have a list of compliant repositories, it is technically possible to execute and measure the performance of all pull-requests in the repository. However, as most pull-requests do not primarily intend to improve performance, this leads to unnecessary waste of compute resources. The rule-based filtering stage ensures that we collect performance metrics for only those pull requests where we can ensure that the pull request is suitable for benchmarking. Most filters in this stage aim to identify unambiguous signals that disqualify a pull request from being used for benchmarking. The prominent filters are listed below:

- **Repository Compliance:** We select repositories that have at least 100 GitHub stars. Below 100 stars, we found that repositories often lacked the necessary community engagement to produce good quality pull requests.

- **Pull Request Status:** We strictly filter for pull requests that have been successfully merged (`state='closed'` with a valid `merged_at` timestamp) within the target date range. We also ensure that we can retrieve and successfully apply the patch to the repository.

- **Benchmarking Infrastructure:** The specific commit tree must contain an Airspeed Velocity (ASV) configuration file (`asv.conf.json`), ensuring the repository supported benchmarking at that point in history.

- **Core Content:** We explicitly exclude commits that only touch non-functional paths, such as `tests/`, `docs/`, `examples/`, `.github/`, `dist-info/`, build artifacts, or packaging metadata (e.g., `pyproject.toml`, `requirements.txt`).

- **Heuristic Message Filtering:** We apply a regex-based pre-filter to the commit message. Commits matching "negative" patterns (e.g., "revert", "release", "bump version", "fix typo", "formatting") are discarded unless they also contain "positive" performance keywords (e.g., "speed", "optimize", "latency", "throughput", "memory", "vectorize"). Ambiguous messages are retained for LLM classification.

- **Complexity Constraints:** To ensure feasibility for both the LLM context and the build system, we exclude commits that change more than 500 files or 80,000 lines of code, or where the patch size exceeds an acceptable context window for a capable local LLM (64,000 tokens). These constraints can be adjusted based on the future capabilities of LLMs.

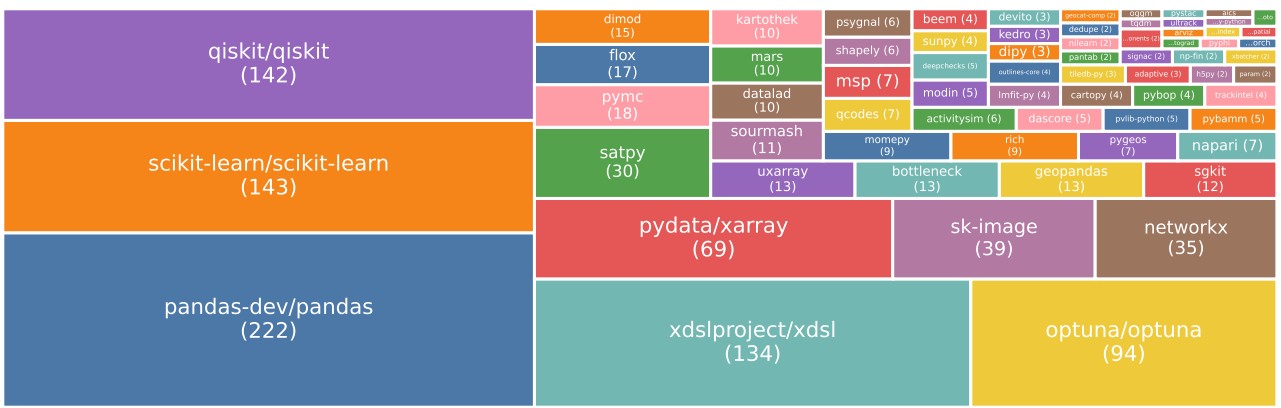

*Figure 6.* Distribution of tasks across repositories in FORMULACODE till November, 2025. FORMULACODE comprises of 957 tasks sampled from 70 diverse open source GitHub repositories. Most repositories are software tools used extensively within scientific communities. FORMULACODE shows a strong long-tail pattern of bespoke repositories that are rarely covered in contemporary code-generation datasets. Table 10 presents a detailed overview.

- **Build Environment:** We clone each repository at the specific commit tree and attempt to build it using uv. uv is a fast python package manager that can be used to install dependencies from a project's dependency files (e.g., pyproject.toml, requirements.txt, or setup.py). If the build fails, we discard the pull request. If the build succeeds, we pin the dependencies to ensure that the build environment can be reproduced. This is a compute-intensive process and, after parallelizing the build process, requires ∼ 13 hours for all pull requests on our machine.

After applying these filters, we are able to select 26717 pull requests from 127 repositories that are suitable for benchmarking.

### A.3. Performance Intent Filtering

The previous stage ensures that we only select pull requests that are suitable for benchmarking. However, it is still possible that the pull request does not primarily intend to improve performance. To ensure that we only select pull requests that are suitable for benchmarking, we utilize a pre-trained local LLM to classify the pull request as performance improving.

The primary objective of this classifier is to filter out pull requests that pass the regex-based heuristic but are not *bona fide* performance optimizations. Common examples of such false positives include commits that contribute new features instead of improving performance, refactor code structure without runtime impact, or maintainability improvements. The classifier analyzes the pull request description, file change summary, and the code patch to make this determination.

The classifier is written in DSPy (Khattab et al., 2023) and the prompt is shown in Figure 13. We explicitly prioritize recall over precision. The prompt is configured to lean towards a "YES" classification in ambiguous cases. This design choice is deliberate, as false positives will be symbolically verified in the subsequent benchmark execution stage, and discarded if they yield no measurable speedup.

### A.4. Problem Statement Construction

To transform a raw pull request into a benchmark task, we must construct a clear, self-contained problem statement that defines the performance goal. We employ a multi-stage pipeline to aggregate context and extract a structured narrative.

**Context Aggregation.** For each candidate pull request, we scrape all available metadata (title, body, labels, and comments, date of creation, and date of merge) that can be used to construct the problem statement. We also fetch the file change summary and the raw patch content to ground the problem statement in the actual code changes. We parse the pull request body and comments to identify linked issues (e.g., #123, owner/repo#123). These references are resolved to their full issue descriptions and discussions, which are also parsed and aggregated into the problem statement. We only include information that was available before or at the time the pull request was created to ensure that the problem statement is self-contained.

**Context Filtering.**    Before attempting extraction, we enforce a strict validity check: a pull request must have at least one linked issue or a descriptive body. The rationale for this constraint is twofold. First, the linked issue typically provides the problem context (the bug report, performance regression analysis, or feature request) that motivated the change. Second, a descriptive pull request provides details of the problem solved, the methodology used, and the solution, which can be helpful for computing metadata for the benchmark task as well as clarifying the overall task goal.

**Context Extraction.**    We consolidate all linked issues into a single document using a static template (shown in Figure 8). In principle, the issue text alone should sufficiently describe the initial observed performance regression or bottleneck. However, in practice, we find that while an issue provides the initial observed performance regression or bottleneck, they frequently bundle multiple optimization directions that are implemented across several pull requests. As a result, a problem statement derived only from the issue can under-specify the problem statement's starting state, leading to an ambiguous task (an agent may optimize a different aspect than the original change).

To ensure that each problem statement provides a clear and self-contained description of the problem, we use another specialized LLM-based classifier to extract relevant problem context from the pull request description. We instruct the agent to specifically extract near-verbatim sentences corresponding to the performance goal and constraints relevant to this PR. Each extracted sentence is symbolically verified to maintain a high degree of textual fidelity (High Longest Common Subsequence ratio) to preserve technical terms, error messages, and code snippets. Any pull request that fails to yield a valid problem context is discarded as it lacks a defined starting state for the benchmark. This LLM-based extraction agent is implemented using DSPy, and the prompt is shown in Figure 10.

**Examples.**    Figure 8 shows problem statements for some FORMULACODE tasks. Each problem statement has an initial set of static instructions, information about the problem extracted from the linked issues, and the initial direction of optimization extracted from the pull request description.

The problem statement construction (§A.4) and the performance intent filtering (§A.3) stages are applied together to yield 3181 problems.

### A.5. Synthesizing Reproducible Environments

**Motivation.**    A critical challenge in benchmarking historical commits is that the build environment (dependencies, compilers, and system libraries) is often implicit and evolves over time. Simply installing the package via `pip` is insufficient for performance benchmarking for two main reasons: First, performance-critical Python packages often rely on compiled extensions (C/C++, Cython, Fortran) that must be built from source to accurately reflect the performance characteristics of the code at that specific commit. Installing pre-built binaries (wheels) would benchmark the packaged version rather than the code in the pull request. Second, developers often introduce bespoke dependencies or modify build configurations in a pull request, rendering previous environments obsolete. To address this, we implement an agentic pipeline to synthesize a reproducible Docker environment for each task.

**Setup.**    For each task, we first construct a Docker container with the base dependencies installed (Refer to the 'Build Environment' subsection under §A.2) containing the source code of the repository at the initial state of the pull request. Our goal is to synthesize a build script that contains shell commands to install an editable version of the package from source. We also want to ensure that certain tools (ASV, PyTest, and our snapshot testing tool) can be successfully run in the container.

**Agent.**    We employ an iterative, reflexive agent to synthesize a valid build script. The agent is described in Figure 10 and has four principal components:

*Validation & Feedback Loop:* The synthesized script is executed in an isolated Docker container. We validate the build using two verification subroutines. (1) A `profile` check ensures that the package is importable, runnable, and we can run the ASV benchmarks under a generous timeout. (2) A `pytest` check ensures that we can run the pytest test suite without errors. If the build or validation fails, the `stderr` and `stdout` logs are fed back to the agent as observations, allowing it to iteratively refine the script (e.g., installing missing system libraries, fixing syntax errors).

*Chronological Retrieval:* We leverage the insight that build requirements rarely change drastically between adjacent commits. For a given task, we sample 10 successful build scripts from the same repository, sourced from a database of successfully built tasks, sorted by commit date. We first attempt to build the container using the script from the nearest chronological

neighbor. If the build or verification fails, we move to the next neighbor until we either find a successful build or run out of neighbors. The failure logs are preserved and used as observations for the agent.

*Agentic Synthesis:* If the retrieved scripts fail (or no history exists), we instantiate an LLM-based agent to generate a new build script. The agent acts as an interactive planner with access to the failure logs and a set of tools that allows it to inspect the repository state (e.g., list directories, read files, parse `setup.py` or `pyproject.toml`). Given 10 interactive turns, the model can either choose to use one of the tools or prematurely end the turns by synthesizing a build script. The largest model we tried (Claude Sonnet 3.5 and GPT OSS 120b) rarely chooses to use tools as the error messages provide sufficient context while the smallest model (Meta Llama 3.3 8B; (AI@Meta, 2024)) often utilizes many tool interactions before synthesizing the build script.

*LLM Choice and Prompt Design.* We find that a locally hosted `openai/gpt-oss-120b` provides the best balance of performance and cost. We also implement a fallback to `anthropic/claude-3-5-sonnet-20241022` if the build script synthesis fails after multiple tries. Overall, the chronological caching and local LLM cascade allows us to successfully synthesize a build script for 1232 out of 3181 PRs (across 75 repositories) at a cost of \$446.

This process yields 1232 reproducible containers for 3181 PRs. We elected to stop the synthesis prematurely due to limited resources. However, with more resources, we expect the number of reproducible containers to substantially increase.

### A.6. Statistical Testing and Robustness

Finally, we must ensure that every retained task reflects a statistically significant and reproducible performance change. Because timing measurements are inherently noisy (e.g., due to OS scheduling, background load, and CPU power management), we adopt the statistical significance validation procedure used by ASV to verify that the observed differences between two code states are significant under repeated measurement on commodity hardware.

**Measurement protocol.** All experiments are run on an AWS EC2 instance specified in §B.5 to ensures hardware isolation. For each candidate pull request, we execute the expert-selected workloads $\mathtt{Workloads} = \{w_1, \ldots, w_n\}$ on both the baseline codebase $\mathtt{Code}_0$ and the expert-optimized codebase $\mathtt{Code}^*_{\mathrm{expert}}$ on the same instance. For each workload $w_i$, ASV repeatedly evaluates the benchmark under a warm-up and multi-sample timing protocol (with interleaved rounds when enabled), yielding independent sample sets of observed runtimes for the baseline and expert-edited codebases:

$$X_i = \{x_{i1}, \ldots, x_{im}\} \text{ from } w_i(\mathtt{Code}_0), \qquad Y_i = \{y_{i1}, \ldots, y_{ik}\} \text{ from } w_i(\mathtt{Code}^*_{\mathrm{expert}}),$$

where $X_i$ and $Y_i$ denote the set of measurements for workload $w_i$ from the baseline and expert-edited code respectively. We preserve ASV's default sampling parameters (unless a repository overrides it via workload-specific attributes), so that the resulting statistical decision procedure matches common practice in the Python benchmarking ecosystem. The Python benchmarking community emphasizes collecting sufficiently independent measurements over a finite duration rather than enforcing a fixed repetition count. In practice, ASV defaults to collecting between 2 and 40 measurements per workload through adaptive stopping, where additional samples are drawn until the estimate converges or the maximum is reached. These defaults can be adjusted on a per-workload basis to match the characteristics of their specific benchmarks.

**Mann–Whitney U test.** To test whether $\mathtt{Code}_0$ and $\mathtt{Code}^*_{\mathrm{expert}}$ exhibit different performance distributions for a workload, we use the Mann–Whitney U test (Mann & Whitney, 1947), a non-parametric two-sample test based on rank ordering. Formally, for samples $X_i$ and $Y_i$, the $U$ statistic can be written as

$$U(X_i, Y_i) = \sum_{a=1}^{m} \sum_{b=1}^{k} \mathbb{I}[x_{ia} > y_{ib}] \ + \ \tfrac{1}{2}\mathbb{I}[x_{ia} = y_{ib}],$$

and the associated two-sided $p$-value quantifies evidence against the null hypothesis.

**Null hypothesis.** For each workload $w_i$, we test

$$H_0 : \ X_i \text{ and } Y_i \text{ are drawn from the same underlying distribution}$$

(i.e., the patch does not induce a statistically detectable change in the benchmark outcome), against the two-sided alternative that the distributions differ. We only consider workloads that reject $H_0$.

**Implementation.** In practice, ASV applies a conservative two-stage decision rule. When sufficient raw samples are available, it applies the Mann–Whitney $U$ test and declares a difference only if the resulting $p$-value is below a stringent threshold (default $p < 0.002$). If the sample sizes are too small for the $U$ test to ever reach this threshold (given the discrete nature of the test), ASV falls back to a pessimistic check based on uncertainty estimates: it computes a 99% confidence interval for each sample distribution and only declares a difference when these intervals do not overlap. This fallback biases towards not claiming a difference unless the separation is unambiguous.

**Dataset Inclusion Criterion.** We discard candidate tasks for which no workload exhibits a statistically significant change between $Code_0$ and $Code_{expert}$ under this rule. This ensures that every retained task in FORMULACODE corresponds to a clear, reproducible, and statistically supported performance difference. Tasks with no positive significant workloads are also discarded.

This yields the final 957 problems used in FORMULACODE.

**Multiple comparisons correction.** Because each task involves testing significance across $\sim$264.58 workloads, a natural concern is whether the per-workload $p$-values require correction for multiple comparisons. We re-ran the filtering procedure with a Holm–Bonferroni step-down correction (Holm, 1979; Armstrong, 2014) applied within each task. Under this more conservative procedure, 35 of 957 tasks were removed from FORMULACODE, and none of the 108 FORMULACODE-V tasks were affected. The minimal impact is expected as expert-authored performance PRs merged by upstream maintainers almost always produce strong, detectable signals across multiple workloads, so the correction has little practical effect. Our codebase incorporates this correction uses this as an additional filter.

### A.7. FORMULACODE-V Construction

The 108 problems in the FORMULACODE-V subset are drawn from the full FORMULACODE benchmark to serve as a computationally tractable evaluation set. Tasks are selected based on the following criteria: (1) the expert patch must produce a statistically significant speedup that is large enough to distinguish meaningful agent improvements from measurement noise, (2) the task's ASV workloads must execute reliably and with low variance across repeated runs, and (3) the task must have a working PyTest suite for correctness validation.

To support the long-tail generalization experiments (§4.4), we stratify the sampling to preserve the repository distribution of the full benchmark. Specifically, FORMULACODE-V draws from a comparable spread of repository popularity quintiles, ensuring that both popular libraries and niche scientific tools are represented.

Tables 7 and 8 compare the optimization type and difficulty distributions between FORMULACODE and FORMULACODE-V. The distributions are broadly similar, with minor overrepresentation of algorithmic and data structure optimizations in FORMULACODE-V due to the selection of tasks with large expert speedups.

### A.8. Dataset Composition Statistics

To better study the characteristics of FORMULACODE, we develop an automated classifier that attempts to infer the kind of optimization based on a curated taxonomy (§B.3). The classifier is similar to the one introduced in §A.3. It takes as input a sample pull request along with the expert-written patch and attempts to categorize the expert-written solutions using a manually curated taxonomy (Table 11). Such a methodology allows us to efficiently and scalably study the composition of an continuously growing set of problems. The prompts for this classifier are presented in Figure 9 and an example is presented in Figure 11.

The distribution of the types of optimizations is presented in Table 7 and the distribution of the inferred difficulty is presented in Table 8. Importantly, the distribution of optimization problems and difficulty changes marginally between FORMULACODE and FORMULACODE-V.

**Difficulty calibration.** The LLM-inferred difficulty in Table 8 captures the structural complexity of the expert-written patch (e.g., number of files changed, algorithmic sophistication) rather than how difficult the task is for an agent. Beyond the LLM-inferred difficulty above, we note two additional difficulty proxies that emerge from our experimental analyses. First, *repository popularity* (measured by GitHub stars) serves as a proxy for distribution shift: agents are less familiar with the conventions and APIs of niche repositories, and our long-tail analysis (§4.4) confirms that performance degrades substantially on lower-popularity quintiles. Second, the *magnitude of the expert speedup* serves as a proxy for optimization

*Table 7.* Patch classification distribution in FORMULACODE and FORMULACODE-V. The problems in FORMULACODE-V are sampled from the best performing tasks in FORMULACODE which is why some categories are overrepresented.

| Inferred Type of Optimization Problem | % FORMULACODE | % FORMULACODE-V |
|---|---|---|
| Accept Less Precise Solution | 0.6584 | - |
| Cache And Reuse | 8.3128 | 4.6296 |
| Database And Storage Tuning | 0.5761 | - |
| Do It Earlier Batch Throttle | 2.4691 | 0.9259 |
| Io And Latency Hiding | 0.0823 | - |
| Micro Optimizations | 20.2469 | 23.1481 |
| Remove Or Reduce Work | 20.0823 | 18.5185 |
| Uncategorized | 1.5638 | - |
| Use Better Algorithm | 20.0823 | 26.8519 |
| Use Better Data Structure And Layout | 9.7119 | 12.9630 |
| Use Higher Level System | 2.9630 | 2.7778 |
| Use Lower Level System | 11.0288 | 9.2593 |
| Use Parallelization | 2.2222 | 0.9259 |

*Table 8.* The inferred difficulty of human solutions in FORMULACODE and FORMULACODE-V.

| Inferred Difficulty | % FORMULACODE | % FORMULACODE-V |
|---|---|---|
| Easy | 54.8971 | 60.1852 |
| Medium | 44.4444 | 37.0370 |
| Hard | 0.6584 | 2.7778 |

headroom—tasks with small expert speedups leave less room for agent improvement, while tasks with large expert speedups indicate a more impactful optimization opportunity. The Adv metric normalizes against the expert speedup, helping control for this variation in available headroom. These three proxies are complementary: LLM-inferred difficulty captures solution complexity, repository popularity captures data familiarity, and expert speedup captures the magnitude of the optimization opportunity.

## A.9. Limitations

**Domain coverage.**   FORMULACODE is currently restricted to scientific Python repositories that ship with Airspeed Velocity (ASV) benchmarking infrastructure. While our corpus of 70 repositories is broader than prior optimization benchmarks, it does not yet cover web frameworks, distributed systems, embedded software, or non-Python languages. Extending FORMULACODE beyond scientific computing is an important direction for future work; the pipeline design is language-agnostic in principle but requires a corresponding performance benchmarking infrastructure for each target ecosystem.

**Dependence on ASV workloads.**   The quality and coverage of FORMULACODE's evaluation signal is bounded by the workloads maintained by each repository's developers. If community-maintained workloads do not exercise the performance-critical paths modified by a pull request, the benchmark signal may be incomplete. We mitigate this partially through our multi-workload design ($\sim$264.58 workloads per task), which provides broad coverage, but acknowledge that workload completeness is ultimately determined by upstream maintainers.

**Evaluation on a subset.**   Due to compute constraints, our full agent evaluation is conducted on FORMULACODE-V (108 tasks) rather than the complete FORMULACODE benchmark (957 tasks). While FORMULACODE-V preserves the repository distribution and difficulty profile of the full benchmark (Appendix §A.8), results may not generalize to the full task distribution. We plan to expand evaluation coverage as additional compute becomes available.

**Hardware specificity.**   All performance measurements are collected on standardized AWS EC2 c5ad.large instances (Appendix §B.5). While this ensures internal consistency and reproducibility, performance characteristics may differ on other architectures (e.g., ARM, GPU-accelerated workloads). The Adv metric, computed as a difference of speedup ratios on the same hardware, is more robust to hardware variation than absolute timing, but cross-platform generalization remains an open question.

```
1    INPUT SIGNATURE
2
3    problem_description : string
4    Problem statement and technical context from PR/issue.
5
6    git_patch : string
7    Git diff showing actual code changes.
8
9    file_change_summary : string
10   A markdown table summarizing all the files changed in the commit along with lines added/removed.
11
12   CLASSIFIER MODULE
13
14   Decide if this commit's PRIMARY intent is to improve product/runtime performance.
15   Label YES only when there is CLEAR, EXPLICIT evidence in the description and/or patch that the runtime gets faster (e.g.,
     algorithm change, fewer allocations, caching, vectorization, reduced I/O, async/non-blocking for throughput, latency
     reduction, memory footprint reduction, fix a speed regression).
16
17   Strong positive signals (weigh these collectively):
18   · PR title/body contains performance intent (e.g., "PERF:", "speed up", "faster", "performance").
19   · Linked issues/comments include benchmark links or timings demonstrating impact.
20   · Low-level/hot-path tweaks (e.g., reuse global context, avoid per-call init/teardown, vectorize C/NumPy).
21
22   Hard NO (non-performance) examples:
23   tests/ASV/harness-only changes; CI/workflows/build/packaging; coverage; pre-commit/format/lints (clippy/ruff/black);
     docs; version bumps; terminology/renames; pure refactors without performance claims; changes aimed at making perf tests
     pass but not improving runtime.
24
25   If ambiguous, weigh the concrete code changes and problem description together.
26   When there are specific performance cues (title keywords, measured timings, fewer allocations, vectorization,
     caching/reuse) lean YES; otherwise NO.
27
28   OUTPUT SIGNATURE
29
30   reasoning : string
31   Deductive reasoning steps leading to the classification.
32
33   label : string
34   Final label: "YES" for performance-related, "NO" otherwise.
35
```

*Figure 7.* Prompt template used by the LLM-based performance intent classifier described in A.3. The prompt defines the input signature (problem description, git patch, and file change summary), the classifier module specifying decision criteria for identifying performance-motivated commits, and the output signature producing a reasoning trace and binary label ("YES"/"NO").

```
1    Example PR
2
3    CLASSIFIER INPUT
4
5    problem_description : string
6    Labels: performance; Description: Fixes #14471.
7    Body: The new ParameterExpression.bind_all is a fast path for producing a numeric result. This has advantages over
     ParameterExpression.bind:
8    · Far fewer Python objects are allocated, since no new ParameterExpression objects need to be constructed and the output
     is guaranteed to be numeric.
9    · There is no historical API requirement to scan the incoming mapping for invalid keys or values, yielding a large
     performance improvement when the same mapping is used to bind many expressions.
10   · This provides a major complexity improvement when a large values dictionary is reused many times.
11   There is still room for further gains because the Rust-space ParameterExpression and SymbolExpr interfaces require more
     heap allocations than strictly necessary, but this already yields substantial speedups.
12   Issues: Fixes #14471.
13   The linked issue reports that ParameterExpression.bind scales with the size of the binding dictionary even when only a
     single parameter is needed, leading to severe performance penalties for large parameter tables.
14   Comments:
15   Currently in draft because there's no tests - I'm just putting it up so Sam and Ian from #14471 can test it out for their
     use case. For the explicit example in that issue, a complete comparison on my machine:
16   <details><summary>Out of date timings</summary>
17   In [1]: from qiskit.circuit import Parameter, ParameterExpression
18       N: int = 100_000
19       parameter_values = {Parameter(f"th_{i}"): 1 for i in range(N)}
20       parameter_values[param := Parameter("my_param")] = 1
21    ... <TRUNCATED>
22   I think it's fine without having the same behavior. For clarity it might be helpful to add a blurb to the bind_all
     docstring to say that "unlike bind, NaN and inf are in the range of expected outputs for this method".
23   LGTM, thanks!
24
25   git_patch : string
26   diff –git a/crates/circuit/src/parameter/parameter_expression.rs b/crates/circuit/src/parameter/parameter_expression.rs
27   index 1f0406f62c7e..98da2ee3e9e6 100644
28   – a/crates/circuit/src/parameter/parameter_expression.rs
29   +++ b/crates/circuit/src/parameter/parameter_expression.rs
30   @@ -1048,6 +1048,40 @@ impl PyParameterExpression {}
31   + ##[pyo3(name = "bind_all")]
32   + ##[pyo3(signature = (values, *))]
33   + pub fn py_bind_all(&self, values: Bound<PyAny>) -> PyResult<Value> {}
34   + let mut partial_map = HashMap::with_capacity(self.inner.name_map.len());
35   + for symbol in self.inner.name_map.values() {}
36   + let py_parameter = symbol.clone().into_pyobject(values.py())?;
37   + partial_map.insert(symbol, values.get_item(py_parameter)?.extract()?);
38   +
39   + let bound = self.inner.expr.bind(&partial_map);
40   ... <Truncated>
41
42   file_change_summary : string
```

| File | Lines Added | Lines Removed | Total Changes |
|---|---|---|---|
| crates/circuit/src/parameter/parameter_expression.rs | 43 | 0 | 43 |
| crates/circuit/src/parameter/symbol_expr.rs | 1 | 1 | 2 |
| releasenotes/notes/parameter-bind-all-22337586429509f9.yaml | 8 | 0 | 8 |
| test/python/circuit/test_parameters.py | 18 | 0 | 18 |

```
44
45   CLASSIFIER MODULE
46
47   Decide if this commit's PRIMARY intent is to improve product/runtime performance.
48   Label YES only when. . .
49   . . . there are specific performance cues (title keywords, measured timings, fewer allocations, vectorization,
     caching/reuse) lean YES; otherwise NO.
50
51   CLASSIFIER OUTPUT
52
53   reasoning : string
54   The pull request introduces a new fast-path method ParameterExpression.bind_all (and Parameter.bind_all) to bind
     parameters to numeric values with far fewer allocations and reduced overhead than the existing bind method. The PR
     explicitly emphasizes performance, using terms such as "fast path" and "huge performance gain", and reports timing
     improvements from tens of milliseconds to microsecond-scale execution.
55   The implementation avoids constructing new ParameterExpression objects and skips expensive validation logic, directly
     producing numeric results. It is labeled with the performance tag and documented as a fast-path method for
     performance-sensitive use cases, making the primary intent of the change to reduce computational overhead and improve
     runtime performance of parameter binding.
56   Therefore, the commit's primary purpose is to make parameter binding substantially faster.
57
58   label : string
59   'YES'
60
```

Figure 8. Example of performance intent classification for a real pull request (https://github.com/Qiskit/qiskit/pull/14782), illustrating how the prompt in Fig. 7 is instantiated and how the classifier produces a reasoning trace and binary label.

```
1    Performance type & difficulty classifier
2
3    INPUT SIGNATURE PROMPT
4
5    problem_description : string
6    Problem statement and technical context from PR/issue.
7
8    git_patch : string
9    Git diff showing actual code changes.
10
11   CLASSIFIER MODULE PROMPT
12
13   Decide the PRIMARY performance optimization technique and the difficulty level of the optimization.
14
15   Category mapping (when performance-related):
16   · Algorithm improvements: complexity reduction; switching to faster algorithms → use_better_algorithm
17   · Data structures / layout: sets, maps, indices; memory layout tuning → use_better_data_structure_and_layout
18   · System-level: C/Rust/NumPy/Vectorized/Native extensions → use_lower_level_system
19   · Approximation / heuristics: trade accuracy for speed → accept_less_precise_solution
20   · Parallelization: threads, processes, parallel algorithms (not just async I/O) → use_parallelization
21   · Cache & reuse: memoization, LRU, materialized results → cache_and_reuse
22   · Scheduling: batching, lazy execution, throttling → do_it_earlier_batch_throttle
23   · Database / storage: indices, query tuning, partitioning → database_and_storage_tuning
24   · Micro-optimizations: hot-path tweaks, guards, inlining → micro_optimizations
25   · I/O / latency hiding: async or non-blocking I/O, overlap I/O and compute → io_and_latency_hiding
26   · Higher-level systems: using optimized libraries or frameworks → use_higher_level_system
27   · Uncategorized: performance-related but does not fit the above categories → uncategorized
28
29   Difficulty (when performance-related):
30   · easy: localized change (< 50 lines), minimal risk
31   · medium: module-level refactor, data structure changes
32   · hard: algorithm rewrite or architectural change
33
34   OUTPUT SIGNATURE PROMPT
35
36   category : OptimizationType
37   The classified optimization category.
38
39   difficulty : DifficultyLevel
40   The difficulty level of the optimization.
41
42   reasoning : string
43   Brief explanation of the classification.
44
```

*Figure 9.* Prompt template used by the LLM-based classifier for assigning each performance task an optimization category and difficulty level ( B.3). The prompt defines the input signature, a taxonomy-driven classification module that maps code changes to optimization types, and an output schema that produces the predicted category, difficulty, and a brief reasoning trace.

```
1    INPUT SIGNATURE
2
3    owner_repo : string
4    The repository this commit belongs to (e.g., scikit-learn/scikit-learn).
5
6    sha : string
7    The commit SHA that is currently checked out.
8
9    commit_date : string
10   The commit date in ISO format (e.g., 2023-10-05T12:34:56Z).
11
12   stderr_logs : string
13   Most recent stderr logs from the last build attempt (up to ~8k tail-end characters).
14
15   stdout_logs : string
16   Most recent stdout logs from the last build attempt (up to ~8k tail-end characters).
17
18   failure_more : string
19   Describes where the failure occurred (e.g., N/A, build failed, asv run failed).
20
21   last_docker_build_script : string
22   The previously generated docker_build.sh script.
23
24   repo_facts_json : string
25   JSON object containing inferred repository facts (paths, package names, versions, etc.).
26
27   toolbelt : string
28   Human-readable summary of available tools and their usage.
29
30   messages_log : string
31   Transcript of prior tool calls, actions, and observations.
32
33   BUILD AGENT MODULE
34
35   An interactive planner for producing a docker_build.sh bash script that builds and installs a Python repository inside
     micromamba environments. The agent may either: (A) Request a tool call with structured JSON arguments, or (B) Output the
     final executable build script.
36   If a tool is required, set next_action to one of: probe_repo | list_tree | read_file | try_import | none.
37
38   Tool call formats:
39   · read_file: {"path": "...", "max_bytes": 65536}
40   · list_tree: {"depth": 2}
41   · try_import: {"candidates": ["foo","bar"]}
42   Return docker_build_script only when fully satisfied with correctness and completeness.
43   Critical constraints on the generated script:
44   · Must be idempotent and safe to run inside Docker.
45   · Fully non-interactive; no user prompts.
46   · Must be valid executable Bash with no syntax errors.
47   · Must use real newline characters (not escaped \n).
48   · Must not output literal \n.
49   Post-install readiness requirements:
50   · After editable install, the environment must be immediately usable.
51   · A lightweight profiling sanity check and a lightweight pytest sanity check must start without immediate errors, even
     for projects that require execution from subdirectories.
52   · Test/benchmark extras and optional dependencies must be installed as needed for import and test discovery to succeed.
53
54   OUTPUT SIGNATURE
55
56   thought : string
57   Brief rationale describing the current decision or plan.
58
59   next_action : string
60   One of probe_repo, list_tree, read_file, try_import, none, or finish.
61
62   action_input : string
63   JSON arguments for the selected tool, or empty if no tool is called.
64
65   error_summary : string
66   Brief summary of the most recent build failure and its possible causes.
67
68   resolution_steps : string
69   Concrete steps required to resolve the failure.
70
71   docker_build_script : string
72   Final executable docker_build.sh script that successfully builds and installs the project from source.
```

*Figure 10.* Prompt structure for the docker build agent ( A.5), defining its input state, tool-calling interface, constraints, and executable script output.

```
 1    Example PR
 2
 3    CLASSIFIER INPUT
 4
 5    problem_description : string
 6    Labels: performance; Description: Fixes #14471.
 7    Body: The new ParameterExpression.bind_all is a fast path for producing a numeric result. This has advantages over
      ParameterExpression.bind:
 8    · Far fewer Python objects are allocated, since no new ParameterExpression objects need to be constructed and the output
      is guaranteed to be numeric.
 9    · There is no historical API requirement to scan the incoming mapping for invalid keys or values, yielding a large
      performance improvement when the same mapping is used to bind many expressions.
10    · This provides a major complexity improvement when a large values dictionary is reused many times.
11    There is still room for further gains because the Rust-space ParameterExpression and SymbolExpr interfaces require more
      heap allocations than strictly necessary, but this already yields substantial speedups.
12    Issues: Fixes #14471.
13    The linked issue reports that ParameterExpression.bind scales with the size of the binding dictionary even when only a
      single parameter is needed, leading to severe performance penalties for large parameter tables.
14    Comments:
15    Currently in draft because there's no tests - I'm just putting it up so Sam and Ian from #14471 can test it out for their
      use case. For the explicit example in that issue, a complete comparison on my machine:
16    <details><summary>Out of date timings</summary>
17    In [1]: from qiskit.circuit import Parameter, ParameterExpression
18        N: int = 100_000
19        parameter_values = {Parameter(f"th_{i}"): 1 for i in range(N)}
20        parameter_values[param := Parameter("my_param")] = 1
21        print("Using the specialised 'Parameter' methods:")
22        %timeit param.bind(parameter_values, allow_unknown_parameters=True)
23    </details> . . . <TRUNCATED>
24    I think it's fine without having the same behavior. For clarity it might be helpful to add a blurb to the bind_all
      docstring to say that "unlike bind, NaN and inf are in the range of expected outputs for this method".
25    LGTM, thanks!
26
27    git_patch : string
28    diff –git a/crates/circuit/src/parameter/parameter_expression.rs b/crates/circuit/src/parameter/parameter_expression.rs
29    index 1f0406f62c7e..98da2ee3e9e6 100644
30    – a/crates/circuit/src/parameter/parameter_expression.rs
31    +++ b/crates/circuit/src/parameter/parameter_expression.rs
32    @@ -1048,6 +1048,40 @@ impl PyParameterExpression {}
33    + ##[pyo3(name = "bind_all")]
34    + ##[pyo3(signature = (values, *))]
35    + pub fn py_bind_all(&self, values: Bound<PyAny>) -> PyResult<Value> {}
36    + let mut partial_map = HashMap::with_capacity(self.inner.name_map.len());
37    + for symbol in self.inner.name_map.values() {}
38    + let py_parameter = symbol.clone().into_pyobject(values.py())?;
39    + partial_map.insert(symbol, values.get_item(py_parameter)?.extract()?);
40    +
41    + let bound = self.inner.expr.bind(&partial_map);
42    + bound.eval(true).ok_or_else(|| {}
43    + PyTypeError::new_err(format!("binding did not produce a numeric quantity: {bound:?}"))
44    + )
45    +
46    ... <Truncated>
47
48    CLASSIFIER MODULE
49
50    Decide the PRIMARY performance optimization technique and the difficulty level of the optimization.
51    Category mapping (when performance-related): . . .
52    . . . algorithm rewrite or architectural change.
53
54    CLASSIFIER OUTPUT
55
56    category : OptimizationType
57    REMOVE_OR_REDUCE_WORK
58
59    difficulty : DifficultyLevel
60    'medium'
61
62    reasoning : string
63    The change adds a specialized 'bind_all' fast-path that skips the heavy allocation and validation logic of the generic
      'bind' method, directly mapping parameters to numeric values and evaluating the expression. This reduces the amount of
      work performed per binding call, yielding large speedups. The implementation touches Rust code and adds tests but is a
      localized, moderate-size change, fitting a medium difficulty level.
64
```

*Figure 11.* Example application of the optimization type and difficulty classifier (Figure 9), illustrating the predicted category, difficulty level, and reasoning for a real pull request (https://github.com/Qiskit/qiskit/pull/14782)

```
1    Judge performance related PR prompt
2
3    INPUT SIGNATURE PROMPT
4
5    problem_description : string
6    Problem statement and technical context from PR/issue.
7
8    git_patch : string
9    Git diff showing actual code changes.
10
11   file_change_summary : string
12   A markdown table summarizing all the files changed in the commit along with lines added/removed.
13
14   JUDGE SIGNATURE PROMPT
15
16   Decide if this commit's PRIMARY intent is to improve product/runtime performance.
17
18   Label YES only when there is CLEAR, EXPLICIT evidence in the description and/or patch that the runtime gets faster (e.g.,
     algorithm change, fewer allocations, caching, vectorization, reduced I/O, async/non-blocking for throughput, latency
     reduction, memory footprint reduction, fix a speed regression).
19
20   Strong positive signals (weigh these collectively):
21   - PR title/body contains performance intent (e.g., "PERF:", "speed up", "faster", "performance").
22   - Linked issues/comments include benchmark links or timings demonstrating impact.
23   - Low-level/hot-path tweaks (e.g., reuse global context, avoid per-call init/teardown, vectorize C/NumPy).
24
25   Hard NO (non-performance) examples: tests/ASV/harness-only changes; CI/workflows/build/packaging; coverage;
     pre-commit/format/lints (clippy/ruff/black); docs; version bumps; terminology/renames; pure refactors without
     performance claims; changes aimed at making perf tests pass but not improving runtime.
26
27   If ambiguous, weigh the concrete code changes and problem description together. When there are specific performance cues
     (title keywords, measured timings, fewer allocations, vectorization, caching/reuse) lean YES; otherwise NO.
28
29   OUTPUT SIGNATURE PROMPT
30
31   reasoning : string
32   Deductive reasoning steps leading to the classification.
33
34   label : string
35   Final label: "YES" for performance-related, "NO" otherwise.'
36
```

*Figure 12.* Structured DSPy prompt used to judge whether a pull request is primarily intended to improve runtime or product performance. The prompt specifies the required inputs (problem description, code diff, and file-level change summary), explicit decision criteria and exclusions for performance-related changes, and an output format consisting of a justification and a binary YES/NO label. The design emphasizes conservative, evidence-based classification, prioritizing explicit runtime improvements over incidental or refactoring-only changes.

```
1    Problem Extractor Prompt description
2
3    INPUT SIGNATURE PROMPT
4
5    pr_title : string
6    The GitHub PR title
7
8    pr_body : string
9    The GitHub PR description
10
11   pr_comments : string
12   Comments on the PR thread.
13
14   PROBLEM EXTRACTOR SIGNATURE
15
16   What problem is this Github PR trying to solve? Extract near-verbatim relevant text following the given JSON output. If
      no relevant context exists for a field, return an empty string for it.
17
18   OUTPUT SIGNATURE PROMPT
19
20   initial_observations: string | list[Any] | None
21   Objective symptoms of the problematic behavior, described in the present tense. Focus strictly on what is happening
      (metrics, user impact, frequency). Do not include causes, hypotheses, or explanations.
22
23   triage_attempts: string | list[Any] | None
24   The investigative steps and reasoning used to narrow down contributing factors—what you checked, what you ruled out, and
      what evidence you gathered to understand where the issue originates.
25
26   solution_overview: string | list[Any] | None
27   A concise description of the change(s) made and how they address the identified bottleneck or constraint.
28
29   solution_observations: string | list[Any] | None
30   What you observe after applying the change—new measurements, behavior differences, and any regressions or trade-offs that
      appeared.
31
```

*Figure 13.* Structured DSPy prompt used to extract the underlying problem and resolution context from a GitHub pull request. The prompt consumes the PR title, description, and discussion, and produces a structured summary capturing observed symptoms, triage steps, the implemented solution, and post-change observations. The design emphasizes near-verbatim extraction and separation of observations, investigation, and outcomes.

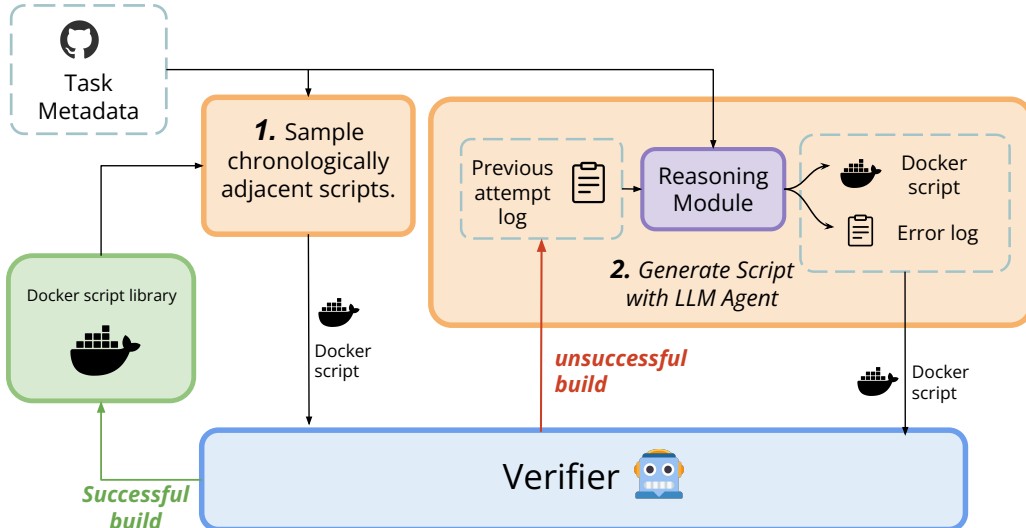

*Figure 14.* Overview of the pipeline for Docker environment synthesis. The system reuses chronologically adjacent build scripts when possible, otherwise invoking an LLM agent that generates and refines Docker scripts using build logs and repository context until a verifier confirms a successful, reproducible build.

*Table 9.* Repositories and Tasks after applying rule-based filters (Filter Stage 1) and LLM-based filters (Filter Stage 2) as described in §A.2. We also showcase the number of tasks, the date of creation of the latest task, and additional information about the functionality and popularity of the repository. Most repositories are software tools used extensively within scientific communities.

| Repository Name | #Stars | #Forks | Filter Stage 1 | Filter Stage 2 | Latest Task Date | Description |
|---|---|---|---|---|---|---|
| **1**. scikit-learn/scikit-learn | 63792 | 26359 | 2434 | 243 | 2025-10-31 | scikit-learn: machine learning in Python |
| **2**. pandas-dev/pandas | 46922 | 19184 | 3298 | 560 | 2025-11-11 | Flexible and powerful data analysis / manipulation library for Python, providing labeled data structures similar to R data.frame objects, statistical functions, and much more |
| **3**. scipy/scipy | 14120 | 5516 | 1454 | 209 | 2025-10-29 | SciPy library main repository |
| **4**. apache/arrow | 16089 | 3884 | 1988 | 267 | 2025-07-22 | Apache Arrow is the universal columnar format and multi-language toolbox for fast data interchange and in-memory analytics |
| **5**. networkx/networkx | 16277 | 3415 | 288 | 44 | 2025-09-16 | NetworkX is a Python package for the creation, manipulation, and study of the structure, dynamics, and functions of complex networks. |
| **6**. Qiskit/qiskit | 6598 | 2659 | 717 | 212 | 2025-11-19 | Qiskit is an open-source SDK for working with quantum computers at the level of pulses, circuits, and application modules. |
| **7**. scikit-image/scikit-image | 6371 | 2320 | 458 | 54 | 2025-11-18 | Image processing in Python |

| Repository Name | #Stars | #Forks | Filter Stage 1 | Filter Stage 2 | Latest Task Date | Description |
|---|---|---|---|---|---|---|
| **8**. pymc-devs/pymc | 9322 | 2146 | 685 | 45 | 2025-09-23 | PyMC (formerly PyMC3) is a Python package for Bayesian statistical modeling focusing on advanced Markov chain Monte Carlo (MCMC) and variational inference (VI) algorithms. |
| **9**. Textualize/rich | 54172 | 1920 | 165 | 11 | 2025-07-25 | Rich is a Python library for rich text and beautiful formatting in the terminal. |
| **10**. tqdm/tqdm | 30580 | 1402 | 12 | 1 | 2022-03-24 | Fast, extensible progress bar for Python and CLI |
| **11**. pydata/xarray | 4004 | 1192 | 609 | 101 | 2025-11-21 | N-D labeled arrays and datasets in Python |
| **12**. optuna/optuna | 12922 | 1177 | 719 | 112 | 2025-11-05 | A hyperparameter optimization framework |
| **13**. quantumlib/Cirq | 4772 | 1151 | 10 | 3 | 2025-11-18 | Python framework for creating, editing, and invoking Noisy Intermediate-Scale Quantum (NISQ) circuits. |
| **14**. pvlib/pvlib-python | 1424 | 1126 | 110 | 8 | 2025-10-03 | A set of documented functions for simulating the performance of photovoltaic energy systems. |
| **15**. ipython/ipyparallel | 2626 | 1006 | 65 | 6 | 2024-10-28 | IPython Parallel: Interactive Parallel Computing in Python |
| **16**. geopandas/geopandas | 4940 | 981 | 314 | 22 | 2025-05-22 | Python tools for geographic data |
| **17**. kedro-org/kedro | 10593 | 971 | 41 | 4 | 2025-07-17 | Kedro is a toolbox for production-ready data science. It uses software engineering best practices to help you create data engineering and data science pipelines that are reproducible, maintainable, and modular. |
| **18**. HIPS/autograd | 7379 | 928 | 13 | 1 | 2017-10-21 | Efficiently computes derivatives of NumPy code. |
| **19**. MDAnalysis/mdanalysis | 1477 | 733 | 196 | 23 | 2025-10-13 | MDAnalysis is a Python library to analyze molecular dynamics simulations. |
| **20**. pybamm-team/PyBaMM | 1387 | 692 | 218 | 17 | 2025-04-29 | PyBaMM (Python Battery Mathematical Modelling) is an open-source battery simulation package written in Python. |
| **21**. modin-project/modin | 10332 | 669 | 50 | 8 | 2025-09-30 | Speed up your Pandas workflows by changing a single line of code |
| **22**. nilearn/nilearn | 1322 | 631 | 138 | 2 | 2025-10-09 | Machine learning for NeuroImaging in Python |
| **23**. sunpy/sunpy | 971 | 626 | 663 | 22 | 2025-05-16 | sunpy is a Python software package that provides fundamental tools for accessing, loading and interacting with solar physics data in Python. |
| **24**. shapely/shapely | 4284 | 600 | 150 | 21 | 2025-05-03 | Manipulation and analysis of geometric objects |
| **25**. dedupeio/dedupe | 4387 | 568 | 25 | 4 | 2023-12-19 | A python library for accurate and scalable data deduplication and entity-resolution. |

*Continued on next page*

| Repository Name | #Stars | #Forks | Filter Stage 1 | Filter Stage 2 | Latest Task Date | Description |
|---|---|---|---|---|---|---|
| **26**. h5py/h5py | 2174 | 547 | 263 | 35 | 2025-08-10 | h5py is a thin, pythonic wrapper around HDF5 |
| **27**. PyWavelets/pywt | 2294 | 517 | 12 | 1 | 2024-07-16 | PyWavelets - Wavelet Transforms in Python |
| **28**. pydicom/pydicom | 2070 | 508 | 86 | 7 | 2025-05-12 | Read, modify and write DICOM files with python code |
| **29**. arviz-devs/arviz | 1737 | 458 | 107 | 5 | 2025-10-21 | Exploratory analysis of Bayesian models |
| **30**. napari/napari | 2512 | 454 | 849 | 69 | 2025-09-30 | napari: a fast, interactive, multi-dimensional image viewer for python |
| **31**. tardis-sn/tardis | 225 | 446 | 268 | 13 | 2025-09-16 | TARDIS - Temperature And Radiative Diffusion In Supernovae |
| **32**. dipy/dipy | 787 | 446 | 194 | 16 | 2025-11-18 | DIPY is the paragon 3D/4D+ medical imaging library in Python. Contains generic methods for spatial normalization, signal processing, machine learning, statistical analysis and visualization of medical images. Additionally, it contains specialized methods for computational anatomy including diffusion, perfusion and structural imaging. |
| **33**. python-control/python-control | 1908 | 444 | 117 | 6 | 2025-06-21 | The Python Control Systems Library is a Python module that implements basic operations for analysis and design of feedback control systems. |
| **34**. SciTools/cartopy | 1545 | 389 | 74 | 6 | 2025-04-26 | Cartopy is a Python package designed for geospatial data processing in order to produce maps and other geospatial data analyses. |
| **35**. holoviz/datashader | 3467 | 377 | 90 | 19 | 2025-10-09 | Quickly and accurately render even the largest data. |
| **36**. microsoft/Qcodes | 396 | 335 | 187 | 10 | 2025-09-05 | Modular data acquisition framework |
| **37**. mars-project/mars | 2748 | 326 | 164 | 51 | 2023-02-16 | Mars is a tensor-based unified framework for large-scale data computation which scales numpy, pandas, scikit-learn and Python functions. |
| **38**. pytroll/satpy | 1146 | 320 | 520 | 45 | 2025-08-02 | Python package for reading, manipulating and writing satellite data |
| **39**. SciTools/iris | 692 | 297 | 109 | 23 | 2025-10-31 | A powerful, format-agnostic, and community-driven Python package for analysing and visualising Earth science data |
| **40**. lmfit/lmfit-py | 1164 | 290 | 205 | 8 | 2022-09-05 | Non-Linear Least Squares Minimization, with flexible Parameter settings, based on scipy.optimize, and with many additional classes and methods for curve fitting. |

| Repository Name | #Stars | #Forks | Filter Stage 1 | Filter Stage 2 | Latest Task Date | Description |
|---|---|---|---|---|---|---|
| 41. deepchecks/deepchecks | 3924 | 286 | 99 | 9 | 2023-12-06 | Deepchecks: Tests for Continuous Validation of ML Models & Data. Deepchecks is a holistic open-source solution for all of your AI & ML validation needs, enabling to thoroughly test your data and models from research to production. |
| 42. devitocodes/devito | 632 | 242 | 99 | 7 | 2025-07-24 | DSL and compiler framework for automated finite-differences and stencil computation |
| 43. danielgtaylor/python-betterproto | 1733 | 233 | 42 | 1 | 2023-12-07 | Better Protobuf / gRPC code generator and library for Python |
| 44. scikit-learn-contrib/metric-learn | 1425 | 229 | 6 | 1 | 2017-11-27 | Metric Learning in Python |
| 45. pydicom/pynetdicom | 551 | 188 | 24 | 1 | 2025-05-24 | A Python implementation of the DICOM networking protocol |
| 46. scverse/anndata | 667 | 175 | 142 | 17 | 2025-07-23 | Annotated data matrix for single-cell genomics |
| 47. apache/arrow-adbc | 498 | 160 | 571 | 63 | 2025-11-07 | Database connectivity API standard and libraries for Apache Arrow |
| 48. man-group/ArcticDB | 2102 | 153 | 11 | 2 | 2025-11-19 | ArcticDB is a high performance data store for time series and tick data |
| 49. stac-utils/pystac | 412 | 127 | 48 | 1 | 2023-03-31 | Python library for working with SpatioTemporal Asset Catalog (STAC) |
| 50. xdslproject/xdsl | 433 | 125 | 2136 | 236 | 2025-11-04 | A Python compiler design toolkit. |
| 51. ActivitySim/activitysim | 217 | 117 | 51 | 10 | 2025-11-12 | An open platform for activity-based travel behavior modeling |
| 52. OGGM/oggm | 245 | 115 | 484 | 36 | 2025-04-01 | Open Global Glacier Model (OGGM): a modular framework for glacier modeling |
| 53. datalad/datalad | 613 | 115 | 426 | 31 | 2024-09-10 | Keep code, data, containers under control with git and git-annex |
| 54. pydata/bottleneck | 1144 | 112 | 61 | 20 | 2025-04-29 | Fast NumPy array functions written in C |
| 55. wmayner/pyphi | 406 | 100 | 25 | 1 | 2024-09-24 | A toolbox for integrated information theory. |
| 56. django-components/ django-components | 1463 | 100 | 53 | 3 | 2025-09-30 | Reusable, composable components for Django templates |
| 57. sourmash-bio/sourmash | 524 | 88 | 297 | 27 | 2025-01-09 | Quickly search, compare, and analyze genomic and metagenomic data sets. |
| 58. tskit-dev/msprime | 201 | 88 | 209 | 9 | 2025-07-24 | Simulate genealogical trees and genomic sequence data using population genetic models |
| 59. numpy/numpy-financial | 384 | 87 | 13 | 4 | 2024-04-04 | Financial functions for NumPy |
| 60. makepath/xarray-spatial | 894 | 85 | 38 | 9 | 2023-02-16 | Spatial analysis algorithms for xarray implemented in numba |
| 61. dwavesystems/dimod | 135 | 84 | 152 | 20 | 2024-06-13 | dimod is a shared API for samplers. |

| Repository Name | #Stars | #Forks | Filter Stage 1 | Filter Stage 2 | Latest Task Date | Description |
|---|---|---|---|---|---|---|
| 62. python-hyper/h11 | 530 | 83 | 18 | 2 | 2025-01-12 | A pure-Python, bring-your-own-I/O implementation of HTTP/1.1 |
| 63. bjodah/chempy | 611 | 81 | 69 | 1 | 2018-03-24 | A package useful for chemistry written in Python |
| 64. holoviz/param | 497 | 79 | 85 | 10 | 2025-02-27 | Declarative parameters for robust Python classes and a rich API for reactive programming |
| 65. inducer/loopy | 615 | 78 | 172 | 15 | 2023-07-27 | A code generator for array computations on CPUs and GPUs |
| 66. holgern/beem | 138 | 75 | 75 | 5 | 2020-12-22 | A Python library for Hive and Steem |
| 67. scverse/spatialdata | 329 | 75 | 20 | 2 | 2025-09-29 | An open and interoperable data framework for spatial omics data |
| 68. pysb/pysb | 188 | 71 | 107 | 7 | 2021-01-20 | PySB is a framework for building mathematical models of biochemical systems as Python programs |
| 69. xorbitsai/xorbits | 1199 | 70 | 186 | 22 | 2024-11-16 | Xorbits is an open-source computing framework that makes it easy to scale data science and machine learning workloads — from data preprocessing to tuning, training, and model serving. |
| 70. pysal/momepy | 563 | 67 | 80 | 12 | 2024-07-16 | Urban Morphology Measuring Toolkit |
| 71. python-adaptive/adaptive | 1203 | 62 | 28 | 5 | 2025-08-21 | :chart_with_upwards_trend: Adaptive: parallel active learning of mathematical functions |
| 72. probabilistic-numerics/ probnum | 459 | 61 | 52 | 7 | 2023-05-04 | Probabilistic numerics in Python |
| 73. neurostuff/NiMARE | 197 | 60 | 14 | 1 | 2025-06-13 | Coordinate- and image-based meta-analysis in Python |
| 74. NCAR/geocat-comp | 140 | 56 | 18 | 2 | 2025-08-18 | GeoCAT-comp provides implementations of computational functions for operating on geosciences data. Many of these functions originated in NCL and were translated into Python. |
| 75. mie-lab/trackintel | 243 | 53 | 55 | 5 | 2024-01-07 | trackintel is a library for the analysis of spatio-temporal tracking data with a focus on human mobility. |
| 76. JDASoftwareGroup/ kartothek | 160 | 53 | 152 | 31 | 2021-03-17 | A dataset library for partitioned datasets stored in Parquet |
| 77. AllenCellModeling/ aicsimageio | 220 | 51 | 50 | 3 | 2023-04-05 | Image Reading, Metadata Conversion, and Image Writing for Microscopy Images in Python |
| 78. dottxt-ai/outlines-core | 254 | 50 | 44 | 5 | 2025-03-31 | Core library for Outlines, providing structured text generation utilities |
| 79. apache/arrow-nanoarrow | 207 | 47 | 109 | 8 | 2025-10-27 | nanoarrow: a (C) library for the Apache Arrow C Data interface |
| 80. pangeo-data/climpred | 252 | 47 | 9 | 2 | 2021-11-20 | :earth_americas: Verification of weather and climate forecasts :earth_africa: |
| 81. pybop-team/PyBOP | 152 | 45 | 78 | 8 | 2025-07-15 | A parameterisation and optimisation package for battery models. |

*Continued on next page*

| Repository Name | #Stars | #Forks | Filter Stage 1 | Filter Stage 2 | Latest Task Date | Description |
|---|---|---|---|---|---|---|
| 82. UXARRAY/uxarray | 202 | 44 | 99 | 22 | 2025-09-11 | Python library for working with unstructured grid model data in xarray |
| 83. pygeos/pygeos | 388 | 43 | 101 | 17 | 2021-11-30 | Wraps GEOS geometry functions in numpy ufuncs |
| 84. innobi/pantab | 120 | 41 | 79 | 7 | 2024-10-31 | Read/Write pandas DataFrames with Tableau Hyper Extracts |
| 85. xarray-contrib/xskillscore | 237 | 41 | 23 | 1 | 2021-11-20 | Metrics for verifying forecasts |
| 86. glotzerlab/signac | 135 | 37 | 17 | 2 | 2025-04-04 | Manage large and heterogeneous data spaces on the file system. |
| 87. sgkit-dev/sgkit | 265 | 37 | 113 | 21 | 2025-09-30 | Scalable genetics toolkit |
| 88. TileDB-Inc/TileDB-Py | 198 | 36 | 51 | 5 | 2025-08-01 | Python API for TileDB |
| 89. IntelPython/dpctl | 117 | 31 | 37 | 2 | 2025-10-02 | Data Parallel Control (dpctl) - Python device control and USM memory for SYCL |
| 90. tensorwerk/hangar-py | 205 | 29 | 19 | 1 | 2019-12-04 | Hangar is version control for tensor data. Commit, branch, merge, revert, and collaborate in the data-defined software era. |
| 91. xarray-contrib/xbatcher | 184 | 28 | 20 | 3 | 2023-07-31 | Batch generation from xarray objects. |
| 92. DASDAE/dascore | 121 | 26 | 122 | 11 | 2025-09-20 | DASCore: A Python package for the analysis of distributed acoustic sensing data. |
| 93. IntelPython/dpnp | 116 | 23 | 680 | 26 | 2025-10-14 | Data Parallel Extension for NumPy |
| 94. not522/ac-library-python | 230 | 23 | 5 | 2 | 2021-11-19 | Python implementation of AtCoder Library |
| 95. xarray-contrib/flox | 133 | 21 | 150 | 39 | 2025-07-17 | Fast groupby reductions for dask and xarray |
| 96. scipp/scipp | 136 | 21 | 268 | 26 | 2025-03-17 | Python library for multi-dimensional data analysis |
| 97. pyapp-kit/psygnal | 115 | 21 | 70 | 10 | 2025-09-24 | Python observer pattern (callback/event system). Modeled after Qt Signals & Slots (but independent of Qt) |
| 98. royerlab/ultrack | 149 | 21 | 68 | 5 | 2025-09-23 | Cell tracking and segmentation software |
| 99. xitorch/xitorch | 155 | 21 | 9 | 2 | 2024-05-24 | Differentiable scientific computing for PyTorch |
| 100. Quansight-Labs/ndindex | 107 | 16 | 12 | 3 | 2025-05-14 | A Python library for manipulating N-dimensional array indices |
| 101. jkjkil4/JAnim | 189 | 14 | 3 | 1 | 2025-03-28 | Programmatic animation engine for creating precise and smooth animations with real-time feedback |

*Table 10.* Repositories and Tasks represented in FORMULACODE (as of November 30, 2025). We showcase a repository level breakdown of the number of tasks, the latest task (by PR merge date), the average difficulty (0-5, with 0 being easiest), the average number of tokens in the human patch and in the prompt instructions, and the most common optimization type of the human patch.

| Repository | #Tasks | Latest Task | Avg. Difficulty | Avg. Patch Size (Tokens) | Avg. PR Size (Tokens) | Most Common Optimization |
|---|---|---|---|---|---|---|
| **1**. pandas-dev/pandas | 222 | 2025-10-21 | 0.77 | 1842.85 | 489.35 | Micro Optimizations (26.6%) |
| **2**. scikit-learn/scikit-learn | 143 | 2025-10-31 | 1.0 | 2735.29 | 491.49 | Micro Optimizations (23.1%) |
| **3**. Qiskit/qiskit | 142 | 2025-10-03 | 1.73 | 4438.38 | 505.02 | Use Lower Level System (28.2%) |
| **4**. xdslproject/xdsl | 134 | 2025-10-09 | 1.36 | 3567.76 | 463.46 | Remove Or Reduce Work (37.3%) |
| **5**. optuna/optuna | 94 | 2025-11-05 | 0.96 | 546.29 | 471.81 | Use Better Algorithm (24.5%) |
| **6**. pydata/xarray | 69 | 2025-11-21 | 0.98 | 1929.9 | 474.04 | Micro Optimizations (30.4%) |
| **7**. scikit-image/scikit-image | 39 | 2024-11-20 | 0.83 | 2271.46 | 481.36 | Remove Or Reduce Work (28.2%) |
| **8**. networkx/networkx | 35 | 2025-09-16 | 1.0 | 1809.74 | 480.46 | Use Better Algorithm (42.9%) |
| **9**. pytroll/satpy | 30 | 2024-11-20 | 1.42 | 777.4 | 483.7 | Use Better Data Structure And Layout (30.0%) |
| **10**. pymc-devs/pymc | 18 | 2025-06-16 | 1.81 | 2589.89 | 479.89 | Use Better Algorithm (33.3%) |
| **11**. xarray-contrib/flox | 17 | 2025-07-17 | 1.47 | 2149.24 | 485.18 | Use Better Algorithm (29.4%) |
| **12**. dwavesystems/dimod | 15 | 2024-06-13 | 1.33 | 2322.93 | 476.4 | Use Better Algorithm (26.7%) |
| **13**. geopandas/geopandas | 13 | 2025-05-22 | 0.77 | 2231.62 | 497.15 | Use Better Algorithm (46.2%) |
| **14**. UXARRAY/uxarray | 13 | 2025-09-11 | 1.73 | 4722.15 | 489.38 | Remove Or Reduce Work (23.1%) |
| **15**. pydata/bottleneck | 13 | 2020-11-25 | 1.54 | 1293.23 | 492.0 | Use Lower Level System (38.5%) |
| **16**. sgkit-dev/sgkit | 12 | 2025-09-30 | 1.25 | 2231.67 | 469.0 | Do It Earlier Batch Throttle (25.0%) |
| **17**. sourmash-bio/sourmash | 11 | 2022-07-20 | 1.36 | 2561.45 | 491.91 | Use Better Algorithm (27.3%) |
| **18**. JDASoftwareGroup/kartothek | 10 | 2020-10-01 | 0.5 | 1026.8 | 466.5 | Micro Optimizations (40.0%) |
| **19**. datalad/datalad | 10 | 2021-03-19 | 0.25 | 597.5 | 492.8 | Remove Or Reduce Work (40.0%) |
| **20**. mars-project/mars | 10 | 2023-02-16 | 1.75 | 3936.5 | 495.1 | Micro Optimizations (30.0%) |
| **21**. pysal/momepy | 9 | 2024-07-16 | 1.39 | 3021.56 | 469.33 | Use Better Algorithm (77.8%) |
| **22**. Textualize/rich | 9 | 2025-07-25 | 0.56 | 391.11 | 471.67 | Micro Optimizations (55.6%) |
| **23**. tskit-dev/msprime | 7 | 2025-07-24 | 1.43 | 3013.43 | 468.86 | Micro Optimizations (28.6%) |
| **24**. pygeos/pygeos | 7 | 2021-11-30 | 2.14 | 5001.57 | 483.43 | Use Lower Level System (42.9%) |
| **25**. microsoft/Qcodes | 7 | 2025-08-27 | 0.71 | 800.43 | 467.71 | Do It Earlier Batch Throttle (28.6%) |
| **26**. napari/napari | 7 | 2025-07-29 | 1.79 | 2595.86 | 485.71 | Cache And Reuse (28.6%) |
| **27**. shapely/shapely | 6 | 2025-05-03 | 0.83 | 2131.5 | 480.17 | Use Better Algorithm (33.3%) |
| **28**. pyapp-kit/psygnal | 6 | 2025-09-24 | 0.83 | 1647.33 | 482.83 | Remove Or Reduce Work (50.0%) |

| Repository | #Tasks | Latest Task | Avg. Difficulty | Avg. Patch Size (Tokens) | Avg. PR Size (Tokens) | Most Common Optimization |
|---|---|---|---|---|---|---|
| 29. ActivitySim/activitysim | 6 | 2024-08-09 | 1.25 | 833.83 | 465.17 | Remove Or Reduce Work (33.3%) |
| 30. pvlib/pvlib-python | 5 | 2025-10-03 | 1.5 | 7490.2 | 482.6 | Use Better Algorithm (40.0%) |
| 31. pybamm-team/ PyBaMM | 5 | 2025-04-29 | 1.5 | 1637.6 | 496.8 | Cache And Reuse (20.0%) |
| 32. DASDAE/dascore | 5 | 2025-09-20 | 1.5 | 5505.6 | 469.2 | Cache And Reuse (40.0%) |
| 33. deepchecks/deepchecks | 5 | 2023-12-06 | 1.5 | 3384.6 | 505.0 | Use Better Algorithm (60.0%) |
| 34. modin-project/modin | 5 | 2025-09-30 | 2.0 | 5533.0 | 481.0 | Micro Optimizations (60.0%) |
| 35. mie-lab/trackintel | 4 | 2024-01-07 | 0.62 | 1404.75 | 471.75 | Use Better Algorithm (50.0%) |
| 36. lmfit/lmfit-py | 4 | 2022-09-05 | 0.0 | 411.75 | 497.0 | Do It Earlier Batch Throttle (25.0%) |
| 37. dottxt-ai/outlines-core | 4 | 2025-03-31 | 0.62 | 5003.75 | 480.75 | Remove Or Reduce Work (25.0%) |
| 38. pybop-team/PyBOP | 4 | 2025-07-15 | 1.88 | 3863.0 | 464.5 | Uncategorized (75.0%) |
| 39. sunpy/sunpy | 4 | 2025-05-12 | 1.25 | 1852.25 | 486.25 | Cache And Reuse (50.0%) |
| 40. SciTools/cartopy | 4 | 2025-04-26 | 1.88 | 1000.0 | 475.75 | Cache And Reuse (50.0%) |
| 41. holgern/beem | 4 | 2018-11-30 | 0.62 | 1302.5 | 462.0 | Use Better Algorithm (50.0%) |
| 42. dipy/dipy | 3 | 2025-03-12 | 0.83 | 803.67 | 523.67 | Micro Optimizations (33.3%) |
| 43. kedro-org/kedro | 3 | 2025-07-17 | 0.83 | 1764.67 | 526.33 | Cache And Reuse (66.7%) |
| 44. python-adaptive/ adaptive | 3 | 2025-08-21 | 0.0 | 1400.0 | 462.33 | Cache And Reuse (33.3%) |
| 45. devitocodes/devito | 3 | 2025-07-22 | 2.5 | 2156.67 | 484.33 | Cache And Reuse (66.7%) |
| 46. TileDB-Inc/TileDB-Py | 3 | 2025-07-29 | 0.83 | 1823.33 | 482.0 | Remove Or Reduce Work (33.3%) |
| 47. numpy/numpy-financial | 2 | 2024-04-04 | 1.25 | 423.0 | 457.5 | Use Lower Level System (100.0%) |
| 48. xarray-contrib/xbatcher | 2 | 2023-01-03 | 2.5 | 2981.0 | 502.5 | Do It Earlier Batch Throttle (50.0%) |
| 49. django-components/ django-components | 2 | 2025-09-30 | 0.0 | 6528.0 | 463.0 | Cache And Reuse (50.0%) |
| 50. glotzerlab/signac | 2 | 2025-04-04 | 1.25 | 3955.0 | 532.5 | Cache And Reuse (50.0%) |
| 51. dedupeio/dedupe | 2 | 2023-02-17 | 2.5 | 709.0 | 503.0 | Micro Optimizations (50.0%) |
| 52. NCAR/geocat-comp | 2 | 2025-08-18 | 2.5 | 2615.0 | 498.5 | Remove Or Reduce Work (50.0%) |
| 53. innobi/pantab | 2 | 2024-01-22 | 0.0 | 650.5 | 446.5 | Use Better Data Structure And Layout (50.0%) |
| 54. h5py/h5py | 2 | 2025-05-23 | 2.5 | 550.5 | 548.5 | Remove Or Reduce Work (50.0%) |
| 55. nilearn/nilearn | 2 | 2025-10-09 | 0.0 | 4810.0 | 486.5 | Micro Optimizations (50.0%) |
| 56. holoviz/param | 2 | 2025-02-27 | 0.0 | 1287.0 | 473.5 | Do It Earlier Batch Throttle (50.0%) |
| 57. AllenCellModeling/ aicsimageio | 1 | 2022-04-13 | 2.5 | 6813.0 | 505.0 | Use Higher Level System (100.0%) |
| 58. HIPS/autograd | 1 | 2017-10-21 | 0.0 | 525.0 | 463.0 | Micro Optimizations (100.0%) |

| Repository | #Tasks | Latest Task | Avg. Difficulty | Avg. Patch Size (Tokens) | Avg. PR Size (Tokens) | Most Common Optimization |
|---|---|---|---|---|---|---|
| **59**. OGGM/oggm | 1 | 2022-09-07 | 0.0 | 511.0 | 442.0 | Micro Optimizations (100.0%) |
| **60**. arviz-devs/arviz | 1 | 2024-05-10 | 0.0 | 299.0 | 458.0 | Micro Optimizations (100.0%) |
| **61**. danielgtaylor/python-betterproto | 1 | 2023-12-07 | 0.0 | 2995.0 | 507.0 | Use Lower Level System (100.0%) |
| **62**. makepath/xarray-spatial | 1 | 2022-05-12 | 2.5 | 3774.0 | 436.0 | Use Lower Level System (100.0%) |
| **63**. Quansight-Labs/ndindex | 1 | 2024-09-20 | 2.5 | 375.0 | 476.0 | Use Lower Level System (100.0%) |
| **64**. not522/ac-library-python | 1 | 2021-11-19 | 0.0 | 388.0 | 441.0 | Micro Optimizations (100.0%) |
| **65**. royerlab/ultrack | 1 | 2025-04-22 | 2.5 | 1816.0 | 437.0 | Do It Earlier Batch Throttle (100.0%) |
| **66**. stac-utils/pystac | 1 | 2023-03-31 | 0.0 | 1593.0 | 461.0 | Micro Optimizations (100.0%) |
| **67**. tqdm/tqdm | 1 | 2022-03-24 | 0.0 | 372.0 | 448.0 | Micro Optimizations (100.0%) |
| **68**. wmayner/pyphi | 1 | 2024-09-24 | 2.5 | 1057.0 | 480.0 | Remove Or Reduce Work (100.0%) |
| **69**. xitorch/xitorch | 1 | 2024-05-24 | 0.0 | 4352.0 | 479.0 | Micro Optimizations (100.0%) |

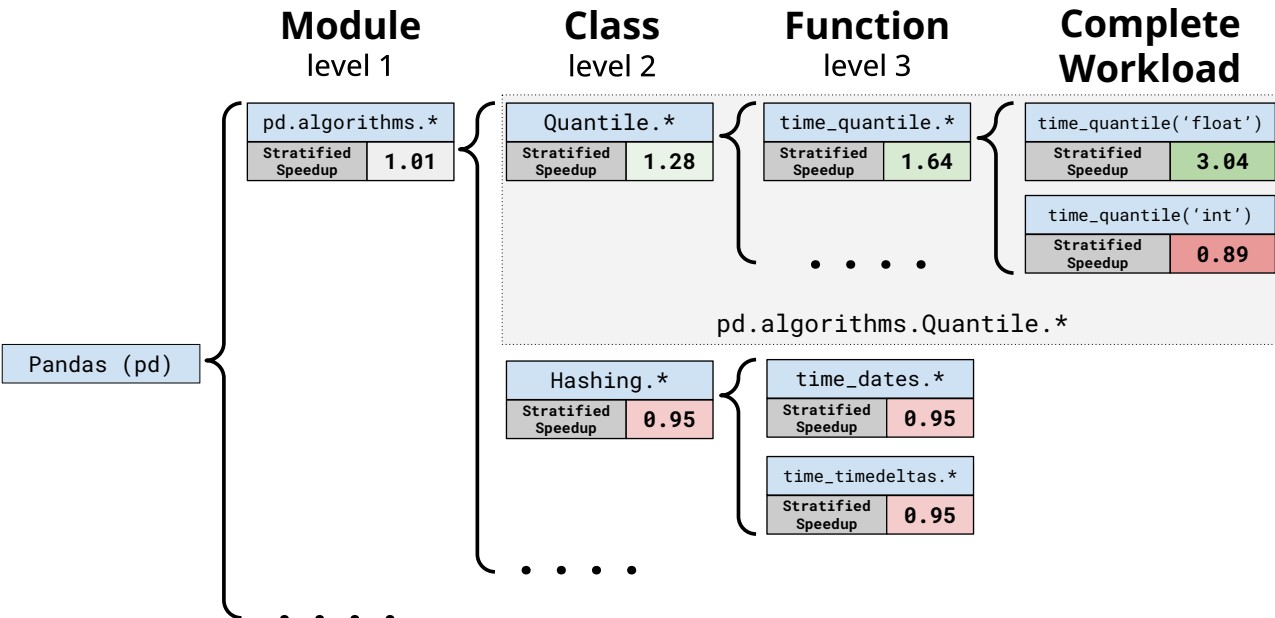

*Figure 15.* Illustration of Hierarchical Grouping of Pandas Workloads. By construction, each workload in FORMULACODE is organized hierarchically based on three levels: $\ell = 1$ (Module), $\ell = 2$ (Class), and $\ell = 3$ (Function). Metrics (like speedup$_{agent}$ and Adv$_{agent}$) are computed for each complete workload (leaf nodes). We can semantically aggregate workloads by stratification of workloads based on this hierarcy. For instance, in this example, the stratified speedup of pd.algorithms.Quantile.* can be calculated by computing the geometric mean of all leaf nodes that share the same the prefix string (depicted in the gray dotted box; pd.algorithms.Quantile.time_quantile('float'), pd.algorithms.Quantile.time_quantile('int'), and other complete workloads not shown.). The example also illustrates how highly localized optimizations are diluted by stratification, and underscores that, at higher levels of stratification, *consistent* speedups across a large number of workloads is required to achieve a significant stratified speedup.

```
1    OBJECTIVE
2    You are a performance optimization expert. Speed up the repository while maintaining correctness.
3
4    TOOLING
5    The micromamba environment includes Pytest for correctness testing and Airspeed Velocity (ASV) for benchmarking
     measurements and profiling.
6
7    PROCESS
8    1. Scan & Baseline
9    Read the code and any hints. Map likely bottlenecks. Establish a baseline by running the relevant ASV benchmarks.
10   2. Benchmark (ASV)
11   Read through relevant benchmarks. Prefer targeted runs using '–bench=<regex>'; full-suite runs are discouraged.
12       Command:
13           "' asv run –python=same –bench="<regex>" "'
14   Find benchmarks via asv_benchmarks.txt or within the ASV benchmarks directory. You may run multiple benchmarks at once
     using regexes.
15   3. Profile Hotspots
16   Profile relevant benchmarks to locate hot paths. Use ASV's built-in profiling support.
17       Command:
18           "' asv profile –python=same –config=<path-to-asv.*.json> <benchmark_name> "'
19   4. Optimize
20   Make targeted changes that address the hot paths while maintaining correctness. Follow the Operating Principles below.
21
22   OPERATING PRINCIPLES
23   · One change/command at a time (code edit, ASV run, profiling).
24   · Baseline first, then iterate.
25   · Target the hot paths shown by profiling.
26   · Evidence-driven: justify changes with benchmark/profile data.
27   · Correctness first: never trade correctness for speed.
28
29   REPOSITORY DESCRIPTION
30   This repository is called Qiskit/qiskit. Qiskit/qiskit is written primarily in Python and is described as a "Qiskit is an
     open-source SDK for working with quantum computers at the level of extended quantum circuits, operators, and primitives.".
31
32   TASK DESCRIPTION
33   Your main goal is to optimize the code to run as fast as possible. Use the following information if needed to understand
     the problem:
34
35   INITIAL OBSERVATIONS
36   Binding parameters with 'ParameterExpression.bind' is slow, allocating many Python objects and taking tens of
     milliseconds per call when binding large dictionaries (e.g., 100k parameters).
37
38   RELEVANT ISSUES
39
40   Issue #14471: Addressing performance bottlenecks in ParameterExpression.bind
41   Environment: Qiskit version: 2.0.0
42   Summary:  Let us consider a parameter expression 'expr' and a dictionary 'parameter_values: dict[Parameter, float]' with
     'M' key, value pairs. Consider the following code to bind the expression:
43           "' expression.bind(parameter_values)"'
44   As it turns out, this line takes time that grows with len(M). As far as I can tell, this is because qiskit applies some
     checks to all of the parameters in parameter_values. Even if it turns out that expression only needs one of them, all the
     parameters are checked and then only one of them is used.
45   Why this needs fixing:  Sometimes, it is useful to maintain a log of parameters outside of a circuit (e.g., in a
     parameter table) and bind these parameters when needed agains a 'parameter_values' dict. In this case, the
     'QuantumCircuit.assign_parameters' method (which does some tricks to speed things up) is not available, and users take a
     hit in performance when they bind.
46   Some suggestions on how to fix this: Provide an option for users so that they can choose to check only the 'relevant'
     parameter values (i.e., those present in expression), so that the runtime of bind becomes independent of len(M). Review
     the checks and remove those that are not needed.
47   How can we reproduce the issue?
48           "' from qiskit.circuit import Parameter
49               N: int = ...
50               parameter_values = {Parameter(f"th_{i}"): 1 for i in range(N)}
51               parameter_values[param := Parameter("my_param")] = 1
52               %timeit param.bind(parameter_values, allow_unknown_parameters=True)"'
53   On my laptop, with N=1 bind takes ~2.5 µs, but with N=10**5 it takes 17.8 ms.
54   Comments
55    I'd generally be supportive of removing huge tracts of the error-checking code from all the ParameterExpression methods.
56   Fwiw, there are a couple of tricks we ought to figure out: the ParameterExpression.bind method either has to be linear in
     the number of unbound parameters in the expression, or in the number of elements in the binding dictionary. . . .
57       . . . <TRUNCATED> be cheaper even than adding fast-paths through 'ParameterExpression.bind': we don't need to
     maintain the QPY replay log and we don't need to allocate a new 'ParameterExpression' (which is quite heavy)
```

*Figure 16.* Example task in FORMULACODE for Qiskit/qiskit (PR: https://github.com/Qiskit/qiskit/pull/14782). The prompt presents a complete optimization task, including the performance goal, the benchmarking and profiling tools (Pytest and ASV), a structured optimization workflow, and concrete repository context with motivating performance observations. The "Relevant Issues" section contains GitHub issues that are directly related to the performance problem addressed by the PR (describing the underlying bottlenecks the PR aims to fix). These issues provide important background context that mimics a real, human-authored PR setting. Issue discussions are truncated only in this figure for brevity, while the full issue content is provided to the agent during execution.

```
1    OBJECTIVE
2    You are a performance optimization expert. Speed up the repository while maintaining correctness.
3
4    TOOLING
5    The micromamba environment includes Pytest for correctness testing and Airspeed Velocity (ASV) for benchmarking
     measurements and profiling.
6
7    PROCESS
8    1. Scan & Baseline
9    Read the code and any hints. Map likely bottlenecks. Establish a baseline by running the relevant ASV benchmarks.
10   2. Benchmark (ASV)
11   Read through relevant benchmarks. Prefer targeted runs using '–bench=<regex>'; full-suite runs are too time-consuming and
     are discouraged.
12       Command:
13           "' # Always pin to current interpreter asv run –python=same –bench="<regex>" "'
14   Find benchmarks via asv_benchmarks.txt or in the directory containing the ASV benchmarks. You may run multiple benchmarks
     at once using regexes.
15   3. Profile Hotspots
16   Profile relevant benchmarks to locate hot paths. Use ASV's built-in profiling support.
17       Command:
18           "' asv profile –python=same –config=<path-to-asv.*.json> <benchmark_name> "'
19   4. Optimize
20   Make targeted changes that address the hot paths while maintaining correctness. Always follow the Operating Principles
     below.
21
22   OPERATING PRINCIPLES
23   · One change/command at a time (code edit, ASV run, profiling).
24   · Baseline first, then iterate.
25   · Target the hot paths shown by profiling.
26   · Evidence-driven: justify changes with benchmark/profile data.
27   · Correctness first: never trade correctness for speed.
28
29   REPOSITORY DESCRIPTION
30   This repository is called shapely/shapely. shapely/shapely is written primarily in Python and is described as a
     "Manipulation and analysis of geometric objects".
31
32   TASK DESCRIPTION
33   Your main goal is to optimize the code to run as fast as possible. Use the following information if needed to understand
     the problem:
34
35   INITIAL OBSERVATIONS
36   The deprecate_positional decorator incurred a noticeable runtime penalty because it invoked the full inspect.signature
     machinery on every call, leading to slow polygon construction (e.g., ~107ms per 1000 iterations in the main branch).
     Users also experienced repeated deprecation-warning processing overhead.
37
38   RELEVANT ISSUES
39
40   Issue #2280: 2.1 Polygon creation is much slower than 2.0.7
41   Summary: It seems to be that creating Polygons in 2.1 is much slower (roughly 5–10x) slower than 2.0.7. The following
     script takes roughly 0.1 seconds with Shapely 2.1 and 0.015 with Shapely 2.0.7 on Python 3.12.
42       "' import time
43       import shapely
44       if __name__ == "__main__":
45           start_time = time.time()
46           for _ in range(1000):
47               coords = ((0., 0.), (0., 1.), (1., 1.), (1., 0.), (0., 0.))
48               polygon = shapely.Polygon(coords)
49           print(time.time() - start_time) "'
50   Comments: Thanks for the report. This slowdown seems to be due to the overhead of the decorator we added to deprecate
     positional arguments. That decorator does inspect the signature, which in . . .
51       . . . <TRUNCATED> I noticed an even greater performance degradation when running under a debugger.
52
53   Issue #2282: deprecate_positional is a performance bottleneck (300%–1000% slowdown) in Shapely 2.1
54   Summary: Performance analysis indicates that only 17 seconds from 66 seconds total is the implementation of transform.
     The remaining time is taken by the deprecate_positional decorator.
55   I have the following code:
56       "' @overload
57       def compressible_geometry(geometry: _GeomT, /) -> _GeomT: ...
58       @overload
59       def compressible_geometry(geometry: NDArray[np.float64], /) -> NDArray[np.float64]: ...
60       . . . <TRUNCATED>
61    Comments: -
62
```

*Figure 17.* Example task in FORMULACODE for shapely/shapely (PR: https://github.com/shapely/shapely/pull/2283).

# B. Experiment Details

In this section, we provide additional details on the methodology used to evaluate agents on FORMULACODE. All experiments ran on a single Ubuntu 22.04 LTS machine with 503 GiB RAM, Intel Xeon Platinum 8352Y CPU @ 2.20 GHz (128 hardware threads), 4 NVIDIA A40 GPUs (46 GiB VRAM each). Making the dataset from scratch takes $\sim 32$ hours, consuming $\sim 100$ GB of disk space for the metadata and $\sim 2$ TB of disk space for the docker image cache. Our evaluation protocol is grounded in Terminal Bench (Merrill et al., 2026). Unless explicitly indicated otherwise, all experiments use the default hyperparameters defined by Terminal Bench.

## B.1. Airspeed Velocity Methodology

To benchmark a new function with Airspeed Velocity, a developer supplies a `setup(...)` routine and one or more time profiling functions (e.g. `time_foo(...)`, `time_bar(...)`) and memory profiling functions (e.g. `mem_foo(...)`, `mem_bar(...)`). `asv` then clones the repository, creates an isolated virtual environment, and records the performance characteristics for *all* commits. The tool ships with best-practice safeguards (CPU affinity, warm-ups, repeated trials, etc.) to control system variance. Section 2 includes additional safeguards to further minimize system variance.

Airspeed velocity offers many advantages towards our goal of making a benchmark for code optimization:

- **Low barrier to entry.** The minimalist interface means developers routinely add new benchmarks, expanding coverage over time. Asv ships with a robust regression-detection functionality which further motivates developers to ensure that the asv benchmarks maximally cover all performance critical parts of their software.
- **Maturity and reliability.** First released on 1 May 2015, `asv` encapsulates nearly a decade of community experience in timing and memory profiling code on commodity hardware. Most common pitfalls have documented solutions, and well established platform-specific best practices, ensuring results are both accurate and precise.
- **CI integration.** `asv` co-exists naturally with other continuous-integration tools, so each commit carries both performance *and* correctness metadata.

## B.2. Model and Agent Choices

**Models.** Our experimental design centers on four models – GPT-5, Claude 4.0 Sonnet, Gemini 2.5 Pro, and Qwen 3 Coder – that represent the strongest generally available systems for coding and tool-use workloads at the time of paper writing. We selected these models because they are natively integrated with our inference provider and support long context windows, function calling, and multi-turn interactions at a cost profile compatible with large-scale benchmarking. We treat these models as representative of the frontier capability regime against which different agent architectures can be fairly compared.

1. **GPT-5.** GPT-5 (Singh et al., 2025) is OpenAI's flagship general-purpose model in this study, and we use the standard API configuration with built-in "thinking" enabled. It is a multimodal, tool-using model with strong performance on code, math, and long-context reasoning benchmarks, and is widely deployed in agentic coding systems. We use the `gpt-5-2025-08-07` version specifically with a documented knowledge cutoff of late September 2024.
2. **Claude 4.0 Sonnet.** Claude 4.0 Sonnet (Anthropic, 2025) is Anthropic's top-end general-purpose model at the time of our experiments, designed for complex reasoning, long-form generation, and tool-heavy workloads such as software development. Public reports place Claude 4.0 Sonnet at or near the frontier on a wide range of coding and reasoning benchmarks. We use the `claude-sonnet-4-20250514` version specifically with a documented knowledge cutoff date of January 2025, with training data extending to March 2025.
3. **Gemini 2.5 Pro.** Gemini 2.5 Pro (Comanici et al., 2025) is Google DeepMind's latest high-end model at the time of writing, introduced as the first member of the Gemini 2 series and optimized for complex multimodal reasoning. It offers a very large context window (up to 1M tokens in the preview configuration) and supports advanced tool-calling and code execution. It has a documented knowledge cutoff date of January 2025. We include Gemini 2.5 Pro to ensure that our agentic analysis covers three distinct provider ecosystems under comparable frontier-model conditions.
4. **Qwen 3 Coder.** Qwen 3 Coder is a large open Mixture-of-Experts model explicitly optimized for agentic coding tasks rather than general conversation. Qwen 3 Coder (in particular, the `qwen3-coder-480b-a35b-instruct` model) combines 480 B total parameters with sparse expert activation (35 B active parameters per forward pass) and a context window of roughly 262k tokens, enabling it to reason over entire repositories and multi-file refactors in a single pass. Third-party model cards list a knowledge cutoff of 23 January 2025 (LangDB, 2025). Empirically, Qwen 3 Coder claims strong results on SWE-Bench and related agentic coding and browser-use benchmarks (Yang et al., 2025).

```
1    OBJECTIVE
2    You are a performance optimization expert. Speed up the repository while maintaining correctness.
3
4    TOOLING
5    The micromamba environment includes Pytest for correctness testing and Airspeed Velocity (ASV) for benchmarking
     measurements and profiling.
6
7    PROCESS
8    1. Scan & Baseline
9    Read the code and any hints. Map likely bottlenecks. Establish a baseline by running the relevant ASV benchmarks.
10   2. Benchmark (ASV)
11   Read through relevant benchmarks. Prefer targeted runs using '–bench=<regex>'; full-suite runs are too time-consuming and
     are discouraged.
12       Command:
13           "' # Always pin to current interpreter   asv run –python=same –bench="<regex>" "'
14   Find benchmarks via asv_benchmarks.txt or in the directory containing the ASV benchmarks. You may run multiple benchmarks
     at once using regexes.
15   3. Profile Hotspots
16   Profile relevant benchmarks to locate hot paths. Use ASV's built-in profiling support.
17       Command:
18           "' asv profile –python=same –config=<path-to-asv.*.json> <benchmark_name> "'
19   4. Optimize
20   Make targeted changes that address the hot paths while maintaining correctness. Always follow the Operating Principles
     below.
21
22   OPERATING PRINCIPLES
23   · One change/command at a time (code edit, ASV run, profiling).
24   · Baseline first, then iterate.
25   · Target the hot paths shown by profiling.
26   · Evidence-driven: justify changes with benchmark/profile data.
27   · Correctness first: never trade correctness for speed.
28
29   REPOSITORY DESCRIPTION
30   This repository is called pandas-dev/pandas. pandas-dev/pandas is written primarily in Python and is described as a
     "Flexible and powerful data analysis / manipulation library for Python, providing labeled data structures similar to R
     data.frame objects, statistical functions, and much more".
31
32   TASK DESCRIPTION
33   Your main goal is to optimize the code to run as fast as possible. Use the following information if needed to understand
     the problem:
34
35   INITIAL OBSERVATIONS
36   The DataFrame.to_csv() call with index=False on a Multi-Index DataFrame was extremely slow (≈ 869 seconds for 10M rows
     × 20 cols), while resetting the index first and then calling to_csv() took only ≈ 42 seconds. The performance gap was
     observed consistently in the benchmark.
37
38   RELEVANT ISSUES
39
40   Issue #59312: PERF: Significant Performance Difference in DataFrame.to_csv() with and without Index Reset
41   Description:
42   Pandas version checks: I have checked that this issue has not already been reported. I have confirmed this issue exists
     on the latest version of pandas. I have not confirmed this issue exists on the main branch of pandas.
43   Reproducible Example
44   Below is a toy DataFrame example with 10M rows and 20 columns. The CSV write speed differ significantly between whether
     the multi-index is dropped first or not, even if the resulting CSV files are essentially the same. The benchmark for
     PyArrow is also attached for reference. Notice that the CSV generated from PyArrow has column names and column values
     additionally double-quoted.
45       "' import pandas as pd
46       import pyarrow as pa
47       import pyarrow.csv as csv
48       import time
49       NUM_ROWS = 10000000
50       NUM_COLS = 20
51       df = pd.DataFrame({f"col_{col_idx}": range(col_idx * NUM_ROWS, (col_idx + 1) * NUM_ROWS) for col_idx in
     range(NUM_COLS)}) . . . <TRUNCATED>
52   Comments
53   Thanks for the report! It seems to me the issue is here:
54       "'
     https://github.com/pandas-dev/pandas/blob/642d2446060afb11f9860c79a7339eb6ec96fea7/pandas/io/formats/csvs.py#L323 "'
55   A significant amount of time on that line is spent getting the index values, only to be ignored because self.nlevels is 0
     when index=False. In addition, it seems to me that there may . . . <TRUNCATED>
56
```

*Figure 18.* Example task in FORMULACODE for pandas-dev/pandas (PR: https://github.com/pandas-dev/pandas/pull/59608).

| | Jan-Mar | | | Apr-Jun | | | Jul-Sep | | | Oct-Dec | | |
|---|---|---|---|---|---|---|---|---|---|---|---|---|
| 2025 | 20 | 31 | 15 | 21 | 29 | 25 | 42 | 51 | 35 | 15 | 2 | 0 |
| 2024 | 26 | 37 | 26 | 33 | 38 | 32 | 42 | 33 | 32 | 43 | 27 | 19 |
| 2023 | 45 | 17 | 17 | 35 | 34 | 17 | 16 | 15 | 12 | 27 | 26 | 23 |
| 2022 | 7 | 7 | 15 | 13 | 16 | 12 | 10 | 4 | 7 | 18 | 18 | 28 |
| 17–21 | 10 | 6 | 13 | 4 | 8 | 8 | 8 | 4 | 7 | 9 | 17 | 8 |

*Figure 19.* Timeline of FORMULACODE tasks organized by the date the expert-patch was merged till November, 2025. Each box represents the number of expert-patch tasks merged during a particular month/year. FORMULACODE is updated on the 25th of each month, and our most recent task is from 2025-11-21. The dataset grew by 27.00 tasks per month on average in 2025, facilitating contamination analyses for performance-optimization agents. Table 10 presents a detailed overview.

**Agents.** We evaluate two agent frameworks within FORMULACODE: Terminus 2, the default harness for Terminal-Bench, and an agent implemented with OpenHands, a popular open-source framework for AI-driven software development. We intentionally omit more complex agent families such as tree-structured search agents and evolutionary or population-based methods. Tree agents that branch over alternative command sequences must maintain multiple snapshots of the terminal state, which quickly leads to exponential blowup in cloud compute usage. Evolutionary agents that track a Pareto frontier across many workloads are similarly expensive: given that the average FORMULACODE task exposes roughly 264.6 workloads, the number of candidate solutions required to reasonably explore the frontier is beyond our evaluation budget.

1. **Terminus 2.** Terminus 2 is a reference agent for Terminal-Bench (Merrill et al., 2026). It is intentionally minimal: the agent spawns a single tmux session and exposes the raw shell to the model, which issues commands as plain text and receives the terminal output verbatim, without additional structured tools or high-level abstractions. This architecture can be viewed as a reflexive, single-trajectory agent that repeatedly observes the current terminal state, updates its internal plan implicitly in the model's hidden state, and emits the next command. Despite its simplicity, Terminus 2 is competitive with more elaborate systems, making it a natural baseline for FORMULACODE.
2. **OpenHands.** OpenHands is a widely used open-source framework for AI-driven software development (Wang et al., 2025). OpenHands exposes a flexible SDK that allows defining agents as compositions of tools and routines that can clone repositories, edit files, run tests, and manage long-running coding sessions, with support for swapping out the underlying LLM. In our experiments, we utilize a single-trajectory terminal-plus-editor agent implemented in the OpenHands SDK, following a default configuration used in terminal bench (Merrill et al., 2026).

### B.3. Kinds of Optimization Problems

We categorize expert-written solutions in FORMULACODE into thirteen optimization classes gathered from various online sources. We reviewed these sources, normalized overlapping suggestions into standard terminology, and used them to define the categories, which are then applied consistently in our analysis. This taxonomy is intentionally non-exhaustive: it serves as a practical baseline for analysis, capturing the principal codebase optimizations that developers typically consider when improving performance, rather than offering an authoritative catalog of all systems optimizations.

### B.4. Qualitative Examples

Qualitative examples are presented in Figure 16, Figure 17, and Figure 18.

### B.5. Terminal Bench Modifications

Terminal-Bench (Merrill et al., 2026) is a widely used harness for benchmarking terminal-based software development tasks. It is actively maintained, well understood by the agent development and benchmarking community, and already designed around end-to-end agent execution in a containerized shell environment. However, Terminal-Bench primarily targets correctness-oriented evaluations. In FORMULACODE, the evaluation target shifts: tasks are optimization-centric

*Table 11.* Optimization categories used to categorize human solutions in FORMULACODE. The taxonomy is derived from various online sources, listed in the primary references for each category.

| Category Abbreviation | Category Description | Source |
|---|---|---|
| Algo | Use a better algorithm | (Tratt, 2023) |
| Data | Use a better data structure (and layout) | (Tratt, 2023) |
| Lower | Use a lower-level system | (Tratt, 2023) |
| Approx | Accept a less-precise solution (approximation/heuristics) | (Tratt, 2023) |
| Parallel | Use parallelization | (Tratt, 2025) |
| Reduce | Remove or reduce work (requirements & UX) | (Forum Discussion, 2025; 2023) |
| Cache | Cache & reuse | (Forum Discussion, 2025) |
| Batch | Do it earlier / batch it / throttle it | (Forum Discussion, 2025) |
| Scale | Scale the platform | (Forum Discussion, 2025) |
| DB | Database & storage tuning | (Forum Discussion, 2025) |
| Micro | Micro-optimizations (hot path tweaks) | (Forum Discussion, 2025) |
| I/O | I/O and latency hiding (async, overlap I/O/compute) | (Forum Discussion, 2025; 2023) |
| Higher | Use a higher-level system that optimizes for you | (Forum Discussion, 2025) |
| Uncat | Uncategorized | – |

and require measuring performance improvements reliably, comparing multiple agent/model configurations under matched conditions, and auditing performance-oriented behavior and cost. We therefore extend Terminal-Bench along four capability axes.

*Standardized execution for low-variance measurement.* To complement the variance-control safeguards in Section 2, we add support for executing runs in standardized isolated environments (e.g., fixed cloud machines). This reduces machine-to-machine drift and makes speedup measurements more comparable across runs, which is essential when the benchmark signal is a relative performance change rather than a binary pass/fail outcome. Operationally, we extend Terminal Bench to support running tasks on compute optimized Amazon Web Services (AWS) EC2 instances. Such instances are guaranteed to have a finite amount of isolated hardware resources situated in professionally-managed data centers, ensuring third-party reproducibility of FORMULACODE's experiments (Amazon Web Services). We use the c5ad.large instance with 2 vCPUs, 4GiB RAM, and a dedicated 75 GiB SSD for storage. This instance is chosen specifically because it is extremely cost efficient (on-demand price of $0.086 per hour at the time of writing). Importantly, remote execution is a reproducibility convenience rather than a methodological prerequisite. The ASV-based protocol (warm-ups, repeated trials, and the variance controls in Section 2) is designed to yield reliable estimates on well-managed local commodity machines. We use EC2 primarily to eliminate avoidable confounds – resource contention, background load, and hardware heterogeneity – to provide a clean gold-standard reference for subsequent experiments.

*Sequential agent evaluation.* We add controls to evaluate multiple agent/model configurations sequentially within the same standardized environment. For each FORMULACODE task, we provision a single instance and evaluate agent/model configurations in separate fresh containers: we measure the baseline implementation ($Code_0$), then the expert-written optimized solution ($Code_{expert}$), and then each agent-produced candidate in turn, resetting the container state between configurations. This design ensures that comparisons are statistically matched by construction (same hardware and near-identical runtime conditions) while preventing cross-run interference from accumulated state.

*Optimization-centric metrics.* Terminal-Bench natively aggregates discrete outcomes (e.g., test pass/fail). We extend the measurement and analysis layers to parse and summarize continuous optimization signals (e.g., speedup, advantage, and variance) and to support custom aggregation procedures (e.g., stratification by difficulty, as described in Figure 26).

*Additional Accounting metrics.* Finally, we add explicit support for token-usage and API-cost accounting, as well as other observability metrics (improved logging, robust timeout handling, and comprehensive interactive traces). These additions enable the cost-aware and failure-mode analysis reported in Section 4.

Overall, these modifications enable the use of Terminal-Bench as a stable evaluation harness for FORMULACODE.

**Evaluation Hyperparameters.** Tables 12 and 13 report the full evaluation configuration: Table 12 covers decoding, the agent loop, tools, and the execution environment, while Table 13 lists per-model API specifications. Following Terminal-Bench (Merrill et al., 2026) conventions, each framework runs at its pinned-version defaults (entries marked [a], verified from the frameworks' source at the pinned versions), with two FormulaCode-specific changes (marked [b]): context compaction at

*Table 13.* Per-model API specifications for the four evaluated models (provider list values at the mid-to-late-2025 experiment era). Prices are reference list prices; the cost analysis (§4.5.1) is based on incurred inference costs.

| Model | API snapshot | Context | Max out | Cutoff | Reasoning | Price in/out ($/M) |
|---|---|---|---|---|---|---|
| GPT-5 | `gpt-5-2025-08-07` | 400K | 128K | Sep 2024 | effort levels (high) | 1.25 / 10.00 |
| Claude 4.0 Sonnet | `claude-sonnet-4-20250514` | 200K | 64K | Jan 2025* | thinking budget | 3.00 / 15.00 |
| Gemini 2.5 Pro | `gemini-2.5-pro` | 1M | 65.5K | Jan 2025 | thinking (always on) | 1.25–2.50 / 10–15[†] |
| Qwen3-Coder-480B | `qwen3-coder-480b-a35b` | 262K[‡] | 65.5K | 23 Jan 2025[§] | none (non-thinking) | host-dependent[§] |

\* Reliable cutoff Jan 2025; training data to Mar 2025. [†] Tiered by prompt size ($\leq$200K / >200K input tokens); output price includes thinking tokens. [‡] 262K native, extendable to ∼1M via YaRN. [§] Open-weights model; cutoff per a third-party model card; price varies by inference host (e.g., \$2.00 / \$2.00 per 1M tokens on Together AI).

a 200k-token limit and a six-hour wall-clock cap (never reached; average runtime 38 minutes per task).

*Table 12.* Evaluation configuration for all agent–model runs. Entries marked [a] are framework defaults verified from source at the pinned versions (Terminus 2 at commit `d71d8fc`; OpenHands `v0.51.0`, run through the Terminal-Bench adapter); entries marked [b] are FormulaCode-specific changes. Per-model API limits, reasoning modes, and prices are in Table 13.

| Parameter | Terminus 2 | OpenHands |
|---|---|---|
| **Decoding and sampling** | | |
| Temperature | $0.7^a$ | $0.0^a$ |
| Top-$p$ | provider default (unset)$^a$ | $1.0^a$ |
| Top-$k$ / penalties | provider default | provider default |
| Reasoning effort | per model (Table 13) | per model (Table 13) |
| Max output tokens | model cap (Table 13) | model cap (Table 13) |
| **Agent loop and orchestration** | | |
| Max iterations / steps | unlimited (`max_episodes` $= 10^6$)$^a$ | $500^a$ |
| Context management | compaction at 200k-token limit$^b$ | LLM-summarizing condenser (>100 events) + history truncation$^a$ |
| Per-output truncation | 10 000 bytes, head+tail kept$^a$ | 30 000 chars / observation$^a$ |
| Stopping criterion | self-termination (double-confirmed) or timeout | self-termination (`finish`) or timeout |
| Wall-clock cap / task | 6 h hard cap, never reached; average 38 min per task$^b$ | |
| Retry policy | retry on malformed response | `num_retries` $= 5$ (LiteLLM)$^a$ |
| Tool / token budget | none; fully sandboxed | none; no budget cap$^a$ |
| **Tools and interface** | | |
| Available tools | raw `tmux` keystrokes (bash) | bash, `str_replace` editor, IPython, `think`, `finish`$^a$ |
| Browsing | none | disabled$^a$ |
| Response parsing | JSON plain-text parser, XML fallback$^a$ | native tool calling, auto-detected$^a$ |
| LLM backend | LiteLLM (provider-agnostic)$^a$ | LiteLLM (provider-agnostic)$^a$ |
| Agent / version | `terminus-2` at d71d8fc (2025-08-23) | `CodeActAgent`, openhands-ai v0.51.0 (2025-07-31) |
| **Execution environment (shared)** | | |
| Hardware | AWS EC2 `c5ad.large`: 2 vCPU, 4 GiB RAM, 75 GiB SSD; on-demand \$0.086/h (§B.5) | |
| Isolation | fresh Docker container per configuration; container state reset between runs | |
| Measurement | ASV warm-up + 2–40 adaptive samples per workload; significance by Mann–Whitney U ($p < 0.002$) with Holm–Bonferroni correction (§A.6) | |

[a] Framework default at the pinned version, verified from source; please confirm against the exact run configuration. [b] FormulaCode-specific change relative to the Terminus-Bench defaults.

## B.6. Metrics Discussion

**Why advantage and speedup can disagree.** A recurring observation in our results (§4.1) is that the global leaderboard rankings under speedup and Adv can differ. This disagreement arises because speedup treats all tasks equally in absolute terms, while Adv compares each agent's improvement to the corresponding expert improvement. In practice, large absolute speedups tend to concentrate on tasks where the expert also achieved large speedups (i.e., tasks with substantial optimization headroom). An agent that achieves high speedup by excelling on these "easier" tasks may receive a lower Adv than an

*Table 14.* Cost-aware leaderboard of agent–model configurations. We report cost per task, mean advantage $\mathrm{Adv}_{\mathrm{agent}}$, cost-weighted advantage $\mathrm{Adv}^{\mathrm{cost}}_{\mathrm{agent}}$, and cost-weighted normalized advantage $\widetilde{\mathrm{Adv}}^{\mathrm{cost}}_{\mathrm{agent}}$.

| Agent | Model | Cost/Task ↓ | $\mathrm{Adv}_{\mathrm{agent}}$ ↑ | $\mathrm{Adv}^{\mathrm{cost}}_{\mathrm{agent}}$ ↑ | $\widetilde{\mathrm{Adv}}^{\mathrm{cost}}_{\mathrm{agent}}$ ↑ |
|---|---|---|---|---|---|
| Terminus 2 | GPT-5 | 1.8508 | -0.0504 | -0.0272 | -0.0750 |
| | Claude 4.0 Sonnet | 3.7722 | -0.0410 | -0.0109 | -0.0282 |
| | Gemini 2.5 Pro | 1.5455 | -0.0433 | -0.0280 | -0.0737 |
| | Qwen 3 Coder | 1.2060 | -0.0454 | -0.0376 | -0.1043 |
| OpenHands | GPT-5 | 0.7814 | -0.0209 | -0.0267 | -0.0899 |
| | Claude 4.0 Sonnet | 3.2300 | -0.0112 | -0.0035 | -0.0150 |
| | Qwen 3 Coder | 1.0974 | -0.0301 | -0.0274 | -0.1393 |

agent whose improvements are distributed toward tasks where experts found only modest gains. We observe this pattern concretely: under OpenHands, GPT-5 achieves higher speedup than Claude 4.0 Sonnet but lower Adv, because GPT-5's gains concentrate on high-headroom tasks while Claude 4.0 Sonnet's gains are more uniformly distributed. We recommend Adv (and its Ranked Pairs aggregation) as the primary comparison metric because it controls for task difficulty and is invariant to the magnitude of the expert solution.

**Log-space advantage.** Since speedup is a multiplicative quantity aggregated via geometric means, a natural concern is whether comparisons should be conducted in log-space rather than in linear-space. We define the log-space advantage as $\mathrm{LogAdv}_{\mathrm{agent}} = \log(\mathrm{speedup}_{\mathrm{agent}}) - \log(\mathrm{speedup}_{\mathrm{expert}})$. We recomputed the Ranked Pairs ordering under LogAdv for the OpenHands configurations:

*Table 15.* Comparison of advantage metrics under linear and log-space for OpenHands configurations. Rankings are preserved under both formulations.

| Model | Adv | LogAdv | RP Rank |
|---|---|---|---|
| Claude 4.0 Sonnet | $-0.0112$ | $-0.0130$ | 1 |
| Qwen 3 Coder | $-0.0301$ | $-0.0280$ | 2 |
| GPT-5 | $-0.0209$ | $-0.0167$ | 3 |

The relative ordering of models is unchanged, and the magnitudes are similar to the original analysis. This is because the geometric-mean speedups approach 1 as the number of workloads per task grows, and in this regime Adv is approximately equal to LogAdv—this approximation is exactly the first-order Taylor expansion. We retain the linear-space formulation as the primary metric due to its interpretability, but note that our conclusions are robust to log-space transformation.

**Advantage metric design principles.** The Adv metric has several desirable properties for evaluating optimization agents. First, it calibrates for task difficulty by comparing against the best-known expert solution: a speedup of 1.05 on an already highly-optimized codebase is valued more than the same speedup on an under-optimized one. Second, if an agent memorizes and reproduces the expert patch verbatim (e.g., due to training data contamination), it receives Adv = 0 rather than spurious credit. Third, because Adv is defined per-workload before aggregation, it captures behavioral similarity between agent and expert strategies even when the surface-level code differs substantially.

## B.7. Additional Analysis

This section lists additional analysis on FORMULACODE-V that was not included in the main paper for space reasons. We analyze (1) the rate of correctness constraint violations across agent/model configurations, (2) the relationship between trajectory length and performance, (3) patterns of tool usage across configurations, (4) patch-level memorization, and (5) qualitative examples of agent patches.

**Correctness Constraint Violations.** Each FORMULACODE-V task is associated with two types of correctness constraints: (1) Snapshot tests, that verify that the optimized codebase preserves each workload's local variables, and (2) the original PyTest suite from the upstream repository which captures broader functional correctness. At initialization, the agent–model configuration receives explicit instructions to maximize performance *while preserving correctness*. If the patch fails either

*Table 16.* Correctness constraint violations by agent–model configuration. For each configuration, we report the total number of rejected solutions (out of 108), along with how many are attributable to PyTest failures versus snapshot test failures.

| Agent | Model | Total ↓ | PyTest ↓ | Snapshot ↓ |
|---|---|---|---|---|
| Terminus 2 | GPT-5 | 54 | 51 | 32 |
| | Claude 4.0 Sonnet | 55 | 52 | 36 |
| | Gemini 2.5 Pro | 55 | 53 | 30 |
| | Qwen 3 Coder | 56 | 54 | 29 |
| OpenHands | GPT-5 | 47 | 42 | 30 |
| | Claude 4.0 Sonnet | 50 | 43 | 34 |
| | Qwen 3 Coder | 50 | 44 | 32 |

constraint, we 'roll back' any performance improvements and revert to the original codebase, ensuring that all reported speedups are strictly correctness-preserving. We therefore ask: how often are candidate performance improving edits rejected solely due to correctness violations?

For each agent–model pair, we count the number of tasks in which the final patch fails at least one test, and then further break this down into PyTest failures and snapshot test failures. Table 16 summarizes these statistics over 108 attempted solutions per configuration.

*Observation: Correctness violations are common and represent a major source of rollbacks.* We find that models spend most of their budget exploring patches that ultimately fail correctness checks. On average, $\sim$48.5% of trajectories are rejected due to correctness violations, with the majority of these failures stemming from PyTest suite violations rather than snapshot test failures. We believe this to be a consequence of the multi-objective nature of the optimization problem. A single-objective setting allows verifying new functionalities with a single tool call. However, in a multi-objective setting, the agent–model configuration must strategically allocate interactions towards running either the benchmarking tool, the snapshot verification tool, or the pytest suite depending on the new functionality it introduces. The tool call distribution in Table 19 supports this hypothesis, as most agents demonstrate an inclination towards running performance validation commands rather than correctness validation commands.

**Trajectory Length and Performance.**   Discovering effective performance optimizations requires a deep understanding of the codebase. Agents must interact with the codebase through a terminal interface to obtain such an understanding. In this experiment, we study the relation between the number of interactions and the global performance achieved by the cumulative trajectory of interactions.

For each task, we record the number of complete command-line agent interactions (interactions where the agent runs a command and receives a response from the environment) and calculate the mean and median trajectory lengths averaged over all tasks. We then calculate the length-weighted advantage as $\texttt{len}(\text{Adv}_\text{agent}) = \frac{\text{Adv}_\text{agent}}{\text{len}_\text{agent}}$. Table 17 showcases these results.

*Observation: Trajectory lengths can be highly skewed.* Some configurations demonstrate highly skewed trajectories. Specifically, Terminus 2 + GPT-5 and Terminus 2 + Gemini 2.5 Pro have mean lengths substantially larger than the median length, suggesting that these configurations occasionally require very long interactive runs. By contrast, OpenHands + Claude 4.0 Sonnet has more stable trajectory lengths across tasks as the deviation between the mean and median is much smaller.

*Observation: Agent choice has a substantial effect on overall behavior.* The same model behaves very differently depending on the chosen agent. For example, GPT-5 produces much longer trajectories in Terminus 2 than in OpenHands, while Claude 4.0 Sonnet and Qwen 3 Coder show the opposite pattern. This suggests that surrounding agent design heavily shapes search behavior.

**Tool-Usage Patterns.**   In all FORMULACODE tasks, agents are given unrestricted access to the bash command line with additional performance profiling and correctness testing tools. In this experiment, we analyze how different configurations employ tools during optimization. For each task, we store the command-line interactions of the agent–model configurations and use an LLM to categorize the input commands based on the primary purpose of the command. The implementation is identical to the performance categorization classifier §A.8. We then aggregate the tool type classifications by total tool uses and tool uses per category. Table 19 summarizes these statistics across all configurations.

*Table 17.* Trajectory length and length-weighted advantage. For each agent–model configuration, we report the mean and median trajectory length (in interaction steps), as well as a length-weighted advantage ($\text{len}(\text{Adv}_{\text{agent}})$).

| Agent | Model | Mean Length ↓ | Median Length ↓ | $\text{len}(\text{Adv}_{\text{agent}})$ ↑ |
|---|---|---|---|---|
| Terminus 2 | GPT-5 | 295.53 | 198.50 | -0.000226 |
| | Claude 4.0 Sonnet | 73.13 | 63.50 | -0.000349 |
| | Gemini 2.5 Pro | 106.99 | 63.50 | -0.000755 |
| | Qwen 3 Coder | 99.91 | 90.50 | -0.000557 |
| OpenHands | GPT-5 | 68.60 | 61.00 | -0.000299 |
| | Claude 4.0 Sonnet | 222.80 | 219.50 | -0.000106 |
| | Qwen 3 Coder | 633.10 | 595.00 | -0.000044 |

*Table 18.* Tool categories used in trajectory classification. The classifier's implementation mirrors that of the optimization category classifier (§B.3).

| Category | Description |
|---|---|
| editing | Text editing or transformation commands (e.g., sed, awk, ed). |
| search | Search/discovery commands for finding files or text (e.g., grep, rg, find, fd). |
| view | Read-only inspection commands for showing file/output snippets (e.g., cat, less, head, tail). |
| fs_ops | Filesystem mutation/metadata operations (e.g., cp, mv, rm, mkdir, chmod). |
| shell_session | Shell navigation/session management commands (e.g., cd, ls, pwd, clear, exit). |
| git | Version-control commands and git-derived shell variable setup (e.g., git, diff, reset). |
| python_exec | Python execution plus Python environment/package commands (e.g., python, pip, micromamba). |
| test | Test-running commands, including snapshot checks (e.g., pytest, snapshot-tool). |
| bench | Benchmark/profiling commands, primarily ASV workflows (e.g., asv run, asv profile). |
| patching | Patch/diff application commands or diff-marker lines (e.g., patch, applypatch, −/+++). |
| other | Commands/fragments that do not match the above classes, including control-flow snippets or terminal noise. |

*Observation: Agents invoke benchmarking tools more than testing tools.* All agent–model configurations show a strong preference for running benchmarking and profiling commands over correctness validation commands, with an average of 14.90% of tool calls dedicated to benchmarking/profiling and only 2.68% of calls dedicated to testing. This proclivity towards performance validation over correctness validation might have a substantial impact on our previous observation that correctness violations are prevalent for all agent–model configurations.

*Observation: Reading dominant tool category.* The most frequently used tool category across all configurations is file-system operations (editing, searching, and viewing files), which accounts for an average of 31.74% of all tool calls. This is consistent with the intuition that developing a holistic understanding of the codebase is a prerequisite for synthesizing effective optimizations.

**Patch-Level Memorization Analysis.** A natural concern when evaluating LLM agents on tasks derived from GitHub pull requests is whether agents achieve their performance gains by memorizing expert-authored patches encountered during pre-training. To assess this, we compute the normalized edit distance between each agent-generated patch and the corresponding expert patch using Python's difflib library. A score of 0 indicates identical patches, while 1 indicates no

*Table 19.* Tool-usage statistics by agent–model configuration. Columns report the total number of tool calls and the percentage distribution of calls across tool categories (Judged by openai/gpt-oss-120b using categories in Table 18). The most effective configurations spend the majority of their tool calls on file-operations (editing, search, and view) and running performance benchmarks (bench), with the remaining calls distributed across a variety of tool categories.

| Agent | Model | Total | editing | search | view | fs_ops | shell | git | python | test | bench | patching | other |
|---|---|---|---|---|---|---|---|---|---|---|---|---|---|
| Terminus 2 | GPT-5 | 13370 | 19.40 | 15.70 | 10.29 | 2.76 | 5.00 | 12.14 | 11.20 | 2.89 | 17.51 | 2.07 | 1.02 |
| | Claude 4.0 Sonnet | 4214 | 6.12 | 8.00 | 11.25 | 7.78 | 8.19 | 0.00 | 16.61 | 2.18 | 6.36 | 0.00 | 33.51 |
| | Gemini 2.5 Pro | 5641 | 11.35 | 6.29 | 5.96 | 7.43 | 11.45 | 2.80 | 5.48 | 0.62 | 16.04 | 0.27 | 32.32 |
| | Qwen 3 Coder | 3565 | 17.59 | 16.89 | 8.61 | 12.17 | 12.45 | 0.45 | 8.72 | 1.49 | 9.99 | 0.36 | 11.28 |
| OpenHands | GPT-5 | 4683 | 14.35 | 19.65 | 26.44 | 0.62 | 1.62 | 4.36 | 7.94 | 3.93 | 18.26 | 0.04 | 2.80 |
| | Claude 4.0 Sonnet | 6323 | 12.92 | 20.94 | 26.51 | 0.76 | 1.57 | 3.56 | 9.90 | 3.67 | 16.86 | 0.03 | 3.29 |
| | Qwen 3 Coder | 8638 | 10.66 | 20.39 | 25.24 | 0.79 | 1.62 | 4.33 | 9.92 | 3.96 | 19.23 | 0.02 | 3.84 |

*Table 20.* Mean normalized edit distance between agent and expert patches on FORMULACODE-V, stratified by advantage proximity. Values close to 1 indicate low patch similarity. Across all configurations, agent and expert patches share only ∼4% of their content.

| Model | $|Adv| \leq 0.05$ | $|Adv| > 0.05$ |
|---|---|---|
| Claude 4.0 Sonnet | 0.94 | 0.97 |
| GPT-5 | 0.96 | 0.96 |
| Gemini 2.5 Pro | 0.98 | 0.95 |
| Qwen 3 Coder | 0.96 | 0.97 |

character-level similarity. We stratify results by whether the agent's advantage is close to the expert's ($|Adv| \leq 0.05$) or far ($|Adv| > 0.05$), reasoning that near-expert advantage might correlate with higher patch similarity if memorization were occurring.

*Observation: Verbatim patch memorization is rare.* Across all configurations, the mean normalized edit distance exceeds 0.94, indicating that agent-generated patches share at most ∼6% of their character-level content with expert patches. Notably, there is no meaningful difference in edit distance between tasks where agents achieve near-expert advantage and those where they do not, suggesting that similar performance outcomes arise from genuinely different code changes rather than memorization.

This analysis provides a useful signal for ruling out exact or near-exact patch-level memorization. However, it does not rule out all forms of contamination: an agent may recover the same high-level optimization strategy (e.g., "switch from a list to a set") while expressing it with substantially different code. Detecting such semantic-level overlap requires execution-based comparison, which the Adv metric already provides by measuring behavioral similarity between agent and expert solutions.

## B.8. Implications of Results

**Agent optimization strategies are qualitatively diverse.** Our stratified advantage analysis (§4.2) reveals that agents do not simply differ in overall capability—they exhibit structurally distinct optimization strategies. OpenHands + Claude 4.0 Sonnet achieves its strongest advantage at the module level ($\ell = 1$), suggesting a preference for broad architectural refactors, while OpenHands + GPT-5 excels at function-level optimizations ($\ell = 3$) but loses effectiveness at coarser granularities. These characteristic profiles are invisible to scalar leaderboard metrics and suggest that ensemble or routing strategies—where different models are deployed for different optimization scopes—could outperform any single configuration.

**Long-tail repositories expose a distribution shift bottleneck.** Agent performance trails experts on repositories in the lowest popularity quintile (Q1; 133–202 GitHub stars), even though expert patches in this regime yield the second-largest speedups (§4.4). This asymmetry suggests that the primary bottleneck is not task difficulty but rather distribution shift: agents are less familiar with the conventions, APIs, and idioms of niche repositories. This finding has direct implications for practical deployment—agents are less helpful on precisely the repositories where human developers may need them most, in under-maintained and under-documented codebases.

**Multi-workload optimization reveals real-world tradeoffs.** Expert solutions achieve the best global speedup despite causing an average regression on some workloads (§4.5.2). This tradeoff pattern—improving many execution paths at the cost of modest regressions on others—mirrors real-world optimization practice, where universal improvement across all inputs is rarely achievable (Balsamo et al., 2004; Jin et al., 2012). Single-workload benchmarks cannot detect such tradeoffs. FORMULACODE's multi-workload design (averaging ∼264.58 workloads per task) provides the resolution needed to study how agents negotiate competing objectives, an aspect critical to understanding their real-world utility.

**Tool usage patterns correlate with optimization success.** Our tool-usage analysis (§B.7) shows that all agent–model configurations preferentially invoke benchmarking and profiling tools over correctness testing tools. Combined with our finding that ∼48.5% of candidate solutions are rejected due to correctness violations, this suggests a systematic allocation failure: agents invest disproportionately in performance validation at the expense of verifying functional correctness. Future agent designs that better balance performance profiling with correctness checking may substantially reduce wasted computation.

## B.9. Qualitative Examples: Human Expert vs. AI Agent Patches

This section presents side-by-side comparisons of human expert and AI agent patches for FORMULACODE tasks. Specifically, the following examples are showcased:

- Figure 20 (modin_modin_2). **Failure mode: Incorrect triage; expert gained edge by identifying performance hotpath.** Modin has an expensive auto-switch backend logic that was being called even when all inputs shared the same backend. The agent was not able to identify the core issue, instead focusing on a caching issue that was not on the performance critical path. The human correctly identifies the issue and implemented a fix to the caching logic.
- Figure 21 (optuna_optuna_4020). **Failure mode: Correct triage; expert gained edge by numpy vectorization delegation.** Optuna's hypervolume computation used a naive recursive algorithm, when a faster $O(N^2)$ approach was possible. Both the human and the agent were able to identify and implement the algorithm. However, the human's solution used fully vectorized numpy operations, while the agent's solution used a Python-level sweep-line approach with bisect. This resulted in the human outperforming the agent despite both having the same asymptotic complexity.
- Figure 22 (optuna_optuna_3647). **Failure mode: Correct triage; expert implemented holistic full-module optimization.** Optuna's implementation for sorting non-dominated Pareto fronts used a naive algorithm that didn't scale well as number of trials increased. Both the human and the agent identified this issue; the agent's implementation utilized a Fenwick tree based algorithm which fixed a single hotpath (when inputs are 2D). However, the expert implementation implemented a holistic rewrite: it optimized the entire call chain to use vectorized numpy operations and merged separate pathways for 2D/N-D optimization, resulting in complementary improvements across the entire multi-objective optimization flow.
- Figure 23 (networkx_networkx_7424). The core issue was that NetworkX's BFS-based component discovery algorithm did not implement an early-termination optimization. Both the human and the agent fix this by implementing an early termination optimization. However, the agent outperforms the human by further optimizing the BFS implementation, achieving an additional $+0.0132$ advantage on top of the human's improvement.
- Figure 24 (pybamm_team_pybamm_1). A sensitivity computation in PyBaMM created a quadratic memory allocation bottleneck due to incremental concatenation without realizing that the full size was known in advance. Both the human and the agent identify the issue and collect all blocks first and concatenate once. The agent further optimizes the concatenation logic by consolidating multiple function calls into one and adding guards for empty inputs, resulting in a $+0.0167$ advantage.
- Figure 25 (shapely_shapely_1982). The deprecate_positional decorator in Shapely called inspect.signature on every invocation, causing 300–1000% slowdowns. The human and agent solution both converged on the same strategy: implementing a caching layer on the decorator. However, the agent implemented additional optimizations to skip the hot-path when no deprecated parameters existed, resulting in a $+0.0132$ advantage.

**Human Expert Patch**

```
diff --git a/modin/core/storage_formats/pandas/query_compiler_caster.py
--- a/modin/core/storage_formats/pandas/query_compiler_caster.py
+++ b/modin/core/storage_formats/pandas/query_compiler_caster.py
@@ -1006,6 +1006,8 @@
        pin_target_backend = None
+       input_backends: set[str] = set()

        def register_query_compilers(arg):
+               input_backends.add(arg_backend)

@@ -1047,10 +1050,12 @@
-       inputs_pinned = (
-           len(input_query_compilers) < 2 and pin_target_backend is not
     None
-       )
-       if not AutoSwitchBackend.get() or inputs_pinned:
        # Skip casting when all inputs share a single backend and either
        # auto-switching is disabled or inputs are pinned
+       if len(input_backends) == 1 and (
+           not AutoSwitchBackend.get() or pin_target_backend is not None
+       ):
            f_to_apply = _get_extension_for_method(...)
            result = f_to_apply(*args, **kwargs)
-           if isinstance(result, QueryCompilerCaster) and inputs_pinned:
+           if (isinstance(result, QueryCompilerCaster)
+                   and pin_target_backend is not None):
                result._set_backend_pinned(True, inplace=True)
            return result
```

**AI Agent Patch**

```
diff --git a/modin/core/storage_formats/pandas/query_compiler_caster.py
--- a/modin/core/storage_formats/pandas/query_compiler_caster.py
+++ b/modin/core/storage_formats/pandas/query_compiler_caster.py
@@ -137,23 +137,52 @@
-def _get_empty_qc_for_default_backend() -> BaseQueryCompiler:
-    """Get an empty query compiler for the default backend."""
+def _get_empty_qc_for_backend(backend: str) -> BaseQueryCompiler:
+    """Get an empty query compiler for the specified backend."""
    from modin.core.execution.dispatching.factories.dispatcher import
     FactoryDispatcher
-    return FactoryDispatcher.get_factory().io_cls.from_pandas(pandas.
     DataFrame())
+    original_backend = Backend.get()
+    try:
+        Backend.put(backend)
+        return FactoryDispatcher.get_factory().io_cls.from_pandas(pandas.
     DataFrame())
+    finally:
+        Backend.put(original_backend)

-_BACKEND_TO_EMPTY_QC: defaultdict[str, BaseQueryCompiler] = defaultdict(
-    _get_empty_qc_for_default_backend
-)
+_BACKEND_TO_EMPTY_QC: dict[str, BaseQueryCompiler] = {}

+def _get_cached_empty_qc(backend: str) -> BaseQueryCompiler:
+    if backend not in _BACKEND_TO_EMPTY_QC:
+        _BACKEND_TO_EMPTY_QC[backend] = _get_empty_qc_for_backend(backend)
+    return _BACKEND_TO_EMPTY_QC[backend]

@@ -1042,7 +1071,7 @@
-           input_qc_for_pre_op_switch = _BACKEND_TO_EMPTY_QC[
     input_backend]
+           input_qc_for_pre_op_switch = _get_cached_empty_qc(
     input_backend)
```

**Summary**

Disables `AutoSwitchBackend` by default and rewrites the casting-skip logic in `query_compiler_caster.py` to track the set of distinct input backends, skipping expensive query-compiler conversions when all inputs share a single backend. Updates four test files and adjusts metric assertions (not shown).

**Summary**

Fixes a bug where the `defaultdict` factory ignores the requested backend when creating empty query compilers, replacing it with an explicit `_get_cached_empty_qc` function that temporarily switches `Backend.put()` to the correct backend. A correctness fix, but not on the performance-critical path.

*Figure 20.* `modin_project-modin_7637`: Modin's `AutoSwitchBackend` feature, enabled by default, triggered an expensive type conversion even when all inputs shared the same backend. The agent solution (`openhands:claude-sonnet-4`) identified and fixed a real bug in the caching logic, but this was not on the performance-critical path, resulting in a $-0.1265$ advantage compared to the human expert's systemic fix that disabled `AutoSwitchBackend` by default and optimized the casting logic to track input backend diversity, skipping conversions when unnecessary.

## Human Expert Patch

```
diff --git a/optuna/_hypervolume/wfg.py b/optuna/_hypervolume/wfg.py
--- a/optuna/_hypervolume/wfg.py
+++ b/optuna/_hypervolume/wfg.py
# New O(N^2) vectorized 3D hypervolume via coordinate compression
+def _compress_coordinate(coords: np.ndarray) -> tuple[np.ndarray, np.
    ndarray]:
+    sorted_indices = np.argsort(coords)
+    values = coords[sorted_indices]
+    r = np.zeros_like(sorted_indices)
+    r[sorted_indices] = np.arange(coords.shape[0], dtype=r.dtype)
+    return r, values

+def _compute_3d(sorted_pareto_sols: np.ndarray, reference_point: np.ndarray
    ) -> float:
+    """O(N^2) 3D hypervolume via cumulative minimum on compressed
      coordinates."""
+    n = sorted_pareto_sols.shape[0]
+    x_vals = np.concatenate([sorted_pareto_sols[:, 0], reference_point
      [:1]])
+    y_ind, y_vals = _compress_coordinate(sorted_pareto_sols[:, 1])
+    y_vals = np.concatenate([y_vals, reference_point[1:2]])
+    z_max = reference_point[2]
+    zs = np.full((n, n), z_max)
+    for i in range(n):
+        zs[i, y_ind[i]] = sorted_pareto_sols[i, 2]
+    zs = np.minimum.accumulate(zs, axis=0)
+    zs = np.minimum.accumulate(zs, axis=1)
+    x_delta = x_vals[1:] - x_vals[:-1]
+    y_delta = y_vals[1:] - y_vals[:-1]
+    return np.sum((z_max - zs) * x_delta[:, np.newaxis] * y_delta[np.
      newaxis, :])

@@ -125,6 +178,13 @@ def compute_hypervolume(
     if reference_point.shape[0] == 2:
         hv = _compute_2d(sorted_pareto_sols, reference_point)
+    elif reference_point.shape[0] == 3:
+        hv = _compute_3d(sorted_pareto_sols, reference_point)
     else:
         hv = _compute_hv(sorted_pareto_sols, reference_point)
```

## AI Agent Patch

```
diff --git a/optuna/_hypervolume/wfg.py b/optuna/_hypervolume/wfg.py
--- a/optuna/_hypervolume/wfg.py
+++ b/optuna/_hypervolume/wfg.py
# x-sweep with incremental 2D Pareto frontier via bisect
+def _compute_3d(sorted_pareto_sols, reference_point) -> float:
+    """Exact 3D hypervolume using x-sweep with incremental 2D frontier."""
+    import bisect
+    xs = pts[:, 0]
+    dx = np.maximum(xs_ext[1:] - xs_ext[:-1], 0.0)
+    y_list: list[float] = []
+    z_list: list[float] = []
+
+    def insert_frontier(y: float, z: float) -> None:
+        i = bisect.bisect_left(y_list, float(y))
+        if i > 0 and z >= z_list[i - 1]:
+            return  # dominated by left neighbor
         # ... (dominance-aware insertion: handle equal y,
         #      remove dominated points to the right)
+        y_list.insert(i, float(y))
+        z_list.insert(i, float(z))
+
+    for i in range(n):
+        insert_frontier(float(pts[i, 1]), float(pts[i, 2]))
+        if y_list:
+            yz = np.column_stack((np.asarray(y_list), np.asarray(z_list)))
+            areas[i] = _compute_2d(yz, ref_yz)
+    return float(np.dot(dx, areas))

@@ -126,7 +190,7 @@ def compute_hypervolume(
-        hv = _compute_hv(sorted_pareto_sols, reference_point)
+        hv = _compute_3d(...) if sorted_pareto_sols.shape[1] == 3 else
          _compute_hv(...)
```

### Summary

Adds a specialized $O(N^2)$ `_compute_3d` function using a `_compress_coordinate` helper that maps $y$-coordinates to integer ranks via `np.argsort`, builds an $N \times N$ grid, and applies `np.minimum.accumulate` along both axes to compute dominated volume in fully vectorized numpy. Also adds a dedicated `elif` branch in `compute_hypervolume` and parameterized tests (not shown).

### Summary

Adds a `_compute_3d` function using an $x$-sweep with incremental 2D Pareto frontier maintenance via `bisect` and Python lists. At each $x$-slice, the frontier is updated with dominance-aware insertion, then the 2D area is computed by delegating to `_compute_2d`. The dispatch in `compute_hypervolume` is modified with an inline ternary for 3D inputs.

*Figure 21.* optuna_optuna_4020: Optuna's `_hypervolume.WFG` class used a naive recursive algorithm for hypervolume computation that had a $O(N^3)$ runtime for the common 3D case, when a $O(N^2)$ approach was possible. Both the human and the agent identified and implemented the faster algorithm. However, the human's solution used fully vectorized numpy operations, while the best agent (`terminus-2:gpt-5`) used a Python-level sweep-line approach with `bisect`. This resulted in the human outperforming the agent with a $-0.03964$ agent advantage despite both having the same asymptotic complexity.

**Human Expert Patch**

```
diff --git a/optuna/study/_multi_objective.py b/optuna/study/
    _multi_objective.py
--- a/optuna/study/_multi_objective.py
+++ b/optuna/study/_multi_objective.py
@@ (selected excerpts)
-def _get_pareto_front_trials_2d(...):
-    ...  # Separate 2D implementation
-def _get_pareto_front_trials_nd(...):
-    ...  # Separate N-D implementation
-def _get_pareto_front_trials_by_trials(...):
-    if len(directions) == 2:
-        return _get_pareto_front_trials_2d(...)
-    return _get_pareto_front_trials_nd(...)
+def _get_pareto_front_trials_by_trials(...):
+    loss_values = np.asarray(...)
+    on_front = _is_pareto_front(loss_values,
+        assume_unique_lexsorted=False)
+    return [t for t, p in zip(trials, on_front) if p]

-def _fast_non_dominated_sort(
-    objective_values, *, penalty=None, n_below=None
+def _fast_non_domination_rank(
+    loss_values, *, penalty=None, n_below=None
 ) -> np.ndarray:
-    ...  # O(n^2) broadcast + defaultdict
+    ...  # Vectorized _calculate_nondomination_rank
+    ...  # + _is_pareto_front with lexsort
```

**AI Agent Patch**

```
diff --git a/optuna/study/_multi_objective.py b/optuna/study/
    _multi_objective.py
--- a/optuna/study/_multi_objective.py
+++ b/optuna/study/_multi_objective.py
@@ -189,42 +189,106 @@
 def _calculate_nondomination_rank(...):
     ...
     # Fast path for 2D objectives.
+    if objective_values.shape[1] == 2:
+        x = objective_values[:, 0]
+        y = objective_values[:, 1]
+        order = np.lexsort((y, x))
+        ys_unique = np.unique(y)
+        y_idx_all = np.searchsorted(ys_unique, y,
+            side='right')
+        m = len(ys_unique)
+        bit = np.zeros(m + 1, dtype=int)
+        def bit_query(i):  # Fenwick tree prefix max
+            ...
+        def bit_update(i, v):
+            ...
+        # Process equal-x groups, BIT for rank
+        ...
+        return ranks, last_rank
+
     # Fallback: original O(n^2) broadcast for >=3D.
     domination_mat = np.all(...) & np.any(...)
```

**Summary**

Complete rewrite of `_multi_objective.py`. Renames `_fast_non_dominated_sort` to `_fast_non_domination_rank`, replaces the $O(n^2)$ broadcast-based algorithm with a vectorized `_is_pareto_front` and `_calculate_nondomination_rank` implementation, merges the separate 2D/N-D Pareto front functions, and updates all callers across the TPE sampler and NSGA-II selection strategy.

**Summary**

Adds a specialized $O(n \log n)$ BIT (Fenwick tree) algorithm for 2D objectives in `_calculate_nondomination_rank`, falling back to the original $O(n^2)$ broadcast for $\geq 3$ objectives. While algorithmically superior for the 2D case, the agent only optimizes the inner ranking function without restructuring callers or the Pareto front computation.

*Figure 22.* `optuna_optuna_3647`: The original implementation of Optuna's non-dominated sorting in multi-objective optimization cases emerged as a performance bottleneck when scaling to large number of trials ($\sim 10000$ trials). Both the best agent (`terminus-2:gpt-5`) and the human expert correctly identified the issue. The agent's solution focused on optimizing the inner ranking function with a specialized $O(n \log n)$ Fenwick-tree algorithm for 2D objectives, while retaining the original $O(n^2)$ broadcast-based approach for higher dimensions. In contrast, the human expert implemented a holistic rewrite of the entire call chain to use vectorized numpy operations and merged separate pathways for 2D/N-D optimization. This broader vectorized rewrite captured multi-objective speedups ($5$–$6\times$) that the agent's localized change missed, resulting in a $-0.03629$ agent advantage.

**Human Expert Patch**

```diff
diff --git a/networkx/algorithms/components/connected.py
@@ -65,7 +65,7 @@ def connected_components(G):
     for v in G:
         if v not in seen:
-            c = _plain_bfs(G, n, v)
+            c = _plain_bfs(G, n - len(seen), v)
             seen.update(c)

diff --git a/networkx/algorithms/components/weakly_connected.py
@@ -62,7 +62,7 @@ def weakly_connected_components(G):
     for v in G:
         if v not in seen:
-            c = set(_plain_bfs(G, n, v))
+            c = set(_plain_bfs(G, n - len(seen), v))
             seen.update(c)
```

**AI Agent Patch**

```diff
diff --git a/networkx/algorithms/components/connected.py b/networkx/
    algorithms/components/connected.py
index ebe0d8c1..922d4904 100644
--- a/networkx/algorithms/components/connected.py
+++ b/networkx/algorithms/components/connected.py
@@ -62,12 +62,15 @@ def connected_components(G):
-    [-n-]{+remaining+} = len(G)
     ...
-            c = _plain_bfs(G, [-n-]{+remaining+}, v)
             seen.update(c)
+            remaining -= len(c)
             yield c
+            if remaining == 0:
+                break
@@ -203,14 +206,16 @@ def _plain_bfs(G, n, source):
+    seen_add = seen.add
     while nextlevel:
         thislevel = nextlevel
         nextlevel = []
+        append = nextlevel.append
         ...
             if w not in seen:
-                [-seen.add(w)-]{+seen_add(w)+}
-                [-nextlevel.append(w)-]{+append(w)+}
diff --git a/networkx/algorithms/components/weakly_connected.py b/networkx/
    algorithms/components/weakly_connected.py
index ecfac50a..a89b7af8 100644
--- a/networkx/algorithms/components/weakly_connected.py
+++ b/networkx/algorithms/components/weakly_connected.py
@@ -59,12 +59,15 @@ def weakly_connected_components(G):
     # (same early-exit optimization as connected_components above)
@@ -166,32 +169,30 @@ def _plain_bfs(G, n, source):
     # (same local-variable caching as connected._plain_bfs above)
     # additionally, converted from generator (yield) to returning seen set:

-    yield source
+    ...
             if len(seen) == n:
-                return
+                return seen
+    return seen
```

**Summary**

Minimal single-line fix in both `connected_components` and `weakly_connected_components`: passes `n - len(seen)` instead of `n` to `_plain_bfs`, tightening the BFS early-termination bound so it stops as soon as all remaining unseen nodes are found. No structural changes to the BFS itself.

**Summary**

Multi-pronged optimization: tracks a `remaining` node count to break out of the component loop early, caches method lookups (`seen.add`, `nextlevel.append`) into local variables, and converts the weakly-connected `_plain_bfs` from a generator to a batch set return, eliminating per-node yield overhead.

*Figure 23.* `networkx_networkx_7424`: NetworkX's `connected_components` and `weakly_connected_components` passed the total graph node count n to `_plain_bfs` without accounting for already-discovered nodes, missing an early-termination optimization. For disconnected graphs with large components explored last, this caused dramatic slowdowns—up to $367\times$ for adversarial cases with $n=1000$. Both the best agent (`openhands:gpt-5`) and the expert identified the core issue, and implemented the same early-termination optimization. However, the agent also implemented additional micro-optimizations that further reduced overhead, resulting in a $+0.0132$ advantage over the human's solution.

**Human Expert Patch**

```diff
diff --git a/src/pybamm/solvers/processed_variable.py b/...
--- a/src/pybamm/solvers/processed_variable.py
+++ b/src/pybamm/solvers/processed_variable.py
@@ -443,16 +443,18 @@ class ProcessedVariable:
             dvar_dp_func = casadi.Function(
                 "dvar_dp", [t_casadi, y_casadi, p_casadi_stacked], [
         dvar_dp]
             )
-            for idx, t in enumerate(ts):
-                u = ys[:, idx]
-                next_dvar_dy_eval = dvar_dy_func(t, u, inputs_stacked)
-                next_dvar_dp_eval = dvar_dp_func(t, u, inputs_stacked)
-                if idx == 0:
-                    dvar_dy_eval = next_dvar_dy_eval
-                    dvar_dp_eval = next_dvar_dp_eval
-                else:
-                    dvar_dy_eval = casadi.diagcat(dvar_dy_eval,
        next_dvar_dy_eval)
-                    dvar_dp_eval = casadi.vertcat(dvar_dp_eval,
        next_dvar_dp_eval)
+            dvar_dy_eval = casadi.diagcat(
+                *[
+                    dvar_dy_func(t, ys[:, idx], inputs_stacked)
+                    for idx, t in enumerate(ts)
+                ]
+            )
+            dvar_dp_eval = casadi.vertcat(
+                *[
+                    dvar_dp_func(t, ys[:, idx], inputs_stacked)
+                    for idx, t in enumerate(ts)
+                ]
+            )

             # Compute sensitivity
             S_var = dvar_dy_eval @ dy_dp + dvar_dp_eval
```

**AI Agent Patch**

```diff
diff --git a/src/pybamm/solvers/processed_variable.py b/...
--- a/src/pybamm/solvers/processed_variable.py
+++ b/src/pybamm/solvers/processed_variable.py
@@ -436,29 +439,30 @@ class ProcessedVariable:
             dvar_dy = casadi.jacobian(var_casadi, y_casadi)
             dvar_dp = casadi.jacobian(var_casadi, p_casadi_stacked)

-            dvar_dy_func = casadi.Function(
-                "dvar_dy", [t_casadi, y_casadi, p_casadi_stacked], [dvar_dy
        ]
-            )
-            dvar_dp_func = casadi.Function(
-                "dvar_dp", [t_casadi, y_casadi, p_casadi_stacked], [
        dvar_dp]
            # Single function returning both jacobians
+            grads_func = casadi.Function(
+                "pv_grads", [t_casadi, y_casadi, p_casadi_stacked],
+                [dvar_dy, dvar_dp]
+            )
-            for idx, t in enumerate(ts):
+
+            dvar_dy_blocks = []
+            dvar_dp_blocks = []
+            for idx in range(ts.size):
+                t = ts[idx]
                u = ys[:, idx]
-                next_dvar_dy_eval = dvar_dy_func(t, u, inputs_stacked)
-                next_dvar_dp_eval = dvar_dp_func(t, u, inputs_stacked)
-                if idx == 0:
-                    dvar_dy_eval = next_dvar_dy_eval
-                    dvar_dp_eval = next_dvar_dp_eval
-                else:
-                    dvar_dy_eval = casadi.diagcat(dvar_dy_eval,
        next_dvar_dy_eval)
-                    dvar_dp_eval = casadi.vertcat(dvar_dp_eval,
        next_dvar_dp_eval)
+                g_dy, g_dp = grads_func(t, u, inputs_stacked)
+                dvar_dy_blocks.append(g_dy)
+                dvar_dp_blocks.append(g_dp)
+
                # Concatenation in one shot
+            dvar_dy_eval = casadi.diagcat(*dvar_dy_blocks)
+            dvar_dp_eval = casadi.vertcat(*dvar_dp_blocks)

             # Compute sensitivity
             S_var = dvar_dy_eval @ dy_dp + dvar_dp_eval
```

**Summary**

Replaced the incremental per-timestep `casadi.diagcat`/`casadi.vertcat` loop with list comprehensions that build all Jacobian blocks first, then concatenate once via unpacking (`*blocks`). Also added a `CHANGELOG.md` entry (not shown).

**Summary**

Consolidated the two separate `casadi.Function` objects (`dvar_dy_func`, `dvar_dp_func`) into a single `grads_func` returning both Jacobians, reducing per-timestep function call overhead. Collects results in lists and concatenates once. Also adds guards for empty time series and empty result lists.

*Figure 24.* `pybamm_team-pybamm_4735`: PyBaMM's `ProcessedVariable` sensitivity computation in `IDAKLUSolver` used an incremental per-timestep concatenation operation, creating a quadratic memory allocation overhead. Both the best agent (`openhands:gpt-5`) and the expert identified that, instead of each loop iteration building a progressively larger matrix by concatenating to the existing result, it would be more efficient to first collect all blocks and then concatenate once at the end. The agent added further micro-optimization: consolidating two accumulation function calls into one and added empty-input guards. This resulted in a +0.0167 agent advantage.

**Human Expert Patch**

```
diff --git a/shapely/decorators.py b/shapely/decorators.py
--- a/shapely/decorators.py
+++ b/shapely/decorators.py
-def deprecate_positional(should_be_kwargs, category=DeprecationWarning):
+def deprecate_positional(
+    should_be_kwargs: Iterable[str],
+    category: type[Warning] = DeprecationWarning,
+):
+    def decorator(func: Callable):
+        code = unwrap(func).__code__
+        pos_names = code.co_varnames[: code.co_argcount]
+        name_to_idx = {name: idx for idx, name in enumerate(pos_names)}
+        deprecate_positions = [
+            (name_to_idx[name], name) for name in should_be_kwargs
+            if name in name_to_idx]
+        if not deprecate_positions:
+            return func
+        warn_from = min(deprecate_positions)[0]
+
+        @lru_cache(10)
+        def make_msg(n_args: int):
+            used = [name for idx, name in deprecate_positions if idx <
+    n_args]
+            # ... (build warning message string)
+            return f"positional argument{plr} {args_txt} for '{func.
+    __name__}' ..."
+
-    def decorator(func):
        @wraps(func)
        def wrapper(*args, **kwargs):
-            ret_value = func(*args, **kwargs)
-            sig = inspect.signature(func)
-            args_bind = sig.bind_partial(*args)
-            warn_args = [f"'{arg}'" for arg in args_bind.arguments.keys()
-                            if arg in should_be_kwargs]
-            if warn_args:
-                # ... (build warning message)
-                warnings.warn(msg, category=category, stacklevel=2)
-            return ret_value
+            result = func(*args, **kwargs)
+            n = len(args)
+            if n > warn_from:
+                warnings.warn(make_msg(n), category=category, stacklevel=2)
+
+            return result
        return wrapper
```

**AI Agent Patch**

```
diff --git a/shapely/decorators.py b/shapely/decorators.py
--- a/shapely/decorators.py
+++ b/shapely/decorators.py
    def decorator(func):
        # Cache signature inspection at decoration time
+        sig = inspect.signature(func)
+        param_names = list(sig.parameters.keys())
+        deprecated_positions = set()
+        for param_name in should_be_kwargs:
+            if param_name in param_names:
+                deprecated_positions.add(param_names.index(param_name))
+        if not deprecated_positions:
+            return func
+        max_deprecated_pos = max(deprecated_positions)

        @wraps(func)
        def wrapper(*args, **kwargs):
-            ret_value = func(*args, **kwargs)
-            sig = inspect.signature(func)
-            args_bind = sig.bind_partial(*args)
            # ... (per-call signature inspection)
            # Fast path: skip if not enough args
+            if len(args) <= max_deprecated_pos:
+                return func(*args, **kwargs)
            # Only check deprecated positions
+            warn_positions = [pos for pos in deprecated_positions if pos <
+    len(args)]
+            if warn_positions:
+                args_bind = sig.bind_partial(*args)
                # ... (build and emit warning)
+            return func(*args, **kwargs)
        return wrapper
```

**Summary**

Completely rewrote the `deprecate_positional` decorator: replaced `inspect.signature` with `inspect.unwrap` and direct `__code__` introspection at decoration time, added an `lru_cache`-backed `make_msg` helper to avoid rebuilding warning strings, and included type annotations and a comprehensive 138-line test suite.

**Summary**

Cached `inspect.signature` at decoration time and precomputed deprecated parameter positions as a set. Added an early-return fast path when no deprecated parameters exist and a second fast path skipping checking when argument count is below the threshold.

*Figure 25.* `shapely_shapely_1982`: The `deprecate_positional` decorator in Shapely called `inspect.signature` and `sig.bind_partial` on every decorated function invocation, causing a 300–1000% performance regression. Users reported significant Polygon creation slowdowns. The best agent (`terminus-2:claude-sonnet-4`) and the human expert converged on nearly identical core strategies. Both implemented a caching layer to move signature inspection from call time to decoration time. The agent added additional micro-optimizations to skip checks when no deprecated parameters exist or when the argument count is below the threshold. This resulted in a $+0.0131$ advantage over the human's solution.

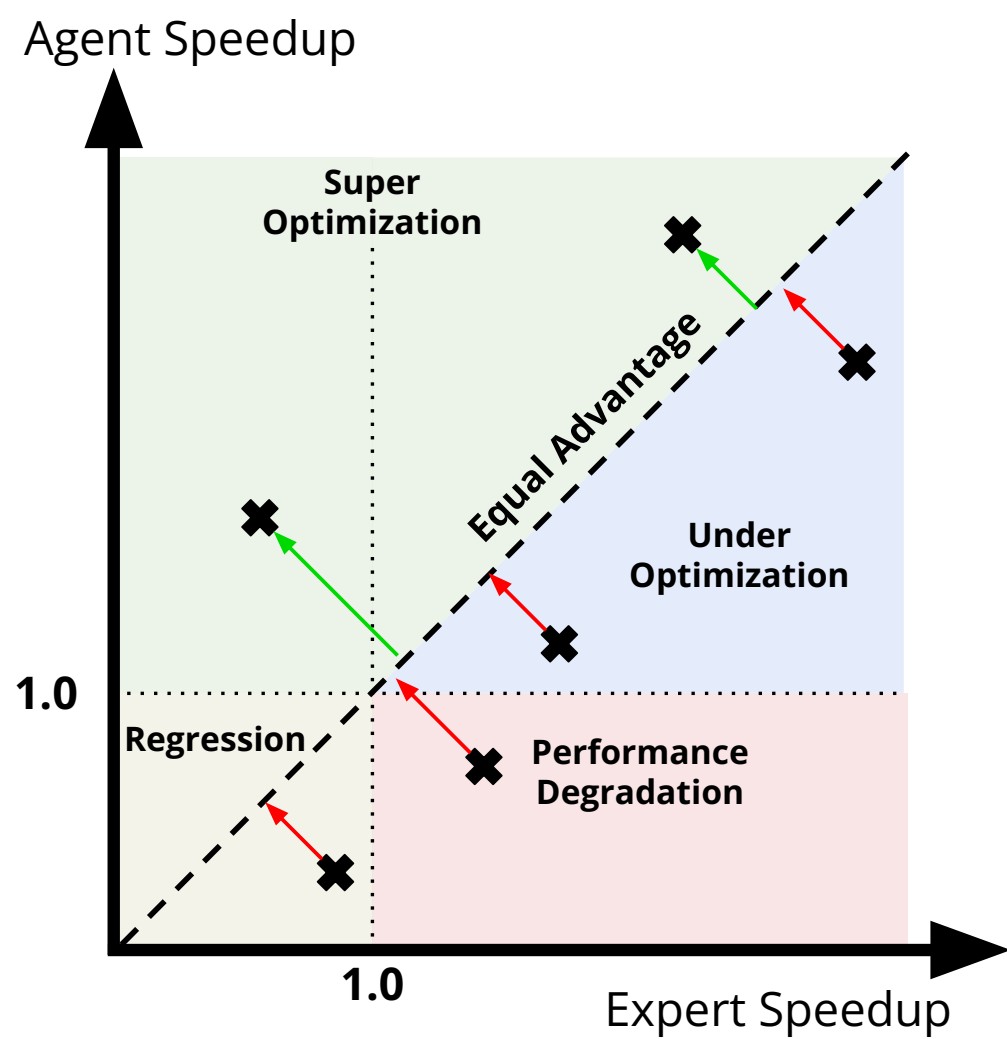

*Figure 26.* Visual intuition for Agent Advantage ($Adv_{agent}$; §2). Each cross (✗) represents an individual workload using the expert-derived speedup ($speedup_{expert}$) and the agent-derived speedup ($speedup_{agent}$). The identity function line represents *equal advantage* (i.e., $speedup_{expert} = speedup_{agent}$). Then, the agent advantage is the mean weighted deviation from the equal advantage line. The plot also showcases four optimization regions clockwise from top: (1) *Super Optimization*: workloads where an agent's code performs better than the expert's code and the baseline. (2) *Under Optimization*: workloads where the agent's code and the expert's code both deliver a positive speedup, but the expert outperforms the agent. (3) *Performance Degredation*: workloads where the expert discovers a speedup while the agent slows down the code. (4) *Regression*: workloads where neither the expert nor the agent slow down the code; usually an intentional tradeoff to optimize other workloads. Figure 26 showcases an example of workload distribution for various agents on FORMULACODE.

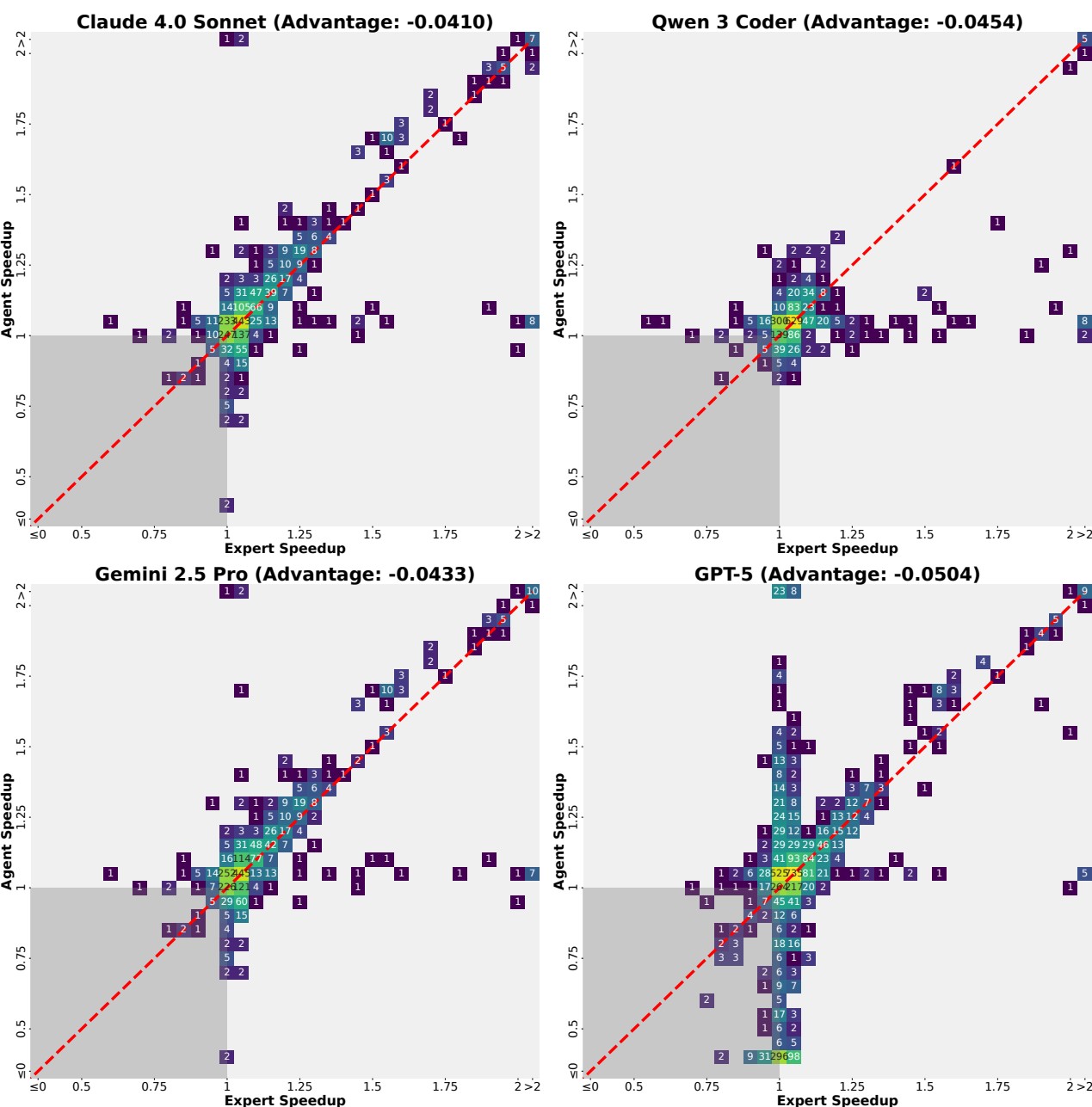

*Figure 27.* Visualization of advantage for Terminus 2 Agents. Refer to Figure 26 for an explanation of each region. Each square represents the number of workloads in that region (within 0.5 units). A speedup of 1.0 indicates no deviation from baseline performance. The red dotted line represents equal advantage. This visualization is helpful to gauge the *holistic* behavior of models across the entire workload distribution. For instance, Claude 4.0 Sonnet (Top Left) achieves a better overall advantage than GPT-5 (Bottom Right) by making measured and surgical optimizations that align with the equal-advantage line, whereas optimizations proposed by GPT-5 are more volatile, with more workloads experiencing performance degradations and effectively bringing the overall advantage down.

