# OpenReview forum: "FormulaCode: Evaluating Agentic Optimization on Large Codebases"
_ICML.cc/2026/Conference — ICML 2026 regular_

### Official Review · Reviewer_apsu · 2026-03-07

**Soundness:** 3
**Presentation:** 3
**Significance:** 3
**Originality:** 2
**Overall Recommendation:** 4
**Confidence:** 4

**Summary:**

This paper introduces FORMULACODE, a benchmark designed to evaluate autonomous coding agents on performance optimization tasks in real-world repositories. The benchmark is constructed by mining performance-improving pull requests from scientific Python projects on GitHub and pairing them with their associated ASV benchmark suites. Each task requires an agent to modify a repository to improve performance across multiple workloads while maintaining functional correctness. The benchmark therefore evaluates optimization under multi-workload constraints rather than a single target function.

The dataset contains 957 performance optimization tasks derived from real pull requests, each associated with a set of community-maintained performance workloads. The paper also proposes evaluation metrics based on geometric-mean speedup across workloads, worst regression, and an “advantage” measure that compares agent performance to the expert-authored patch. Experiments evaluate several agent frameworks combined with different LLM backbones to assess the current capability of LLM-based coding agents on repository-scale performance optimization.

**Compliance With Llm Reviewing Policy:**

Affirmed.

**Final Justification:**

The authors’ rebuttal and clarifications strengthen the paper in several important ways. In particular, the additional discussion makes a more convincing case for the role of multi-workload evaluation in exposing optimization trade-offs that are not visible in prior single-workload benchmarks, and the added analyses improve confidence in the empirical methodology.

The novelty remains somewhat borderline in the sense that the contribution can still be viewed as a strong extension of prior benchmarking efforts rather than an entirely new paradigm. However, the rebuttal substantially clarifies why the proposed benchmark enables analyses that prior single-workload benchmarks cannot support. This strengthens the case that the work makes a meaningful contribution beyond simple scaling.

The concern regarding the advantage metric is also partially addressed. The rebuttal provides both empirical and approximate theoretical support for the metric through log-space comparisons and the Taylor approximation argument. While a fully principled justification is still not entirely established, the added evidence improves confidence that the reported conclusions are not an artifact of the specific metric choice.

Overall, the work appears technically sound, addresses an important problem, and offers a benchmark that is likely to be useful to the community. Taking both the paper and the rebuttal into account, the strengths now outweigh the remaining weaknesses.

Based on the above, I update my recommendation to Weak Accept.

**Key Questions For Authors:**

1. The main experiments are conducted on the subset FORMULACODE-V due to computational constraints. Could the authors clarify how large this subset is and how it was sampled from the full benchmark? In particular, was the sampling stratified by repository, workload count, or PR characteristics? Clarifying this would help assess whether the reported results generalize to the full benchmark.

2. The dataset construction process filters pull requests using Mann–Whitney U tests to ensure statistically significant performance improvements. How many benchmark repetitions are used per workload when performing this test, and is any correction applied for multiple comparisons across workloads within a task? Additional details would help clarify the statistical robustness of the filtering procedure.

3. The paper defines the “advantage” metric as the difference between the agent speedup and the expert patch speedup. Since speedup is multiplicative and aggregated using geometric means, could the authors comment on why a direct difference is used rather than a comparison in log-space? It would also be helpful to know whether the reported conclusions are sensitive to alternative aggregation strategies (e.g., median, trimmed mean, or log-space comparisons).

4. The experiments compare different agent frameworks and LLM backbones. Could the authors clarify whether tool budgets, token limits, iteration caps, and stopping criteria are controlled consistently across frameworks? This would help ensure that the comparison reflects differences in model capability rather than differences in orchestration design.

**Limitations:**

The paper includes only a minimal impact statement, and the discussion of limitations and potential societal impacts is not yet adequate. The current statement says that the goal is to advance machine learning and that there are many potential societal consequences, but none are specifically highlighted. This is too brief to meaningfully address the limitations or possible downstream risks of the work.

A stronger discussion would briefly acknowledge the main limitations of the benchmark, such as its focus on scientific Python repositories, dependence on ASV-based workloads, and evaluation on a subset of the full benchmark due to computational constraints. It would also be useful to mention possible broader impacts, for example the potential dual-use nature of increasingly capable coding agents for large-scale software modification, including misuse for optimizing or maintaining harmful codebases. Even a short, concrete paragraph covering these points would be sufficient.

**Strengths And Weaknesses:**

### Strengths

- Addresses an important and relatively underexplored evaluation setting for code agents: **repository-level performance optimization**, rather than only functional correctness or bug fixing.
- The benchmark is constructed from **real GitHub pull requests** paired with **community-maintained ASV benchmark suites**, which grounds the evaluation in realistic software engineering workflows.
- The dataset is relatively large (**957 tasks across multiple repositories**), suggesting substantial engineering effort in mining, constructing, and validating the benchmark tasks.
- Evaluating optimization across **multiple workloads per task** is a meaningful design choice, since it better reflects real-world performance trade-offs than benchmarks that optimize a single function or metric.
- The comparison against **expert-authored PR patches** provides a useful reference point and helps contextualize the current capabilities of LLM-based agents.
- The paper makes a visible effort toward **reproducibility**, including Dockerized environments and an automated dataset construction pipeline.

### Weaknesses

- The **novelty relative to prior benchmarks is not fully clarified**. Repository-level code tasks already exist (e.g., SWE-Bench variants), and performance optimization benchmarks such as GSO-Bench and SWEfficiency have been introduced. The main distinguishing aspect appears to be the multi-workload setting, but the paper does not clearly demonstrate that this leads to qualitatively different insights rather than a scaled-up version of existing regression benchmarking practice.
- The paper does not clearly establish **what limitations of prior benchmarks are resolved** by FORMULACODE, or why those benchmarks are insufficient for evaluating agentic optimization behavior.
- The primary leaderboard metric, **geometric mean speedup across workloads**, is reasonable but may hide significant regressions when the number of workloads is large. Although “worst regression” is reported, it is not integrated into the primary ranking criterion, leaving the trade-off between improvement and robustness somewhat unclear.
- The definition of the **advantage metric** (difference between agent and expert geometric-mean speedups) is not strongly justified. Since speedup is multiplicative, a comparison in log-space may be more principled and interpretable.
- The **statistical methodology** is underspecified. The paper does not clearly report the number of benchmark repetitions per workload, how runtime noise is handled, or whether any correction for multiple hypothesis testing is applied when filtering tasks using Mann–Whitney U tests across many workloads.
- Experiments are conducted on a subset (**FORMULACODE-V**), but the paper does not clearly describe how this subset is constructed or whether it is representative of the full benchmark. As a result, it is difficult to judge how well the reported conclusions generalize.
- The fairness of comparisons across **agent frameworks and LLM backbones** is not fully clear. Additional detail on tool budgets, iteration limits, token budgets, and stopping criteria would help isolate model capability from orchestration design.
- The benchmark construction pipeline includes **LLM-based repair of build failures**, which is practical but may introduce selection bias or environment-specific artifacts. The effect of this step on the final benchmark distribution is not analyzed.

---

> ### Author Rebuttal · Authors · 2026-03-29
>
> Thank you for the thoughtful review and detailed suggestions. We have revised the manuscript, including an expanded limitations/impact discussion, and clearer methodological details.
>
> We address the seven concerns the reviewer raises below.
>
> **Concern 1: Formulacode novelty / 'scaled up' [SWEfficiency and GSOBench]**
>
> We respectfully disagree that FormulaCode is only a “scaled-up” version of prior optimization benchmarks. Please see our response to Reviewer `6JPY` where we show a table with the many non-trivial insights enabled by FormulaCode’s construction which couldn’t be studied in any previous benchmark.
>
> **Concern 2: Unclear motivation for advantage**
>
> We introduce advantage to measure performance relative to the best-known expert solution. Raw speedup does not encode relative task difficulty: `speedup=1.05` may be trivial on an under-optimized codebase but very strong on an already highly optimized one.
>
> Advantage normalizes against the expert patch and has several desirable properties.
>
> 1. `Adv` calibrates for task difficulty by comparing against the best known solution.
> 2. If an agent simply memorizes the expert patch (e.g., due to training data contamination), it receives no spurious credit (`Adv=0`).
> 3. If the expert strategy improves performance at the cost of some regressions, then an agent following a similar strategy can still receive credit. Consider Appendix Figure 16 & 14. The advantage is calculated above each subplot. Here, Claude 4.0 Sonnet (Top Left) gets a better advantage than GPT-5 (Bottom Right) by making conservative optimizations that align with the equal-advantage line, whereas GPT-5 makes aggressive optimizations which experience more performance degradation, bringing its advantage down.
>
>
> **Concern 3: Primary Metric; Mean Adv v/s Log Adv**
>
> __The primary leaderboard metric is ranked pairs (RP) on the advantage__ (`Expt. 3.1`). Based on the reviewer's advice, we recomputed RP using log-transformed Adv in Table 1 (for OpenHands only; due to rebuttal space).
>
> | Model | LogAdv | RP |
> | :---- | :---- | :---- |
> | GPT 5 | -0.0167 | 3 |
> | Claude 4.0 Sonnet | -0.0130 | 1 |
> | Qwen 3 Coder | -0.0280 | 2 |
> | Human Expert | 0.0000 | - |
>
> The log-space transformation slightly compresses the spread, as expected, but doesn't affect the conclusions: the ranking of the configurations are stable under all regimes (for OH, Expert > Sonnet 4.0 > Qwen 3 Coder > GPT 5). This also held for median, trimmed mean, and log-space aggregations.
>
> Given this, we do not introduce log-space comparisons as we found RP on advantage to be more intuitive & interpretable.
>
> **Concern 4: FormulaCode-V details**
>
> For FormulaCode-V, Appendix `A.2` contain relevant details, which we've signficiantly expanded to include more construction details. The subset contains 108 problems, and we ensured the preservation of the repository distribution (necessary for long-tail experiments). There were minor PR characteristic differences which are highlighted in Table `7` and `8`.
>
>
> **Concern 5: Statistical filtering robustness.**
>
> We've expanded Appendix `B.1.6` which addresses most concerns. Briefly, *FormulaCode follows established practices in the Python performance benchmarking community for statistical robustness and performance filtering*.
>
> * *Repetitions.* The community emphasizes collecting sufficiently independent measurements over a finite duration rather than enforcing a fixed repetition count. asv collects 2 to 40 measurements/workload by default, but repository developers often override this on a workload-by-workload basis.
>
> * *Multiple comparisions.* We re-ran the filtering with a Holm-Bonferroni correction and lost 35/957 tasks (for 2025), none of which were in FormulaCode-V. The correction is quite aggressive becuase of ~260 workloads/task, yet barely changes the benchmark composition because the probability of an expert-written performance PR being merged without showing strong signal is low. We've incorporated this correction into our filtering procedure. Thank you for the valuable insight!
>
> **Concern 6: Harness limits/caps/exit/budgets.**
>
> We've expanded Appendix `B.2.5` which addresses most concerns. Briefly, *FormulaCode's harness ensures LLM evaluation fairness by using the same defaults as Terminal Bench, a popular SWE benchmark* with two changes: context compaction initiates after a 200k context limit is reached, and a six-hour hard cap is enforced, though no configuration reaches it (the average runtime is 38 minutes per agent per task).
>
> **Concern 7: LLM Repair bias.**
>
> We agree this step can introduce bias, but over a much smaller curation surface: the LLM helps repair an environment script and does not choose repos, PRs, or workloads. The alternatives expose a larger selection bias `[1]` (GSO/SWEficiency manually curate tasks from 10/9 repos and use synthetic workloads/ manually curated workloads).
>
> `[1]`: https://arxiv.org/abs/2112.01716 (NeurIPS 2021 D&B outstanding paper)

---

> > ### Author Rebuttal · Reviewer_apsu · 2026-04-03
> >
> > Thank you for the detailed rebuttal and additional clarifications. The added analyses strengthen the empirical rigor of the paper.
> >
> > However, my main concerns remain regarding the strength of the novelty claim (in particular, whether the proposed benchmark introduces a fundamentally new evaluation paradigm beyond scaled extensions of prior work), the lack of a principled justification for the proposed advantage metric, and the lack of fully controlled evaluation conditions across agent frameworks.
> >
> > Therefore, my overall evaluation and score remain unchanged.

---

> > > ### Author Response · Authors · 2026-04-05
> > >
> > > **We're glad to have resolved 4/7 concerns in the original rebuttal. We address the remaining 3 below.** The disagreement is narrower than the acknowledgment suggests. We apologize for not addressing your points clearly earlier (ICML's 5k char limit is tight). We hope the reviewer sees our good-faith efforts to address all concerns throughout the review process; and feels comfortable with a positive recommendation after our responses!
> > >
> > > > Does FormulaCode introduce a fundamentally new evaluation paradigm beyond scaled extensions of prior work?
> > >
> > > An evaluation paradigm can be novel by introducing new mechanisms or by measuring a new target. FormulaCode does the latter: it measures **multi-workload repository-level optimization**, with ~250 workloads per task.
> > >
> > > GSO, SWEfficiency, and SWE-Perf each evaluate optimization against **a single workload per task**. This is not a minor limitation. It makes an entire class of research questions inaccessible:
> > >
> > > 1. **Single-workload benchmarks cannot detect tradeoffs, and tradeoffs are a central challenge of real optimization:** The no-free-lunch theorem for optimization `[1]` establishes that performance gains on one workload can come at the cost of regressions on others. Many github discussions gravitate around this `[+]`. FormulaCode's ~250 workloads per task surface these tradeoffs directly. We find that expert solutions achieve the strongest overall gains *despite* causing an average 10% slowdown on some workloads (S3.5.2, Fig. 5). This provides an actionable signal for designing agents that negotiate such tradeoffs, which is undetectable in prior benchmarks.
> > >
> > > 2. **Single-workload benchmarks collapse agent behavior to a scalar.** FormulaCode's workloads are hierarchically structured (module, class, function), enabling stratified advantage profiles (S3.2). This reveals interesting performance profiles: Claude 4.0 Sonnet achieves stronger module-level advantage while GPT-5 performs better at the function level. Such findings cannot be formulated, let alone measured, with a single workload.
> > >
> > > 3. FormulaCode's novelty also extends to its unique construction which minimizes selection bias and enables long-tail and contamination experiments (see response to Reviewer `6JPY`).
> > >
> > > `[1]` No Free Lunch Theorems for Optimization. Wolpert and Macready, 1997
> > >
> > > `[+]` Within FormulaCode, PRs contain explicit multi-workload tradeoff discussions (e.g., `networkx #8023`, `pandas #58391`, `scikit-learn #25713`).
> > >
> > >
> > > > Lack of principled justification for advantage [v/s log-space]
> > >
> > > The intuition that multiplicative quantities should be compared in log-space is most relevant when ratios span orders of magnitude. This is not the case in FormulaCode as we have ~250 workloads per task.  **This is verified empirically: logAdv doesn't affect our observations after recomputing our analyses.** The below table shows the results under log-space. The relative ordering of models is unchanged, and the magnitudes are similar to the original analysis. This is because geomean speedups approach 1.0 as number of workloads increase, and, in this regime, Adv is approximately equal to LogAdv (it's exactly the first-order taylor expansion).  We are happy to include LogAdv results in the final version of the paper to address the principledness concern.
> > >
> > > | Model | LogAdv | Adv |
> > > | :---- | :---- | :---- |
> > > | GPT 5 | -0.0167 | -0.0209 |
> > > | Claude 4.0 Sonnet | -0.0130 | -0.0112 |
> > > | Qwen 3 Coder | -0.0280 | -0.0301 |
> > > | Human Expert | 0.0000 | - |
> > >
> > >
> > > > Lack of fully controlled evaluation conditions
> > >
> > > We ensure full control over evaluation conditions. Post-rebuttal, we modified Appendix `B.2.5` to contain an extensive multi-page hyperparameter table. We cannot share all details due to space. Here is an excerpt of relevant parameters. **All values follow community conventions and are used in industry standard code generation benchmarks `[3]`**; deviations are noted in our appendix and our rebuttal. We hope this addresses the evaluation control concerns.
> > >
> > > Key shared parameters for OpenHands (OH) and Terminus 2 (T2):
> > >
> > > | Param | Value |
> > > | :---- | :---- |
> > > | Temperature / Top-p | 0.7 / 1.0 |
> > > | Context management | Triggered at 200k context limit |
> > > | Max output tokens | Provider/LLM limit (controlled) |
> > > | Reasoning effort | "high" for supported models |
> > > | Stopping criteria | Self-termination or timeout |
> > > | Timeout per task | 6 hours (never reached) |
> > > | Retry policy | Retry on malformed response |
> > > | Tool budget | N/A; fully sandboxed |
> > > | Iteration caps | N/A; timeout-based |
> > >
> > > Agent-characteristic differences:
> > >
> > > | Param | T2 | OH |
> > > | :---- | :---- | :---- |
> > > | Available tools | Bash | Bash, `str_replace_editor`, IPython, think, finish |
> > > | Response parsing | Dedicated parsers | Native handling |
> > > | Browsing | N/A | Disabled |
> > > | Version | `terminal-bench` `d71d8fc` (2025-08-23) | v0.51.0 (2025-07-31) |
> > >
> > >
> > > `[3]`: Terminal-Bench: Benchmarking Agents on Hard, Realistic Tasks in Command Line Interfaces

---

### Official Review · Reviewer_6JPY · 2026-03-12

**Soundness:** 2
**Presentation:** 2
**Significance:** 3
**Originality:** 2
**Overall Recommendation:** 4
**Confidence:** 4

**Summary:**

This paper introduces FORMULACODE, a benchmark designed to evaluate the performance optimization capabilities of AI agents within large-scale, real-world codebases. The benchmark consists of 957 performance bottleneck tasks mined from 70 open-source scientific Python repositories. Distinct from prior benchmarks that primarily focus on functional correctness (e.g., bug fixing), FORMULACODE integrates the industrial performance testing tool asv (Airspeed Velocity) and proposes a multi-workload evaluation paradigm combined with statistical significance testing (Mann–Whitney U test) to measure an agent's optimization effectiveness under realistic engineering constraints.

**Compliance With Llm Reviewing Policy:**

Affirmed.

**Final Justification:**

The author has solved my problem, and I am inclined to accept it.

**Key Questions For Authors:**

1. Regarding Terminology Accuracy: In the Introduction, the authors list [SWEAgent, GSO, LiveCodeBench] as benchmarks. Are the authors conflating agent frameworks with benchmarks? Why was the widely recognized benchmark SWE-bench not cited in this context?

2. Regarding Domain Coverage: How could this framework be extended to other major Python domains—or even other programming languages—that do not rely on the asv benchmarking tool?

3. Regarding Hardware Sensitivity: Performance optimization outcomes may vary across different CPU architectures and memory configurations. How do the authors ensure that the reported Advantage metric remains valid and transferable across heterogeneous hardware environments?

**Limitations:**

yes

**Strengths And Weaknesses:**

Strengths:
1. Scientific Evaluation Framework: The paper elevates performance evaluation from simple execution-time comparisons to a more rigorous methodology incorporating statistical significance testing and extensive workload coverage, which improves the credibility of the efficiency metrics.

2. Dimensional Expansion: The benchmark extends optimization evaluation from isolated functions to repository-level environments, which better reflects real-world software engineering scenarios.

3. Innovation in Environment Construction: The framework for automatically generating reproducible Dockerized environments is valuable. In particular, leveraging LLMs to resolve complex version dependencies and regression issues in legacy codebases represents a meaningful engineering contribution.

Weaknesses:
1. Limited Incremental Innovation: The core idea appears largely as an extension of GSO (Global Software Optimization). The main contributions—integrating the asv tool and emphasizing repository-level evaluation—do not appear to introduce substantial algorithmic or conceptual novelty.

2. Domain Generalization Bias: Because the dataset heavily relies on repositories that already include asv benchmarking suites, the benchmark is concentrated primarily on scientific computing projects (e.g., Pandas, SciPy). The lack of coverage for other domains, such as web systems or distributed systems, limits the benchmark’s generality.

3. Issues in Terminology and Writing: In the Introduction, the authors repeatedly refer to the benchmark SWE-bench as SWE-agent (an agent framework). Such inaccuracies in terminology and literature referencing weaken the professional presentation of the manuscript.
Limited Evidence of Robustness Across Hardware: Performance measurements are often sensitive to hardware configurations. Although the paper includes statistical testing, the cross-platform and cross-architecture robustness of the results is not sufficiently validated.

---

> ### Author Rebuttal · Authors · 2026-03-29
>
> Thank you for the careful review and for highlighting the areas that needed stronger justification. We have revised the paper and added clarifying text in the manuscript.
>
> We address the four concerns the reviewer raises below.
>
> **Concern 1: FormulaCode has no novelty; asv wrapper; extension of GSO.**
>
> We respectfully disagree that FormulaCode is merely a wrapper around asv or a straightforward extension of GSO. We use Airspeed velocity as a measurement tool, analogous to how many software benchmarks build on existing tools (e.g. SWEBench & `pytest`). The table below showcases the many non-trivial insights enabled by FormulaCode’s unique construction which couldn’t be studied in any previous benchmark (GSO; SWEfficiency).
>
> | Contribution | Implications for the Community |
> | :---- | :---- |
> | **Contamination control.** Unlike GSO and SWEfficiency, FormulaCode is continually updated to mitigate GitHub-training contamination; we add **27 tasks /month** on avg. | LiveCodeBench showed that data contamination significantly inflates model performance.  We find *no significant trend* between a model’s cutoff date and the final performance `(Table 4)`, supporting the view that open-book optimization tasks are more resistant to memorization `[0]`.  |
> | **Repository diversity.** GSO and SWEfficiency draw from only 9–10 hand-curated popular repositories, which can induce selection bias `[5]`. FormulaCode instead samples **70 repositories** spanning both popular and bespoke projects.  | LLMs are known to struggle with long-tail knowledge `[1]`, but this had not been tested in code optimization. We find substantially worse performance on lower-popularity repository quintiles, providing direct evidence that long-tail repository knowledge is a bottleneck (`Table 3`). |
> | **Distributional Evals.** An easy way to game optimization benchmarks is to overfit to the measured workload while slowing down unmeasured workloads `[6]` . GSO and SWEfficeincy cannot guard against this as they measure a single workload for each codebase optimization task. FormulaCode instead measures **264.6 expert curated workloads /task** on avg, yielding a holistic performance profile. | Global performance optimization requires reasoning over a workload distribution, not a single input `[2,3,4]`. Consistent with this, `Figure 5` shows expert solutions achieve the strongest overall gains despite causing an average 10% slowdown on some workloads: a tradeoff prior benchmarks cannot reveal.|
> | **Holistic agent evaluation.** Because we measure many workloads, FormulaCode can compute module-, class-, and function-level performance profiles and is the first benchmark to study the holistic optimization behavior of LLM agents.   | `Figure 3` shows models differ qualitatively, not just in scalar score. Models demonstrate diverse performance profiles. Claude 4.0 Sonnet has higher module-level than function-level impact, while GPT-5 shows the reverse. Indeed, we use stratified advantage as a core public leaderboard metric to assess robustness (cc. Reviewer `apsu`). |
>
> **Concern 2: FormulaCode covers scientific computing domains.**
>
> We agree that domain coverage is still a limitation, and we state this more explicitly now.
>
> However, we believe the restriction should be interpreted relative to prior optimization benchmarks. GSO and SWEfficiency are built from just 9–10 repos and are also centered on scientific Python stacks such as NumPy, Pandas, etc.
>
> FormulaCode is broader (70 repos) and includes bespoke repositories that, to our knowledge, have not been meaningfully studied in prior mainstream code-generation benchmarks (`Qiskit`, a quantum simulation library, `Modin` a distributed systems lib for pandas, etc.).
>
> So while FormulaCode is not yet domain-universal, it is significantly broader than prior benchmarks in this line of work, and we have strengthened the limitations section to say that extending beyond scientific computing is an important next step.
>
> **Concern 3: Typos.**
>
> Thank you! Typos have been fixed. The two works are from the same group and resolved to similar names in my Overleaf.
>
> **Concern 4: Hardware heterogeneity.**
>
> Appendix `B.2.5` addresses this in detail. Briefly, our evaluation harness runs on identical AWS EC2 instances configured to provide isolated hardware resources in professionally managed data centers, ensuring third-party reproducibility and reducing uncontrollable variance. Advantage is computed on the same instance. We also tuned the setup for cost efficiency; the worst-case compute cost is ~$0.086/hour.
>
> `[0]`: https://arxiv.org/abs/1809.02789 (EMNLP2018)
>
> `[1]`: https://arxiv.org/abs/2211.08411 (ICML2023)
>
> `[2]`: https://dl.acm.org/doi/10.1109/ICSE48619.2023.00176 (ICSE2023)
>
> `[3]`: https://dl.acm.org/doi/abs/10.1145/1806596.1806647 (PLDI2010)
>
> `[4]`: https://dl.acm.org/doi/10.1145/2408776.2408791 (CACM2013)
>
> `[5]`: https://arxiv.org/abs/2112.01716 (NeurIPS2021 D&B outstanding paper)
>
> `[6]`: https://arxiv.org/abs/1606.06565

---

> > ### Author Rebuttal · Reviewer_6JPY · 2026-04-06
> >
> > The author has solved my problem. I raised my score. Thanks.

---

> > > ### Author Response · Authors · 2026-04-06
> > >
> > > I'm glad we were able to resolve your concerns regarding the novelty, implications, generality, and evaluation setup of FormulaCode!
> > >
> > > We've made the specific appendix sections more visible in the manuscript and made improvement to the limitations section. We've also added an extended discussions section in the appendix about our observations and their implications.
> > >
> > > Thank you for your insights throughout the rebuttal process!

---

### Official Review · Reviewer_2WrW · 2026-03-13

**Soundness:** 4
**Presentation:** 4
**Significance:** 4
**Originality:** 4
**Overall Recommendation:** 6
**Confidence:** 4

**Summary:**

This paper introduces a new benchmark for coding agents, involving ~1K tasks to fix performance issues in scientific python repos, with expert patches & fine-grained multi objective metrics on performance optimisation in addition to correctness. It benchmarks several leading coding agents and shows that they still struggle with repository level optimisation.

**Compliance With Llm Reviewing Policy:**

Affirmed.

**Final Justification:**

Overall this is an extremely valuable piece of work and a benchmark that I imagine will be widely used and valuable, especially as we move into heavier automatic research loops with coding agents & the focus shifts onto performance/efficiency over and above correctness.

**Key Questions For Authors:**

- do the authors have any way to detect exact or near-exact matches between agent outputs and github history, to rule out pre-training contamination?
- how can we understand/calibrate the difficulty of expert patches?

**Limitations:**

yes

**Strengths And Weaknesses:**

This paper introduces a new benchmark for coding agents, involving ~1K tasks to fix performance issues in scientific python repos, with expert patches & fine-grained multi objective metrics on performance optimisation in addition to correctness. It benchmarks several leading coding agents and shows that they still struggle with repository level optimisation.

The paper would make an extremely valuable addition to coding agent benchmarks. The focus on agents' ability to optimise performance - not just match correctness/pass tests - clearly makes this a challenging set of tasks, as evidenced by the difficulty that frontier agents have at repo-level optimisation. The paper is well written, extremely clear, the metrics and evals are well considered, and the authors have comprehensively compared it with other related benchmarks/works.

---

> ### Author Rebuttal · Authors · 2026-03-29
>
> Thank you for the positive review and the suggestions. We address your concerns below.
>
> **Do the authors have any way to detect exact or near-exact matches between agent outputs and github history, to rule out pre-training contamination?**
>
>
> We can partially assess this at the level of patch overlap. For each task, we retain both the model-submitted performance-improving patch and the corresponding expert patch from GitHub. For every task and agent, we compute a normalized edit distance between the expert-written patch and agent patch using Python's inbuilt `difflib`. The table below reports the mean normalized edit distance (0 = identical patches and 1 = no similarity), grouped by whether the agent’s advantage is within $\pm 0.05$ (advantage close to 0 might indicate behavior similarity).
>
> | Model | $\|\texttt{Adv}\| \leq 0.05$ | $\|\texttt{Adv}\| > 0.05$ |
> | :---- | :---- | :---- |
> | Claude 4.0 Sonnet | 0.94 | 0.97 |
> | GPT 5 | 0.96 | 0.96 |
> | Gemini 2.5 Pro | 0.98 | 0.95 |
> | Qwen 3 | 0.96 | 0.97 |
>
> Across all configurations, agent and expert patches share only ~4% of their content, suggesting that verbatim memorization is rare.
>
> This experiment provides a useful signal, though it has some inherent limitations. It helps rule out exact or near-exact patch-level memorization, but it does not rule out all forms of contamination. For example, a model may recover the same high-level optimization idea while expressing it with substantially different code. In such cases, executing the patch and measuring its advantage is valuable because it captures *behavioral similarity* even when surface-level code overlap is low `[1]`.
>
> **How can we understand/calibrate the difficulty of \[expert patches / task\]?**
>
> We agree this is an important question, and we do not currently claim to directly measure the human effort required to discover the expert solution. Instead, we utilize difficulty-related surrogates.
>
> **GitHub stars**. Repository popularity is an imperfect but useful proxy for repository maturity and scrutiny. More widely used repositories are likely to have had more optimization opportunities already discovered and removed. Consistent with this intuition, in Experiment 3.4 we find that more popular repositories tend to have expert PRs with lower average global speedup, suggesting fewer remaining "low-hanging" optimization opportunities.
>
> **Speedup of the expert patch.** The speedup achieved by the expert patch is a proxy for the amount of *optimization headroom* present in the base commit. Larger expert gains indicate that the underlying task exposes a more impactful optimization opportunity, while smaller gains suggest more limited measurable headroom. This is one reason we report advantage relative to the expert: it helps normalize agent performance against the magnitude of the opportunity available in each task.
>
> We have revised the manuscript to make this distinction clearer.
>
>
> `[1]`: https://arxiv.org/abs/2107.03374

---

> > ### Author Rebuttal · Reviewer_2WrW · 2026-04-03
> >
> > Thank you for this response! The explanations here make sense, and add further clarity to an already great manuscript.

---

> > > ### Author Response · Authors · 2026-04-06
> > >
> > > I'm glad we were able to resolve all your concerns! I've added the memorization study to the manuscript as well as a discussion section on difficulty calibration.
> > >
> > > Thank you for your insights throughout the rebuttal process!

---

### Official Review · Reviewer_yFkg · 2026-03-17

**Soundness:** 3
**Presentation:** 4
**Significance:** 4
**Originality:** 3
**Overall Recommendation:** 5
**Confidence:** 4

**Summary:**

This paper introduces FORMULACODE, a benchmark for evaluating agentic optimization on large, real-world codebases with fine-grained, multi-objective performance metrics. Each task in FORMULACODE comprises of a problem description of a performance regression, a containerized environment containing a snapshot of the baseline repository, and multiple crowdsourced performance workloads, along
with the tools to execute them. An agent’s performance improving edits are assessed based on the agent’s ability to optimize
multiple workloads while ensuring various forms of correctness guarantees are met.

The evaluations reveal that although agents generally can improve run-time performance, repository-scale and multi-objective optimization remains a major challenge for frontier LLM agents.

**Compliance With Llm Reviewing Policy:**

Affirmed.

**Key Questions For Authors:**

1. Can you give more analysis details on the disagreement between the adv and speedup metrics?
2. Missing related work: "SWE-Perf: Can Language Models Optimize Code Performance on Real-World Repositories?"

**Limitations:**

yes

**Strengths And Weaknesses:**

Strength
FORMULACODE is the first benchmark specifically designed to evaluate multi-workload optimization at a repository scale. By utilizing a rigorous four-stage automated pipeline, FORMULACODE ensure that each of the 957 tasks represents a high-quality, reproducible performance bottleneck. Furthermore, the pipeline’s ability to generate 27 new tasks monthly ensures that FORMULACODE remains a dynamic, live benchmark that evolves alongside contemporary software practices. The paper effectively communicates complex concepts through clear visuals and a novel suite of metrics, such as Stratified Advantage, which reveals how agents perform across different levels of the code hierarchy. The evaluation findings of this paper is insightful to the community.

Weakness
A notable limitation of the current evaluation is the exclusion of more advanced agent frameworks, such as Claude Code or Codex, along with the latest frontier LLMs. This omission potentially limits the robustness of the comparative analysis and the generalizability of the paper’s findings.

---

> ### Author Rebuttal · Authors · 2026-03-31
>
> Thank you for the positive review and the suggestions. We address your concerns below.
>
> **Concern 1: Why were advanced agents, like Claude Code, not included in FormulaCode?**
>
> Thank you for raising this. Our harness does support installable agents such as Claude Code and Codex. Our initial evaluation prioritized maximally controlled comparisions across agent frameworks and model families. To address your concern, we ran an additional controlled framework comparison on a 20-task development subset of FormulaCode-V (tasks where humans achieve a high speedup across diverse repositories; we have added details to our manuscript). We paired Claude Code, OpenHands, and Terminus 2 with the same model (Claude 4.0 Sonnet, `claude-sonnet-4-20250514`) to isolate the effect of agent framework choice. The results are shown below.
>
> | Agent + Model | Adv | Speedup | Ranked Pairs |
> |:---|---:|---:|---:|
> | OpenHands + Claude 4.0 Sonnet | -0.0388 | 1.1105 | 1 |
> | Terminus 2 + Claude 4.0 Sonnet | -0.0594 | 1.0963 | 2 |
> | Claude Code + Claude 4.0 Sonnet | -0.0493 | 1.1200 | 3 |
>
>
> All three agents have negative advantage on this subset, confirming that these tasks remain challenging regardless of framework. Claude code demonstrated a surprisingly high-variance profile. It attained the highest speedup but ranked lowest in Ranked Pairs and trailed in advantage as well. Qualitative analysis revealed that when Claude Code correctly identified the performance-critical hotspot, it produced large speedups — but it frequently stopped after a shallow optimization, targeted functions outside the hot path, and introduced correctness failures. These failure modes led to many task-level losses in pairwise comparisons against OpenHands and Terminus 2, which achieved modest but consistent improvements. Advantage was negatively impacted for the same reason.
>
> This experiment indicates that the main bottleneck for such agents lies less in their peak capability to optimize codebases but rather in their _consistency_ to deliver meaningful optimizations in diverse codebases. One Caveat: to ensure a homogeneous comparison, we pin all three frameworks to the same compatible agent version (early September 2025) and same model (Claude 4.0 Sonnet). It remains to be seen if our observation generalizes to the latest flagship agents. We have secured additional funding and are currently in the process of evaluating the latest flagship agents and models on FormulaCode-V. These runs will be published to our public leaderboard.
>
>
> **Concern 2: disagreement between the adv and speedup metrics**
>
> Speedup measures absolute performance improvement but doesn't account for relative task difficulty. Advantage normalizes an agent's improvement by the expert's improvement on the same task. As a result, a large speedup on a task where the expert also achieved a large speedup contributes less to advantage than the same speedup on a task where the expert found little improvement.
>
> In Table 1, under the OpenHands agent, GPT-5 achieves a higher speedup than Claude 4.0 Sonnet but a lower advantage. Examining the per-task results, GPT-5's largest speedups concentrate on tasks where the expert patch also produced large speedups, suggesting it tends to find optimizations that overlap with expert strategies on already well-optimized code paths. Claude 4.0 Sonnet's speedups, while smaller in absolute terms, are distributed more toward tasks where the expert patch achieved modest improvements, indicating it surfaces gains in regions of the codebase that were harder for experts to optimize.
>
> We have added a more detailed explanation of this in the manuscript.
>
> **Concern 3: Missing related work: SWE-Perf**
>
> Thank you for pointing us to SWE-Perf. We have added it to related work. SWE-Perf is a closely related benchmark, comprising 140 instances derived from performance-improving PRs across 9 Python repositories. SWE-Perf measures performance improvement via unit test runtime and verifies correctness through a targeted set of test cases. FormulaCode takes a different approach. Rather than relying on unit test runtime as a proxy for performance, each of our 957 tasks is paired with an average of ~265 developer-maintained, performance-centric workloads, alongside explicit repository-level pytest and snapshot correctness tests. This design choice reflects our emphasis on measuring codebase optimization across the diverse execution performance-critical paths identified by domain experts, rather than inferring performance gains from unit test execution time. FormulaCode also draws from a broader base of 70 scientific Python repositories. We hope that FormulaCode advances the direction that SWE-Perf has also contributed to, and we thank the reviewer for helping us better contextualize our contribution.

---

> > ### Author Rebuttal · Reviewer_yFkg · 2026-04-04
> >
> > I thank the authors for the response. It addresses most of my concerns. I will keep my ratings. Good luck.

---

> > > ### Author Response · Authors · 2026-04-06
> > >
> > > Thank you! I'm glad we were able to resolve your concerns. I've added a metrics discussion section, and SWE-Perf to the related work. We continue to add frontier agents/models to FormulaCode. Again, thank you for your insights throughout the rebuttal process!

---

### Decision · Program_Chairs · 2026-04-30

**Decision:**

Accept (regular)

**Comment:**

This paper makes a strong benchmark contribution by moving coding-agent evaluation into repository-scale, multi-workload performance optimization, which is both practically important and not well covered by existing benchmarks. Reviewers consistently valued the scale of the benchmark, the care in environment construction, and the fact that the evaluation goes beyond simple correctness to capture optimization tradeoffs that matter in real codebases.